# Single-cell transcriptomics reveal how root tissues adapt to soil stress

Mingyuan Zhu[1,2,9], Che-Wei Hsu[1,2,9], Lucas L. Peralta Ogorek[3], Isaiah W. Taylor[1,2], Salvatore La Cavera[4], Dyoni M. Oliveira[5,6], Lokesh Verma[3], Poonam Mehra[3], Medhavinee Mijar[1,2], Ari Sadanandom[7], Fernando Perez-Cota[4], Wout Boerjan[5,6], Trevor M. Nolan[1,2,8], Malcolm J. Bennett[3✉], Philip N. Benfey[1,2,10] & Bipin K. Pandey[3✉]

Land plants thrive in soils showing vastly different properties and environmental stresses[1]. Root systems can adapt to contrasting soil conditions and stresses, yet how their responses are programmed at the individual cell scale remains unclear. Using single-cell RNA sequencing and spatial transcriptomic approaches, we showed major expression changes in outer root cell types when comparing the single-cell transcriptomes of rice roots grown in gel versus soil conditions. These tissue-specific transcriptional responses are related to nutrient homeostasis, cell wall integrity and defence in response to heterogeneous soil versus homogeneous gel growth conditions. We also demonstrate how the model soil stress, termed compaction, triggers expression changes in cell wall remodelling and barrier formation in outer and inner root tissues, regulated by abscisic acid released from phloem cells. Our study reveals how root tissues communicate and adapt to contrasting soil conditions at single-cell resolution.

Crops such as rice thrive in arable soils that show natural heterogeneity. The heterogeneous nature of soils is characterized by uneven distributions of nutrients, water, microorganisms and organic content that pose a stark contrast to the uniformity of growth media, underscoring the fundamental importance to understand how plant roots navigate, adapt and thrive at molecular and cellular levels in natural soils. Roots have evolved diverse strategies to tackle the variability in soil conditions[1,2]. However, our understanding of how roots respond to complex soil environments at a cellular level of resolution remains limited. The application of single-cell RNA sequencing (scRNA-seq) and spatial transcriptomic approaches to plant organs grown in diverse environments has the potential to reveal gene-expression complexity throughout root developmental stages, and identify mechanisms governing cell type-specific responses to environmental stresses[3,4]. Soil stresses represent a major challenge in global agriculture[5]. For example, soil compaction stress reduces root penetration, thereby affecting nutrient and water uptake and subsequently crop yield[6]. Roots have developed adaptive growth responses for compacted soils; but the underlying genes, cell group-specific transcriptional responses and molecular mechanisms remain poorly understood. To discover the mechanisms governing root responses to soil compaction at a cellular resolution, we pioneered transcriptomic profiling of rice root tissues grown in soils with and without compaction using single-cell approaches.

## Rice root scRNA-seq and spatial transcriptomic atlas

Protoplasts of rice primary roots, obtained from gel-based conditions, were initially adopted to generate a high-quality scRNA-seq reference dataset to show cell identities and differentiation trajectories[7], and later compared to equivalent datasets generated from soil-grown roots. Using the 10X Genomics scRNA-seq platform, we profiled more than 47,000 root cells gathered from Xkitaake rice primary root tissues 2–3 days after germination across ten sets of independently grown seedlings. To enhance depth and assess data variability across laboratories, we integrated a previously published scRNA-seq dataset[8] with our datasets. All datasets underwent processing with the COPILOT (cell preprocessing pipeline kallisto bustools) pipeline[9], resulting in the integration of more than 79,000 high-quality cells to construct the final scRNA-seq atlas (Fig. 1a,b, Supplementary Video 1, Supplementary Table 1 and Supplementary Data 1). To mitigate the effect of protoplasting on gene expression, we identified protoplasting-induced genes by means of bulk RNA sequencing (RNA-seq, Supplementary Table 2) and excluded them from data integration and differential expression analysis. This approach ensured the robustness of our scRNA-seq findings.

To ensure representation from all major developmental stages, we assigned developmental annotations to major cell types by comparing each cell's transcriptome with bulk RNA-seq expression profiles of manually dissected root tissue segments corresponding to meristematic, elongation and maturation zones (Extended Data Fig. 1a and Supplementary Data 2). scRNA-seq studies often rely on pseudotime analysis to infer developmental stages, a computational method that orders cells on the basis of gene expression similarities but does not represent actual time. This approach is influenced by the choice of the starting point, which can affect interpretations. In our study, we integrated bulk RNA-seq data from rice root tissues at distinct developmental stages. Using stage-specific marker genes, we directly annotated developmental

[1]Department of Biology, Duke University, Durham, NC, USA. [2]Howard Hughes Medical Institute, Duke University, Durham, NC, USA. [3]Plant and Crop Science Department, School of Biosciences, University of Nottingham, Nottingham, UK. [4]Optics and Photonics Group, Faculty of Engineering, University of Nottingham, Nottingham, UK. [5]Department of Plant Biotechnology and Bioinformatics, Ghent University, Ghent, Belgium. [6]VIB Center for Plant Systems Biology, Ghent, Belgium. [7]Department of Biosciences, University of Durham, Durham, UK. [8]Division of Biology and Biological Engineering, California Institute of Technology, Pasadena, CA, USA. [9]These authors contributed equally: Mingyuan Zhu, Che-Wei Hsu. [10]Deceased: Philip N. Benfey. ✉e-mail: malcolm.bennett@nottingham.ac.uk; philip.benfey@duke.edu; bipin.pandey@nottingham.ac.uk

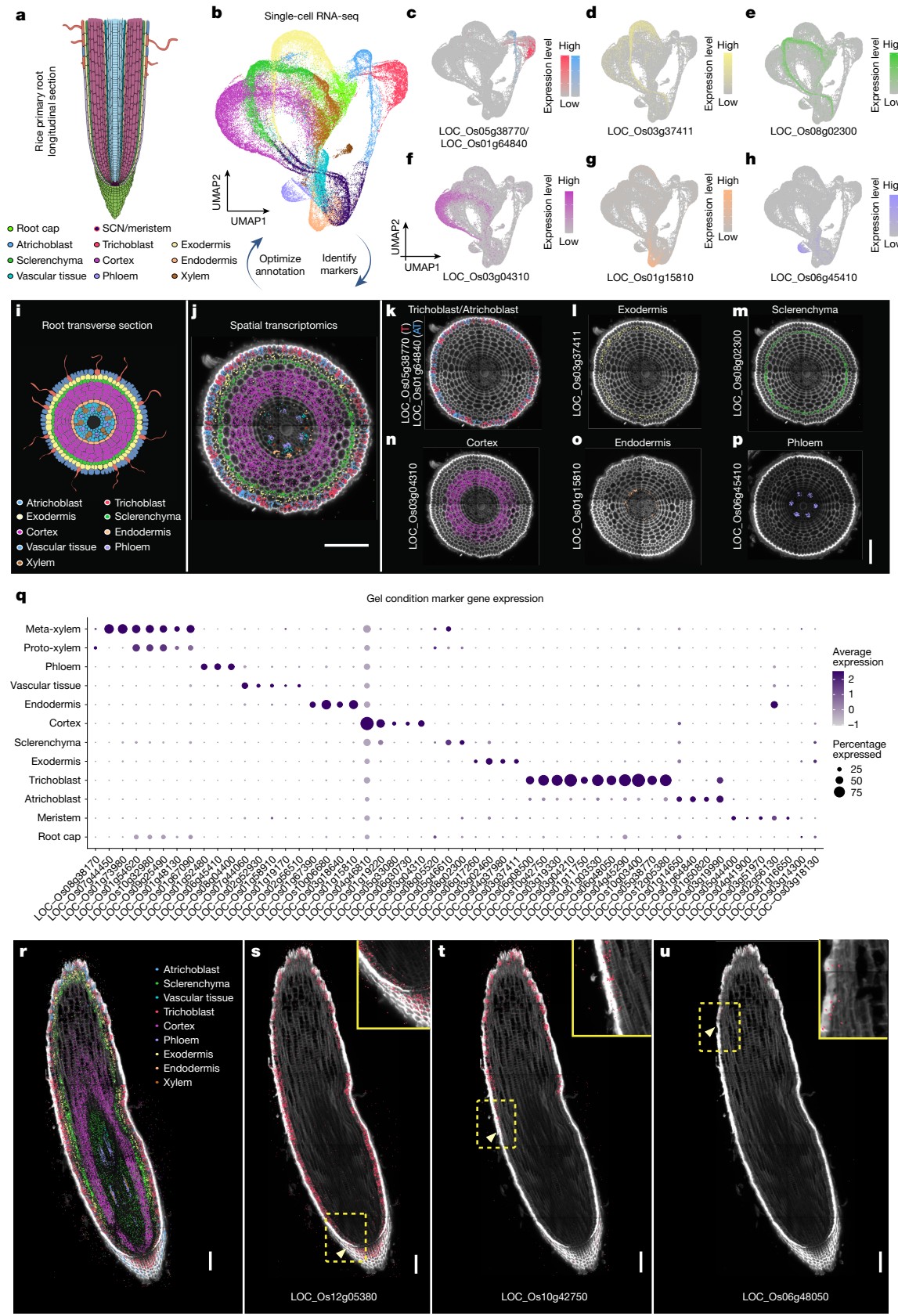

**Fig. 1** | See next page for caption.

stages in our scRNA-seq dataset, aligning it more closely with experimental observations. This developmental-stage annotation is unique in our scRNA-seq dataset, compared to the previously published ones[8,10,11].

High Spearman correlation among transcriptional profiles of all samples confirmed minimal batch effects (Extended Data Fig. 1b). For cell type annotation, we initially used principal component analysis

**Fig. 1 | scRNA-seq and spatial transcriptomic analysis reveals trajectories and markers for rice root cell types. a**, Illustration of rice primary root anatomy with different cell types highlighted. The stem cell niche (SCN), initial cells, daughter cells and other meristematic cells are labelled as SCN/meristem. Non-conducting stele tissues (pericycle, procambium and vascular ground tissue) are annotated as vascular tissue. **b**, UMAP visualization of major root cell clusters, with each dot representing a single cell. **c–h**, Expression of identified cell type markers in scRNA-seq data, with colour scales indicating normalized, corrected UMI counts: trichoblast and/or atrichoblast (**c**), exodermis (**d**), sclerenchyma (**e**), cortex (**f**), endodermis (**g**) and phloem (**h**). **i**, Schematic of the rice primary root transverse section. **j**, Spatial transcriptomic visualization of major cell type markers in transverse root sections. Each dot represents a detected mRNA molecule, colour-coded by cell type. **k–p**, Spatial expression of cell type markers in 5-day-old rice roots using Molecular Cartography:

trichoblast and/or atrichoblast (**k**), exodermis (**l**), sclerenchyma (**m**), cortex (**n**), endodermis (**o**) and phloem (**p**). *n* = 9 biological replicates. **q**, Dot plot of cell type marker expression in gel samples. Dot size indicates the percentage of cells expressing each gene, and colour intensity represents average scaled expression. The full marker gene and their annotation list is in Supplementary Table 3. **r**, Visualization of major cell type marker expression in rice root longitudinal sections. Each dot denotes a detected mRNA molecule, with different colours denoting different cell types. **s–u**, Spatial analysis of trichoblast markers LOC_Os12g05380, *OsGT3* (**s**); LOC_Os10g42750, *OsCSLD1* (**t**); LOC_Os06g48050 (**u**), with detected mRNA molecules shown in red. Yellow arrowheads indicate the earliest expression along the proximal–distal root axis. Insets highlight expression initiation regions (yellow boxes, **s**,**t** 2X and **u** 3X). *n* = 3 biological replicates for gel-grown root longitudinal sections. Scale bars, 100 μm. Panels **a**,**i** adapted with permission from Xiaoying Zhu.

(PCA) and clustering techniques, followed by the calculation of *z*-scores for published markers (Methods). However, because of the limited number of validated cell type markers, a substantial proportion of cells remained unannotated. To address this limitation, we used Molecular Cartography, an optimized multiplexed fluorescence in situ hybridization technology[12,13]. This allowed us to explore in situ gene-expression profiles for the candidate cell type markers identified in our putatively annotated cell type clusters. Our spatial transcriptomic experiments validated the cell type-specific expression of more than 40 markers (Fig. 1i–p, Extended Data Fig. 2, Supplementary Table 3 and Supplementary Data 3 and 4). We then refined our cell type annotation, relying on the expression patterns of validated markers in scRNA-seq clusters (Fig. 1c–h,q). The iterative feedback loop between scRNA-seq and spatial transcriptomics significantly increased the number of reliable markers for the major root cell types, enhancing cell annotation quality.

Integrating scRNA-seq and spatial transcriptomic data also showed temporal gene-expression dynamics. Pseudotime analysis demonstrated a continuous differentiation trajectory of rice root epidermis (Extended Data Fig. 1c,d), and we examined temporal expression patterns of differentially expressed genes (DEGs) involved in root hair differentiation (Extended Data Fig. 1e). Among the DEGs, three genes (LOC_Os12g05380, *OsGT3*, encoding a putative xylosyltransferase (XXT); LOC_Os10g42750, *OsCSLD1*, encoding Cellulose Synthase Like D1; LOC_Os06g48050, unannotated) showed a sequential pattern of expression along root hair differentiation, detected in both scRNA-seq data and spatial transcriptomics data (Fig. 1r–u and Extended Data Fig. 1f–k). Overall, our approach generated a high-resolution scRNA-seq atlas, with confidently annotated cell types and developmental stages.

## Soil-grown roots modify expression in outer tissues

To investigate the cell type-specific responses to natural soil condition relative to the gel-based condition, we used our standardized soil-based growth regime[14]. Xkitaake rice seedlings were cultivated in soils for 3 days, their roots were then harvested, after which scRNA-seq was conducted on protoplasts isolated from two biological replicates of 1-cm root tip segments. Root tip samples encompassed meristem, elongation and early maturation zones. To leverage our gel-based data for interpreting our soil-based data, gel-based and soil-based scRNA-seq datasets were integrated (Methods). Although the scRNA datasets obtained from different growth conditions could be distinguished with sample-wise correlation analyses, the gel-grown roots showed high one-to-one similarities with the soil-grown roots across almost all cell clusters (Extended Data Fig. 3b,c,h). Most validated cell type markers detected in our Molecular Cartography examination remained expressed in their target cell types under soil conditions (Fig. 2d–l, Extended Data Fig. 5a and Supplementary Data 5 and 6). We thus relied on the expression patterns of these marker genes to annotate the major cell types in our soil-based scRNA-seq data (Extended Data Fig. 3a,b and Supplementary

Table 4). We verified the relatively high correlation of individual cell types between the two growth conditions (Extended Data Fig. 3d,f). Our scRNA-seq analysis showed a notable decrease in the number of root hair cells detected under soil conditions. To verify this reduction in trichoblast cell numbers, we examined the expression of a root hair cell-specific marker line (*proCSLD1::VENUS-N7*). Imaging showed highly similar expression patterns between gel and soil conditions, indicating that the observed decrease is probably due to the loss of root hair cells during the protoplasting process (Extended Data Fig. 3i–k). In conclusion, we managed to annotate our soil-based scRNA-seq data with our knowledge gained from both gel-based scRNA-seq data and spatial transcriptomic application to soil-grown roots.

To delve into the mechanisms governing cell type-specific responses to soil growth conditions, we conducted differential expression analysis for confidently annotated root cell types and developmental stage-enriched groups (Methods). This analysis revealed 11,259 DEGs (fold change greater than 1.5, false discovery rate less than 0.05, Supplementary Table 5). Notably, 31% of DEGs were altered in a single cell type or developmental stage, indicating that changes in growth conditions modulate distinct sets of genes in specific cell type contexts (Extended Data Fig. 5m,n). Most DEGs were found in the outer root cell types (epidermis, exodermis, sclerenchyma and cortex), whereas the inner stele layers (such as phloem and endodermis) showed relatively minor changes (Fig. 2a–c,m and Extended Data Fig. 5m). This pattern indicates that even under non-stressed soil conditions, roots modify their gene expression compared to gel-grown roots, particularly in outer cell layers.

Gene ontology (GO) analysis of these DEGs revealed the functional classes enriched in outer tissues of soil-grown roots notably include nutrient metabolism (particularly phosphate and nitrogen pathways), alongside vesicle-mediated transport, cell wall integrity, hormone-mediated signalling and defence responses, compared to axenic gel conditions (Fig. 2n–p, Supplementary Table 6). The increased expression of genes involved in nitrogen and phosphorus metabolism in outer cell layers further indicates that root cells dynamically adjust their metabolic processes to respond to fluctuating nutrient availability in soils. We also identified several micro nutrient (zinc and boron) uptake-related genes (*OsZIP10* and *OsBOR1*) showing enhanced expression in the outer cell layers. Our scRNA-seq analysis indicates that roots use various adaptive strategies to improve nutrient uptake, including strengthening cell wall integrity, enhancing cell communication by means of hormone signalling and using vesicle-mediated transport mechanisms, in response to the heterogeneous distribution of nutrients in the soils.

In our study, we used the model rice variety Xkitaake to establish our scRNA-seq resource given the wealth of functional resources available in this background including mutant collections[15] and exploited in our recent study[2]. However, Xkitaake is a transgenic line containing the *XA21* gene, which encodes a plasma membrane-localized protein that

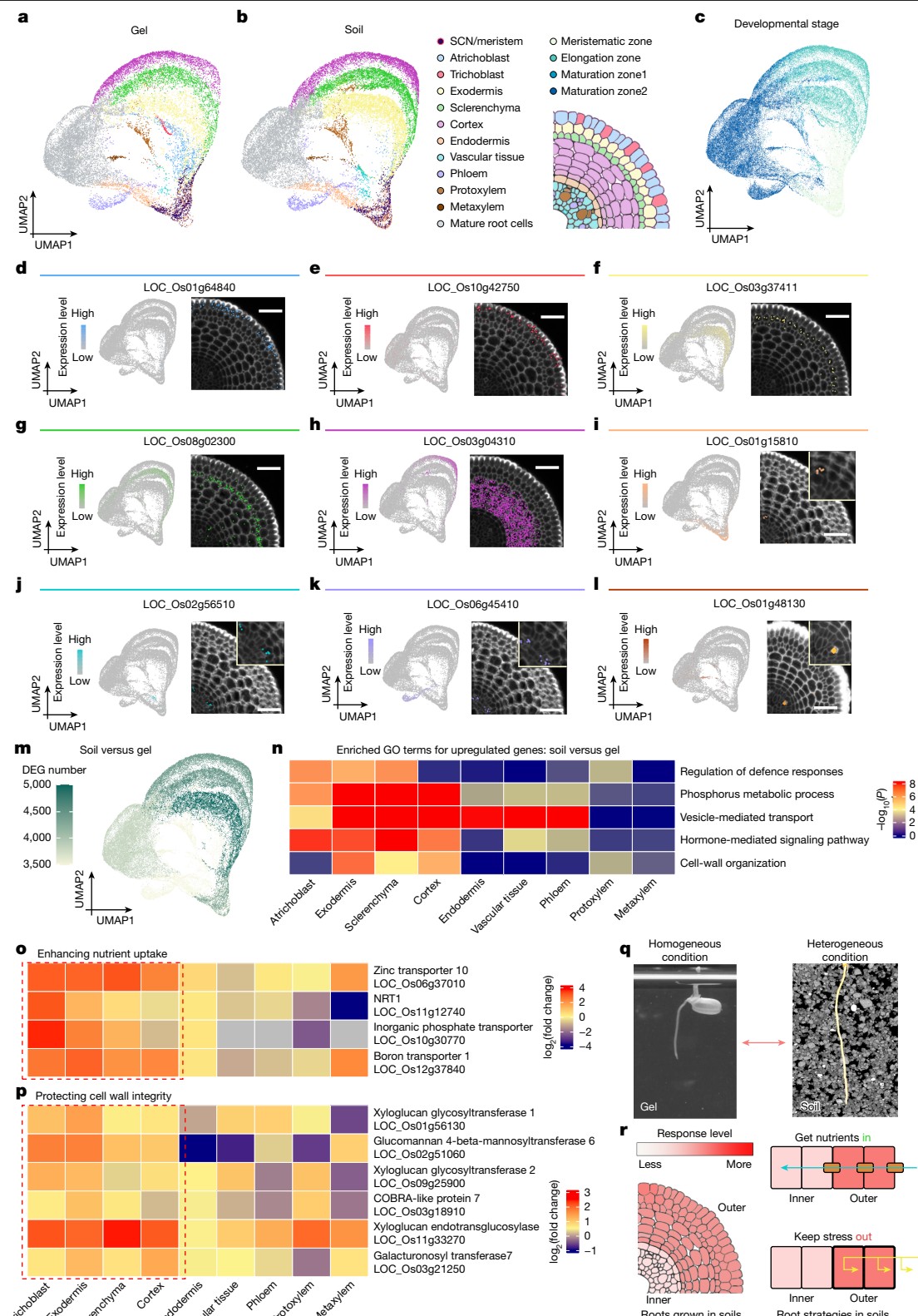

**Fig. 2** | See next page for caption.

confers resistance to *Xanthomonas oryzae* pv. *oryzae* (Xoo) in rice. To assess the potential influence of *XA21* on rice root gene expression, we conducted scRNA-seq on Kitaake genotype under both gel and soil conditions (Extended Data Fig. 4a–e). Cell type annotation revealed similar enrichment of DEGs in outer root cells as observed in Xkitaake (Extended Data Fig. 4a and Supplementary Tables 7 and 8). We further

validated the enriched GO terms through a comparative scRNA-seq analysis of the Kitaake genotype. The GO term enrichment patterns and associated gene expression changes (Supplementary Table 9), related to nutrient homeostasis, cell wall integrity, hormone-mediated signalling and vesicle-mediated transport were consistent between Xkitaake and Kitaake in the scRNA-seq analysis (Extended Data Fig. 4c–e).

 

**Fig. 2 | In comparison to artificial gel, growth in soils induces differential gene expression in outer root cell layers. a,b**, UMAP projection of scRNA-seq from 21,356 cells from roots grown in gel (**a**) and 27,744 cells from roots grown in soils (**b**). Colours indicate cell type identity. **c**, UMAP projection with developmental-stage annotations, based on bulk RNA-seq data from rice roots grown in gel. **d–l**, Cell type marker expression in scRNA-seq and spatial data from roots in non-compacted soils: atrichoblast (**d**), trichoblast (**e**), exodermis (**f**), sclerenchyma (**g**), cortex (**h**), endodermis (**i**), vascular tissue (**j**), phloem (**k**) and xylem (**l**). Colour scale in feature plots shows normalized UMI counts. Spatial transcriptomics show marker expression in transverse sections, with dots representing detected mRNA molecules. Insets show magnified views (**i** ×1.8, **j–l** ×1.6). *n* = 4 biological replicates. Full marker gene list is in Supplementary Table 3. **m**, UMAP visualization of DEG numbers in gel versus soil conditions. The outer cell layers (exodermis, sclerenchyma and cortex) have more DEGs compared to the inner cell layers (endodermis and stele). **n**, Top enriched GO terms for upregulated genes in soil-grown roots include defence response, phosphorus metabolism, vesicle transport, hormone signalling and cell wall organization, mainly in outer cell layers. *P* values were calculated using a one-tailed hypergeometric test with g:Profiler2 g:SCS for multiple comparison correction. **o,p**, Nutrient uptake (**o**) and cell wall strengthening genes (**p**) are induced in soil-grown roots, particularly in epidermis, exodermis, sclerenchyma and cortex (red box), highlighting their role in adapting to heterogeneous soil environments. Grey boxes indicate genes not detected in the analysis. **q**, Schematics illustrating rice roots grown in homogeneous gel versus heterogeneous soils. **r**, Single-cell transcriptomics indicate that outer cell layers respond more to soil heterogeneity, enhancing nutrient uptake to support root development while mitigating local stress effects on growth. Scale bars, 50 μm.

For defence responses, scRNA-seq analysis of Xkitaake compared to Kitaake showed relatively higher expression of defence-related genes in Xkitaake (Extended Data Fig. 4h,i), indicating that *XA21* can enhance defence responses under changing growth conditions. However, when examining specific defence response genes in Kitaake under soil versus gel conditions, we also detected their induction in soil conditions (Extended Data Fig. 4f,g,j,k). Thus, although *XA21* is not required for the enhanced defence response observed in natural soils as compared to the gel growth regime, it amplifies the defence response triggered by these growth condition changes.

Hence, compared to when propagated in sterile homogeneous gel, roots grown in soils seem to adapt to their heterogeneous environment by upregulating defence, nutrient and cell wall-related gene expression across all the cell types. The outer cell layers are more responsive compared to the inner cell layers, reinforcing nutrient uptake (that is, 'get nutrients in') and cell wall integrity, to facilitate root exploration for heterogeneous resources in soil (Fig. 2q,r). This cell layer-specific responsiveness also helps to protect developing roots from abiotic and biotic signals (that is, 'keep stress out') that are unevenly distributed in natural soils (Fig. 2q,r). These important insights highlight the benefit of applying single-cell profiling approaches on samples grown in a natural soil environment.

## Soil compaction triggers root ABA and barrier formation

Root systems can adapt to contrasting soil stresses, yet how their responses are programmed at the individual cell scale remains unclear. Soil compaction reduces the ratio of air spaces versus soil particles, resulting in higher mechanical strength that impedes root growth and triggers adaptive responses[6]. To show how individual root cell types exposed to compaction stress modify their gene-expression profiles, scRNA-seq and spatial transcriptomic datasets were generated from roots grown at higher soil bulk density (1.6 g cm$^{-3}$ compared to 1.2 g cm$^{-3}$; Fig. 3a–c and Methods). Molecular Cartography revealed most validated cell type markers remained expressed in their target cell types under compacted soil growth conditions (Fig. 3c, Extended Data Fig. 5b–l and Supplementary Data 7 and 8). This is consistent with the detected high correlation between two soil conditions across most cell layers (Extended Data Fig. 3e,g).

Next, we performed a comparative analysis to determine the most transcriptionally affected cell groups and understand the nature of their responses to soil compaction stress. We further checked the DEGs for each confidently annotated root cell type and developmental stage-enriched groups. We identified 7,947 DEGs (fold change greater than 1.5, false discovery rate less than 0.05). Here 42% of DEGs were altered in a single cell type or developmental stage (Extended Data Fig. 6a–d). Notably, exodermis and endodermis emerged as the two cell layers particularly influenced by soil compaction, showing the highest number of DEGs (Fig. 3d, Extended Data Fig. 6b and Supplementary Table 10).

Analysing enriched GO terms for the most affected cell types, exodermis and endodermis, showed a significant association with cell wall component metabolism (Fig. 3e, Extended Data Fig. 6f–h and Supplementary Table 11). The group of cell wall-related proteins with differential gene expressions included EXPANSINS (EXPA), a family of plant cell wall regulatory proteins that facilitate turgor-driven cell enlargement[16]. Notably, bulk RNA-seq of Xkitaake also showed similar induction of EXPA in compacted soil conditions (Supplementary Table 12 and Extended Data Fig. 7c). The upregulation of EXPA gene expression in both the exodermis and cortex (Extended Data Fig. 7c and Supplementary Table 10), is consistent with the observed enlarged cell area for both cell types under compacted soil conditions as roots undergo radial expansion (Extended Data Fig. 7e–h and Supplementary Table 13), necessitating cell wall remodelling of outer root tissues[17]. A deeper analysis of the DEGs pinpointed several genes encoding xylanase inhibitors, indicative of secondary cell wall formation, given xylan's significant role as a secondary cell wall component[18]. In addition, xylanase inhibitors are important defence components, primarily found in the cell walls of monocots where they inhibit the hemicellulose-degrading activity of microbial xylanases. This indicates that root defence responses are also activated by soil compaction. In addition, we observed upregulation of other genes involved in cell wall metabolism in the exodermis with soil compaction, including *XTH22* (LOC_Os02g57770), *OsARF6* (LOC_Os02g06910)[19] and *OsBRI1* (LOC_Os07g40630)[20] (Fig. 3f, top). Similar upregulation of cell wall-relevant genes was detected in the endodermis (Extended Data Figs. 6c and 7a–d), indicating the induction of cell wall metabolism in both exodermis and endodermis.

A group of water stress-responsive genes also showed enhanced expression in both exodermis and endodermis under compacted soil conditions (Extended Data Fig. 6e). The induction of water stress relevant genes indicates root tissues experience water stress in compacted soils, leading us to investigate the expression of genes relevant to abscisic acid (ABA), which is tightly linked to water stress[21–23]. Significantly, enriched GO terms for upregulated genes in exodermis also included the class 'response to ABA' (Fig. 3e). We thus checked the spatial expression of ABA biosynthesis genes in our scRNA-seq dataset. We identified strong upregulation of *OsAAO1* and *OsNCED* genes (which encode enzymes catalysing the last steps of ABA biosynthesis) in phloem-derived vascular tissue[23] (Fig. 3f, bottom and g and Extended Data Fig. 8a). We also noted induced expression of ABA-responsive genes in other outer cell layers (beside exodermis) in response to soil compaction (Fig. 3h and Extended Data Fig. 8c). Moreover, we also found similar induction of ABA biosynthesis genes in our bulk RNA-seq in compacted soil conditions (Extended Data Fig. 8b).

Our scRNA-seq analysis shows that ABA biosynthesis occurs predominantly in inner cell layers, whereas ABA responses are activated in outer cell layers. This aligns with published findings that ABA synthesized in the root stele moves radially outwards with water flux to activate responses in outer tissues[24]. Hence, our scRNA-seq dataset demonstrates how compaction stress drives coordinated, cell-specific

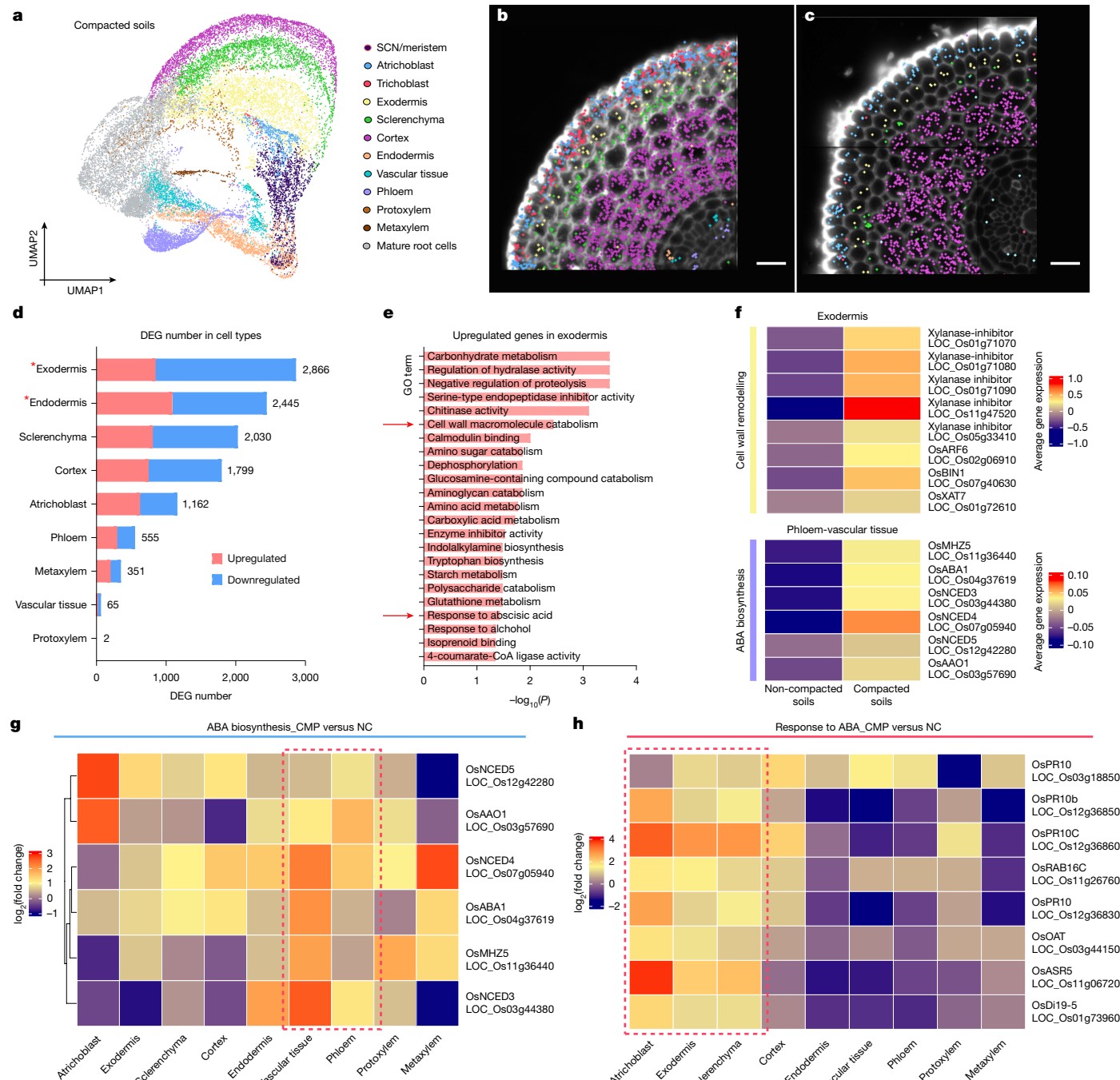

**Fig. 3 | Soil compaction stress triggers root cell type-specific expression changes including ABA and barrier formation genes in stele and exodermal tissues. a**, UMAP visualization of scRNA-seq of rice primary roots grown in compacted soils. Colours indicate cell type annotation. **b,c**, Spatial expression maps of major cell type markers in transverse root sections from non-compacted (**b**) and compacted (**c**) soils. Dots represent detected mRNA molecules, colour-coded by cell type. *n* = 4 biological replicates for compacted soil-grown roots. **d**, The number of DEGs between non-compacted and compacted soil conditions for nine annotated rice primary root cell types. The numbers next to the bars represent the total number of DEGs in the specific cell type. Exodermis and endodermis, marked by red asterisks, are the two cell types with the most DEGs, indicating that they are particularly influenced by soil compaction. **e**, Enriched GO terms for upregulated exodermis genes under compaction. Cell wall metabolism and ABA responses are highlighted (red arrows).

The one-tailed hypergeometric test with g:Profiler2 g:SCS algorithm was used for *P* value calculation. **f**, Heatmap presenting the average of normalized gene expression for the upregulated DEGs relevant to cell wall remodelling in exodermis (top), and ABA biosynthesis in phloem-related vascular tissue (bottom). Colour bars indicate the scaled expression level in these cell types. **g**, Heatmap showing the spatial expression pattern of key ABA biosynthesis genes in compacted versus non-compacted soil conditions. The vascular tissues and phloem cell files are demarcated with a rectangular border highlighting the tissue-specific induction of ABA biosynthesis genes. **h**, Heatmap showing the spatial expression pattern of key ABA-responsive genes in compacted versus non-compacted soil conditions. The outer cell layers are marked with a rectangular border highlighting the outer tissue-specific induction of ABA-responsive genes. Scale bars, 25 μm.

responses to stress signals, such as ABA, progressing from the inner to the outer root cell layers.

## ABA induced root barriers reduce water loss during compaction

Soil compaction is known to exert water stress on roots as moisture release is reduced from the smaller soil pores[25]. The coordinated regulation of suberin and lignin accumulation in roots is essential to maintain the water balance for various plant species[26,27]. Our scRNA-seq analysis revealed upregulated expression of many lignin and suberin biosynthesis genes in outer (exodermis) and inner (endodermis) root cell layers that can form apoplastic water-impermeable barriers (Extended Data Fig. 6i). We found similar induction of several lignin and suberin biosynthesis genes in our bulk RNA-seq dataset in compacted soil conditions (Extended Data Fig. 6i). To validate our expression results, histochemical staining was performed for lignin (basic fuchsin) and suberin (fluorol yellow) in mature wild-type (WT) rice root tissues exposed to non-compacted and compacted soil conditions. Our imaging of these barrier components revealed higher lignification (Fig. 4a versus 4e) and suberization (Fig. 4b versus 4f) in root exodermal, endodermal and vascular cell types exposed to compacted soil conditions. To test whether compaction stress induced barrier formation is regulated by ABA, we characterized lignin and suberin levels in roots of the rice ABA biosynthesis mutant *mhz5* grown in compacted soils[28,29]. *MHZ5* expression is significantly induced by soil compaction in the phloem-related vascular tissue in our scRNA-seq dataset (Fig. 3f). In contrast to WT, *mhz5* roots did not show induction of lignin and suberin levels in response to compacted soil conditions (Fig. 4c versus 4g and Fig. 4d versus 4h and Extended Data Fig. 8m–t). Moreover, we also quantified lignin levels in WT and *mhz5* root tips grown under both compacted and non-compacted soils, showing a significant increase of lignin in WT, whereas the *mhz5* mutant showed no substantial difference under compacted conditions (Extended Data Fig. 8d). To further confirm the role of ABA in regulating barrier formation under compacted soil conditions, we analysed lignin and suberin patterns in two additional ABA biosynthetic mutants (*aba1* and *aba2*), both of which showed minimal barrier induction in compacted soils (Extended Data Fig. 8e–l) Hence, our results show that ABA has a key role in triggering barrier formation during compaction stress conditions, similar to the radial oxygen loss barrier being induced in stagnant soil conditions[30].

What is the physiological and functional importance of lignin and suberin barrier formation? One key link between secondary cell wall formation in barriers is the enhanced cell wall stiffness that helps to protect roots from soil mechanical stress. The higher expression and accumulation of lignin and suberin in the endodermis prompted us to analyse the cell wall stiffness of endodermal cells under both non-compacted and compacted soil conditions. Our phonon imaging revealed increased stiffness in the endodermal cell layer under compacted soil conditions (Extended Data Fig. 7i–n), providing direct evidence of rice roots enhancing cell wall rigidity to deal with mechanical stress.

On the basis of the induction of key water stress-responsive genes and enhanced barrier formation under compaction, we sought to delineate the actual role of the barriers in dealing with water stress in compacted soils. To evaluate this, we performed radial water loss experiments using WT and *mhz5* root tips grown in either non-compacted or compacted soil conditions. Three-centimetre-long root tips were excised from soil-grown roots and kept in a humidity-controlled environment to quantify the weight loss, as an indirect measurement of radial water loss. WT root segments grown in non-compacted soil conditions lost half of their water content in just 17 min, whereas root tips exposed to compaction stress took almost 25 min. Hence, cumulative water loss in compacted root tips was roughly 1.5 times slower than in non-compacted root tips (Fig. 4i). This reduction in radial water loss

after exposure to compaction stress is not observed in *mhz5* mutant root tips (Fig. 4i,j and Extended Data Fig. 8u).

Our results show that ABA has a key role in triggering adaptive responses to compaction stress, which include induction of lignin and suberin barriers in the exodermis and stele cell types, which collectively act to prevent root radial water loss. The induction of ABA biosynthesis is a hallmark of physiological water stress conditions[31]. Our scRNA-seq approach provides spatial insights into the cascade of signalling events taking place in specific cell types when roots are exposed to soil compaction. In response to this soil stress phloem cells upregulate expression of biosynthesis genes for the abiotic stress signal ABA, which then targets outer root cell types such as the exodermis to form water-impermeable barriers to reduce root moisture loss (Fig. 4k).

## Discussion

Our study shows how cellular-resolution transcriptomic approaches can provide unprecedented new insights into root–soil interactions and adaptive responses. Most root stress studies performed so far have been conducted in aseptic growth systems such as gel-based media. However, plant roots are normally exposed to a heterogeneous soil environment, encompassing a range of textures, microbiomes and levels of moisture and nutrients[32]. Single-cell transcriptomics showed key transcriptional differences among cell types when grown in a natural soil system versus an axenic gel system. Transcriptional differences were predominantly confined to outer root tissues, whereas inner root cell types showed limited response. Upregulated genes in soil-grown roots included *NB-ARC*, *WRKY48* and those encoding cupin domain proteins and strip rust proteins, known to respond to bacterial, viral and fungal pathogens. Transcript levels of nucleotide-binding leucine-rich repeat genes (NLRs) are normally low in the absence of pathogens. The elevated spatial expression of NLRs indicates that outer root cell types are exposed to the soil microbes when cultured in real soil environments. Alternatively, plant roots may deliberately upregulate immune response component expression in outer root cell layers to prepare for the biotic heterogeneity in soil environments. In soil-grown roots, upregulated genes include transporters for macronutrients (nitrate and phosphate) and micronutrients (zinc, iron, magnesium, boron and potassium), as well as genes involved in defence responses, vesicle-mediated transport and cell wall remodelling. This expression pattern illustrates how plant roots sense diverse elements within natural soils and change molecular responses enhancing readiness to biotic challenges, nutrient transport to drive growth and development to explore the soil environments effectively. Thus, sensing of the external environment concomitantly with cell signalling and cellular reprogramming collectively orchestrate the growth and adaptation of plant roots in soil environments.

Root cell types growing in soils have to sense and respond to not only biotic but also abiotic stresses. Our study also explored how root cell types responded to the model abiotic soil stress, compaction. Radial expansion of outer root cell types (including exodermis, cortex and epidermis) represents a hallmark of plant adaptive growth response to soil compaction stress[32] (Extended Data Fig. 7e–g). This adaptive growth response, primarily driven by radial cortical cell expansion, will necessitate the remodelling of cortical cell walls and, as a result of this expansion, all surrounding outer root cell layers would also undergo cell wall modifications. Consistent with this, our scRNA-seq dataset revealed enrichment of cell wall remodelling gene classes, including EXPA and GRPs (glycine rich protein genes), in outer cell types (Extended Data Fig. 7c). Also, considering the increased mechanical stress applied to neighbouring cell layers due to the expansion of cortical cells[17] the responsive cell layers can either enhance the cell wall stiffness by cell wall remodelling, or expand themselves to release the stress. The accumulation of lignin and suberin at the exodermis and endodermis may also serve to enhance the mechanical stability of root tips.

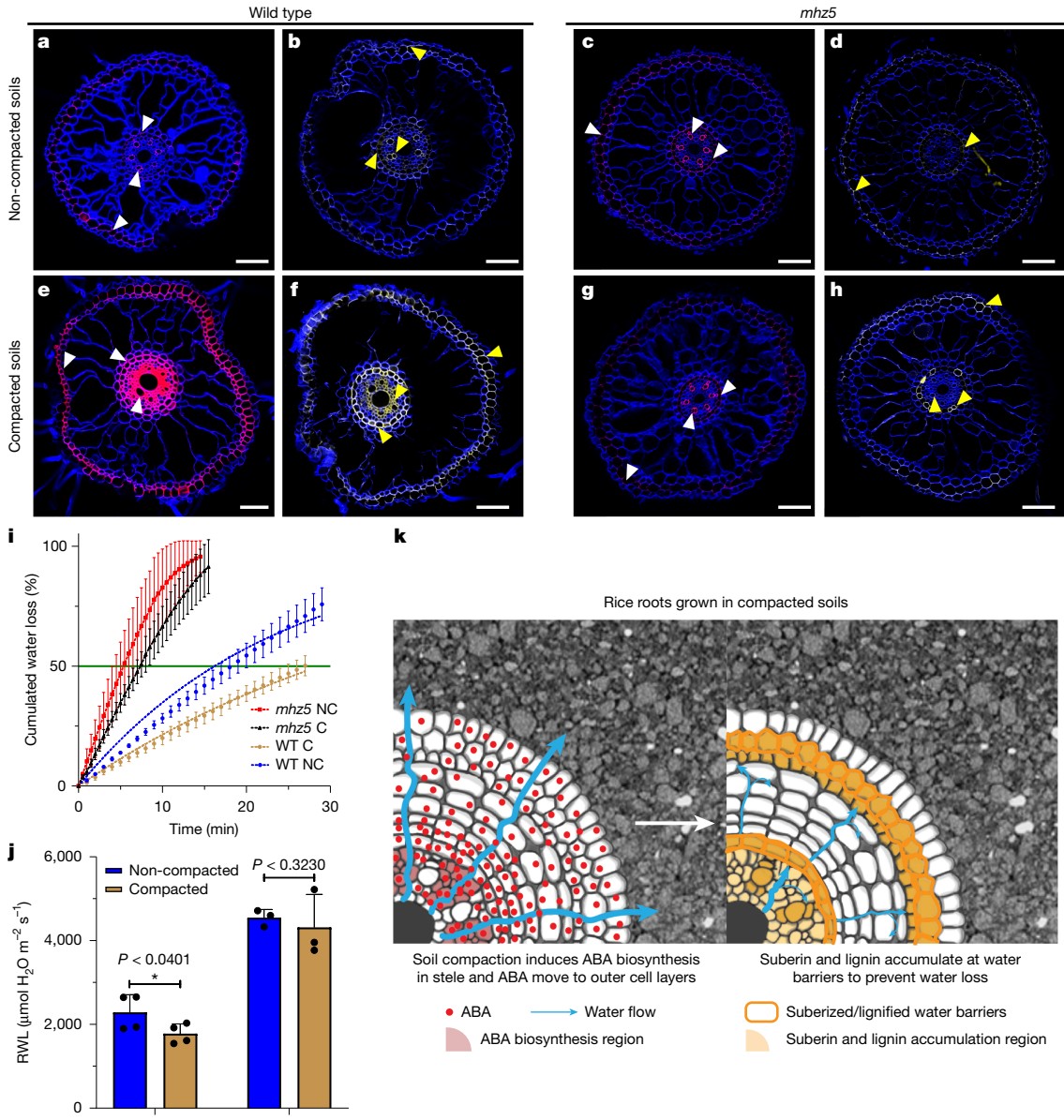

**Fig. 4 | ABA-dependent suberin and lignin deposition protects rice roots against radial water loss under soil compaction. a–h**, Histochemical staining of WT and *mhz5* root cross-sections from non-compacted (**a–d**) and compacted (**e–h**) soil conditions. **a,b,e,f**, Lignin staining (basic fuchsin, magenta, white arrowheads) of WT non-compacted (**a**) and compacted soil grown root (**e**) and suberin staining (fluorol yellow, yellow, yellow arrowheads) of WT radial root sections in non-compacted (**b**) and compacted soil (**f**) are shown. **c,d,g,h**, Similarly, lignin imaging of *mhz5* roots in non-compacted (**c**) and compacted soils (**g**) and suberin imaging of *mhz5* roots are shown in non-compacted (**d**) and compacted soil (**h**) conditions. Sections are roughly 2 cm behind the root tip. Staining experiments were repeated three times independently (*n* = 6 for non-compacted and *n* = 4 for compacted soils per experiment). **i**, Cumulative water loss in WT and *mhz5* segments (3 cm long including the root tip) under non-compacted or compacted conditions. Data are mean ± s.d. The models fitted are shown as a dashed line for both genotypes

and growth conditions (two-phase decay). The green line marks the time when 50% of water was lost. C, compacted soils, NC, non-compacted soils. *n* = 5 replicates per genotype and conditions. **j**, Radial water loss rates quantified at the time point when 50% of the water was lost from roots. Statistical comparison was done by a one-tailed *t*-test. Bars indicate mean ± s.d. *n* = 4 for WT and *n* = 3 for *mhz5*. WT (*P* < 0.0401): * denotes a significant difference with *P* < 0.05; *mhz5* (*P* < 0.3230): difference not significant. **k**, Schematics illustrating rice root cell type-specific responses to soil compaction stress. Phloem relevant vascular tissue upregulates the expression of ABA biosynthesis genes. ABA targets outer root cell types, potentially following the outward water flow. ABA reaches outer cell layers, such as the exodermis, to induce water-impermeable barriers. ABA promotes suberin and lignin accumulation, forming water-impermeable barriers that enhance structural support, reduce radial water loss and protect root systems under compaction stress. Scale bars 50 μm (**b,c,e**) and 75 μm (**a,d,f–h**). Panel **k** adapted with permission from Xiaoying Zhu.

Indeed, our phonon imaging (Brillouin microscopy) provides direct evidence that rice roots reinforce cell wall rigidity at barriers to support and protect root systems and plants under compaction stress.

Soil compaction not only imposes mechanical stress on roots, but also reduces water and nutrient absorption. The latter is due, in part, to compacted soil pores being more difficult for roots to extract water

from[28,33], creating water stress-like conditions. Consistent with this, our scRNA-seq dataset revealed upregulation of key ABA biosynthesis genes in root vascular cell types in response to compaction stress (Fig. 3g and Extended Data Fig. 8a). Elevated ABA levels target outer root layers, potentially through the outward water flow[24], triggering induced expression of ABA-responsive genes in these cell types (Fig. 3h).

ABA-dependent root adaptive responses to compaction stress included elevated lignin and suberin accumulation in water barriers and stele cell types at the root maturation zone, as opposed to the younger regions (Extended Data Fig. 8m–t). We demonstrate that ABA-dependent formation of these barriers facilitates water retention in root tips during compaction stress.

Our study provides direct evidence of cell wall remodelling through increased expression of suberin and lignin biosynthesis genes, specifically in the exodermis and endodermis, as revealed by scRNA-seq data. This enhanced accumulation of suberin and lignin was further validated using fluorescent dye staining and direct quantification of lignin in rice roots. Our scRNA-seq analysis revealed many aspects of cell wall remodelling closely linked to cell wall properties, morphology and growth. Beyond EXPA genes, we examined the expression of cellulose synthase (CESA) and xyloglucan biosynthesis genes (Extended Data Fig. 7a–d). Both groups showed enhanced expression, with CESAs slightly induced in sclerenchyma and xylem cells, whereas xyloglucan biosynthesis genes showed strong but less cell type-specific patterns. These findings indicate that distinct aspects of cell wall remodelling are regulated by different cell type-specific mechanisms.

Besides ABA, ethylene and auxin are also reported to have roles in root responses to soil compaction. Comparative heatmaps of ethylene biosynthesis and signalling-related genes under compacted versus non-compacted soil conditions (Extended Data Fig. 9c,d,f) show the induction of several prominent ERF and EIL genes. Their enhanced expression was also supported by bulk RNA-seq data (Extended Data Figs. 9 and 10). However, no cell type-specific expression patterns were observed for these genes. Similarly, auxin signalling genes show increased expression under soil compaction (Extended Data Fig. 9a,b,e), but without cell type-specific induction. These findings indicate that ABA, rather than ethylene or auxin, drives cell type-specific gene expression changes in response to soil compaction.

Is the ABA-mediated radial water loss prevention functionally connected to root growth in compacted soils? As our previous study has revealed that *mhz5*, as well as other ABA biosynthesis mutants, *aba1* and *aba2*, all have relatively longer roots than WT in compacted soils[28], we propose that increased water loss triggers enhanced root elongation. This may be a direct consequence of impaired cortical radial expansion and potentially reflects a root strategy to rapidly explore water resources.

In summary, our single-cell and spatial transcriptomics data provide insights into how root cells sense and respond to their biotic soil environment and abiotic stresses such as compaction in a cell type-specific manner. The single-cell resolution of our approaches has been instrumental in pinpointing key genes and cell types, pathways and processes, stress signals and inter-cellular signalling mechanisms that enable roots to adapt to growth in soils. Leveraging these new soil-grown root datasets will underpin efforts to develop crops more resilient to complex edaphic stresses and contribute to future-proofing plants against challenging environmental conditions.

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

## Methods

### Plant materials and growth conditions

The rice line used in this study is Xkitaake, a Kitaake line transformed with the *XA21* gene driven by the maize (*Zea mays*) ubiquitin promoter[15,34]. To ensure that the presence of *XA21* does not influence the gene-expression trends observed in our scRNA-seq analysis, we also included the non-transgenic Kitaake line for comparison. Seeds were dehulled, sterilized with 50% bleach for 30 min and rinsed five times with sterilized water. Rice seedlings were then inserted into Yoshida's media solidified with 0.15% gellan gum (Gelzan, Caisson)[35], with the embryo positioned facing upwards. Rice seeds were kept at 30 °C in the dark for 2 or 3 days until they germinated.

For gel-based growth conditions, germinated rice seedlings were then transferred to a Percival growth chamber set to 28 °C, and constant light (2,000 LUX) for 2–3 days before harvesting.

To establish the soil-based growth conditions, Wedowee sandy loam soils from Johnston County, NC, USA (15% clay, 75% sand, 15 g kg$^{-1}$ organic C, 1 g kg$^{-1}$ total N, cation exchange capacity (CEC) 6.4 meg per 100 g, base salt, 83%, P = 199 g m$^{-3}$, K = 78 g m$^{-3}$, Ca = 804 g m$^{-3}$, pH 5.8) was used. Soils were air dried, crushed and then passed through a sieve with a 2-mm mesh size. To allow packing of soils to certain bulk densities, the soils were lightly sprayed with sterilized water and mixed thoroughly. Non-compacted soil condition was packed up to 1.2 g cm$^{-3}$ (1.2 bulk density (BD)), Compacted soil was pressed to make 1.6 g cm$^{-3}$ (1.6 BD). Soils were packed in three-dimensionally printed mesocosms at bulk density of 1.2 or 1.6 BD and then saturated with sterilized water[14]. In the soils (both compacted and non-compacted) used in our experiments, excess water was drained through gravitational pull to mimic the near-field capacity conditions. Germinated rice seedlings (maximum of four seedlings per mesocosm) with equivalent length radicles (roughly 0.5 cm) were placed below the soil surface (10 mm), and then grown in a Percival growth chamber set to 28 °C and constant light (2000 LUX) for 2–3 days before harvesting.

### Bulk RNA-seq profiling of rice root sections from meristem, elongation and maturation zones

Sections (root tip to end of lateral root cap, meristem; end of lateral root cap to the start of root hair elongation, elongation; 1 mm each beyond the start of root hair elongation, maturation 1 and maturation 2) were collected into 10 µl of RNA-later (Ambion) in the lid of a 1.5-ml tube. Samples were frozen in liquid nitrogen and stored at −80 °C, and then processed by grinding with a blue homogenization pestle. RNA was isolated using the Zymo MagBead RNA Isolation kit according to the manufacturer's protocol (Zymo). RNA was used as input into the Lexogen QuantSeq 3′ FWD RNA-Seq library preparation procedure according to the manufacturer's protocol, using the unique molecular identifier (UMI) PCR add-on kit. Libraries were indexed and pooled on an Illumina NextSeq. Reads were aligned to Michigan State University Rice genome v.7 with the STAR aligner[36], deduplicated using UMI-Tools[37] and counted with HTSeq-Count.

### scRNA-seq profiling of rice root protoplasts using the 10X Genomics Chromium system

For rice seedling harvesting, gel-grown rice seedlings were directly pulled out from the growth media and root tips were cut in the enzyme solution within the optimal osmotic conditions. For soil-grown rice seedlings, the three-dimensionally printed mesocosms were opened and rinsed with gentle water flow. The seedlings were exposed and further rinsed with a gentle water flow to remove attached soil particles. Gentle brushing with a small paint brush was also carried out to remove the remaining soil particles. The root tips were then cut in the enzyme solution with the optimal osmotic conditions.

For gel-based scRNA-seq protoplasting sample, roughly 1-cm root tips were harvested from 15–40 roots, chopped with sharp razor for 1 min and then placed into a 35-mm petri dish containing a 70-µm cell strainer and 4.5 ml enzyme solution (4% [w/v] cellulase (ONOZUKA R-10, GoldBio), 1.5% Macerozyme R10 (GoldBio), 0.24% Pectolyase (Sigma P3026), 0.4 M mannitol, 20 mM MES (pH 5.7), 20 mM KCl, 10 mM CaCl$_2$, 0.1% bovine serum albumin and 0.000194% (v/v) 2-mercaptoethanol). The digestion was incubated on an 85-rpm shaker at 28 °C for 2.5 h with extra pipette mixing every 30 min. The resulting cell solution was filtered twice through 40-µm cell strainers, transferred into a Falcon Round-Bottom Polystyrene Test Tubes and then centrifuged for 5 min at 500*g* in a swinging bucket rotor. The pellet was washed with 2 ml of washing solution (0.4 M mannitol, 20 mM MES (pH 5.7), 20 mM KCl, 5 mM CaCl$_2$, 0.1% bovine serum albumin and 0.000194% (v/v) 2-mercaptoethanol), and centrifuged again at 500*g* for 3 min. The washing step was repeated for one more time and the pellet resuspended in the washing solution (normally 50–80 µl) without CaCl$_2$ at a concentration of roughly 2,000 cells per µl. Cell concentration was counted using a C-chips disposable hemocytometer (Fuchs Rosenthal, 20 µl, VWR, catalogue no. 22-600-102).

For soil-based scRNA-seq protoplasting, the procedure mirrors that of gel-based RNA-seq protoplasting, with modifications to chopping time (reduced to 45 s) and digestion time (extended from 2.5 to 3 h). These adjustments aim to enhance protoplast yield without introducing excessive debris. Despite careful washing of soil from root tips, a significant number of epidermal cells were probably removed, potentially altering the proportions of trichoblast and atrichoblast cells under different growth conditions. We conducted root trichoblast cell-specific reporter image analysis in gel, non-compacted and compacted soil conditions and we did not see a difference in the number of cells expressing the *proCSLD1:VENUS-N7* reporter[38].

For chromium-based droplet production, we loaded 16,000 (32,000) cells, with the aim to capture 10,000 (20,000 for High Throughput version) cells per sample with the 10X Genomics Chromium 3′ Gene expression v.3 (for sc_7), v.3.1 (for sc_108, sc_109, sc_115, sc_116, sc_192, sc_193, sc_194, sc_195 and sc_196) or v.3.1 High Throughput (for sc_199, sc_200, sc_201 and sc_202, sc_303, sc_304, sc_305 and sc_306) kits.

### scRNA-seq data preprocessing

Raw sequencing reads underwent demultiplexing from Illumina BCL files to generate FASTQ files for each sample using CellRanger mkfastq (v.3.1.0, 10X Genomics). Subsequently, reads were aligned to the *Oryza sativa* genome BSgenome object (BSgenome.Osativa.MSU.MSU7) along with the MSU7 gene annotation file. This alignment was carried out using the scKB script within the COPILOT preprocessing pipeline[9], which integrates kallisto[39] and bustools[40,41]. Quality filtering of cells was performed with the R package COPILOT[9]. COPILOT uses a non-arbitrary scheme to eliminate empty droplets and low-quality cells, using a 5% mitochondrial expression threshold as the criterion for searching the initial cut-off defining low-quality cells (parameter mt.threshold set to 5). A single iteration of COPILOT filtering (parameter filtering.ratio set to 1) was applied, effectively segregating high-quality cells from the background, as indicated by barcode rank plots. To address issues related to doublets and outliers, the resulting high-quality cells underwent further filtering, removing the top 1% of cells based on UMI counts. Putative doublets were identified and removed using DoubletFinder[42] with the estimated doublet rate from the 10X Genomics Chromium Single Cell 3′ Reagent Kit user guide.

### Normalization, annotation and integration of scRNA-seq datasets

Downstream analyses were conducted using Seurat v.3.1.5. Individual processing and examination of samples were performed, followed by data normalization using SCTransform[43]. As a standard step in scRNA-seq data processing, we identified protoplasting-induced genes using bulk RNA-seq (Supplementary Table 2). These genes were excluded from our analysis. Specifically, we conducted bulk RNA-seq

comparisons between intact roots and digested roots to identify general protoplasting-induced genes. Furthermore, we compared roots digested for 2.5 versus 3 h to account for digestion time effects and further minimize their impact on gene-expression trends in the gel versus soil comparison (Extended Data Fig. 10).

All detected genes, excluding those associated with mitochondria, chloroplasts or affected by protoplasting (absolute $\log_2$ fold change greater than or equal to 2), were retained for analysis (Supplementary Tables 2, 5, 8 and 10). PCA was executed by calculating 50 principal components using the RunPCA function (with approx = FALSE). Subsequently, uniform manifold approximation and projection (UMAP) nonlinear dimensionality reduction was computed by means of the RunUMAP function using all 50 principal components with default parameters.

These processing steps are detailed and documented in Jupyter notebooks (provided on GitHub at https://github.com/zhumy09/scRNA-seq-for-rice).

Data integration was carried out using Seurat v.3.1.5, following the Seurat reference-based integration pipeline[44,45]. The sample with the highest median UMI/gene per cell and the highest number of detected genes was selected as the reference (sample name, tz2; Supplementary Data 1). The 12 WT replicates (tz2, tz1, sc_108, sc_109, sc_7, sc_115, sc_116, sc_192, sc_193, sc_194, sc_195, sc_196) were used to construct the WT atlas shown in Fig. 1, including two previously published samples (tz1, tz2; Supplementary Data 1). For the integrated object containing eight samples shown in Figs. 2 and 3, comprising gel-grown (sc_192, sc_193, sc_194, sc_195) and soil-grown samples (sc_199, sc_200, sc_201, sc_202), sample sc_201 was chosen as the reference. These processing steps are detailed and documented in Jupyter notebooks (provided on GitHub at https://github.com/zhumy09/scRNA-seq-for-rice).

The cell type annotation for both integrated objects was based on markers (Supplementary Table 3) that have been previously validated and show specific local expression on the atlas UMAP. Marker gene-expression $z$-scores were calculated depending on clustering. Clusters were defined using the Seurat FindClusters function by testing the modularity parameter, ranging from res = 2 (low) to res = 300 (high), until the reasonable cluster numbers were reached. Coarse and finely-resolved clusters were annotated by comparing average marker gene $z$-scores. Cells annotated with the same cell identity by both resolutions were considered confidently annotated, forming the consensus annotation. This combination effectively annotated rare cell types while capturing major cell types given that high resolution and low noise provided by low-resolution are balanced. New reference expression profiles for each cell type were built by averaging the expression values for cells in the consensus annotation. All cells were then re-annotated using the correlation-based approach, which calculates Pearson correlation coefficients between each cell and reference expression profiles for cell types, assigning each cell the cell type with the highest correlation coefficient.

To eliminate the potential occurrence of specific cell groups being filtered out during our COPILOT-based scRNA-seq data preprocessing, possibility as a result of induced cell stresses, we also conducted an examination of the cell type identities for the low-quality cells and found no enrichment of any particular cell type (Supplementary Data 9). This confirmed that we have inclusively incorporated high-quality cells representative of all major cell types in an unbiased manner.

For developmental-stage annotation, correlation annotation compared each cell from scRNA-seq to bulk data from morphologically defined sections (Supplementary Data 2) for both the 12-sample WT atlas and the 8-sample integrated object grown in gel versus soil.

### Plotting gene-expression values on the UMAP projection
We examined the gene expression patterns by plotting the log-normalized, 'corrected' counts produced by the SCTransform function rather than the batch-corrected 'integrated' values. The UMAPs were generated with the 'featureplot' function in the Seurat package.

Jupyter notebooks illustrating the gene expression plotting process are available on GitHub at https://github.com/zhumy09/scRNA-seq-for-rice.

### Pseudotime estimation and heatmaps of gene-expression trends
Rice root epidermal cells were extracted from the integrated Seurat objects (12 gel-grown Xkitaake). Pseudotime was then inferred on the SCT assay of the extracted epidermal cells using Monocle3 (ref. 46). The learn_graph and order_cell functions in Monocle3 package were used to generate pseudotime metadata. Owing to the complexity of defining epidermal principal points, we opted to calculate pseudotime values separately for atrichoblast and trichoblast cells. Subsequently, these values were merged back into the pseudotime metadata. Furthermore, we manually delineated ten developmental groups. The construction of a UMAP representing the pseudotime trajectory and gene expression (SCT) was achieved using the 'plot_cells' and 'plot_genes_in_pseudotime' functions in the Monocle3 package. Differential expression analysis for genes was conducted using the 'graph_test' function within Monocle3. The modular expression trends of DEGs were visualized using the ComplexHeatmap package in R[47].

Jupyter notebooks illustrating the pseudotime analysis process are available on GitHub at https://github.com/zhumy09/scRNA-seq-for-rice.

### Pseudobulk differential expression analysis
Pseudobulk methods, which aggregate cell-level counts for subpopulations of interest on a per-sample basis, have been identified as top performers for cross-condition comparisons in scRNA-seq[48,49]. Hence, we used a pseudobulk approach implemented in muscat (multi-sample multi-group scRNA-seq analysis tools)[48].

Differential expression analysis was conducted for our non-compacted soil-based samples versus gel-based samples, as well as for our compacted soil-based samples versus non-compacted soil-based samples. Pseudobulk expression profiles for individual cell types in each sample were aggregated for these subpopulations by summing the raw counts (RNA assay) using the 'aggregateData' function. Subsequently, differential expression testing was performed using the edgeR method[50] incorporated in the 'pbDS' function. A gene was considered differentially expressed in a given subpopulation if the false discovery rate adjusted $P$ value was less than or equal to 0.05, absolute fold change was greater than or equal to 1.5, and detection frequency was greater than or equal to 10% in any of the included conditions. GO enrichment analysis was carried out on the DEGs using the R package gprofiler2 (ref. 51). Visualizations were generated using Seurat[45], ComplexHeatmap[47] and ggplot2 (ref. 52). The full tables containing gene-expression trends and GO term enrichment information for all detected genes and GO terms from the scRNA-seq data comparison across various growth conditions is available in Supplementary Data 10.

Jupyter notebooks illustrating the pseudobulk differential expression analysis process are available on GitHub at https://github.com/zhumy09/scRNA-seq-for-rice.

### Spatial transcriptomic sample preparation
The spatial transcriptomic sample preparation followed the protocol provided by Resolve Biosciences, with minor adjustment. Root parts of rice seedlings were isolated and fixed in a paraformaldehyde (PFA)-Triton-X solution: 4% [w/v] PFA (Sigma, catalogue no. 158127) and 0.03% Triton-X (Fisher Sci, catalogue no. AC327371000) in 1× PBS solution. The fixation was conducted within a 20-ml glass scintillation vial (Fisher Sci, catalogue no. 03-340-25N). The vial, containing rice roots, was placed on ice under a vacuum chamber. Vacuum was applied to the rice roots for 10 min, and this was repeated four times. Subsequently, the rice roots were rinsed with 1× PBS and dehydrated with an ethanol gradient (15, 30, 50, 70, 80, 90 and 100%),

each concentration for 1 h on ice. The roots were then kept in 100% ethanol overnight.

For clearing the roots, a mixture of ethanol and Histo-clear (VWR, catalogue no. 101412-878) was applied in the following concentrations: 100% ethanol, 75% ethanol + 25% Histo-clear, 50% ethanol + 50% Histo-clear, 25% ethanol + 75% Histo-clear and 2× 100% Histo-clear, each for 1 h. The Histo-clear was then aspirated, and the vial was filled halfway with a mixture of 100% Histo-clear and melted paraplast (Leica, catalogue no. 39601006). The roots were included overnight at precisely 60 °C. The top half of Histo-clear was later replaced with paraplast, following an embedding routine that involved exchanging the top half of the embedding solution twice a day for 2 or 3 days until the sample stayed at the bottom of the containers.

The embedded roots were then mounted into plastic tissue embedding moulds (VWR, catalogue no. 15160-339) with properly adjusted orientation using flamed forceps. Paraplast-embedded roots were cut into 10-μm sections. These root tissue sections were transferred to cover slips provided by Resolve Biosciences, and the cover slip was placed in a slide dryer at 42 °C overnight. To prevent detachment issues, the cover slip could be placed in a 60 °C incubator for 5–30 min before proceeding to the next step.

Tissue sections mounted were deparaffinized with Histo-clear (100% Histo-clear, 100% Histo-clear, 25% ethanol + 75% Histo-clear, 50% ethanol + 50% Histo-clear, 75% ethanol + 25% Histo-clear, 100% ethanol). This was followed by rehydration with an ethanol gradient (100, 90, 80, 70, 50, 30%). The tissue was then permeabilized with proteinase K (Invitrogen, catalogue no. 25530049) buffer: 10 μm ml$^{-1}$ Proteinase K, 100 mM Tris-HCl, 50 mM EDTA) and a 0.2% [w/v] glycine (Promega, catalogue no. H5073) solution. The tissue was also refixed with a 4% [w/v] PFA solution and acetylated with an acetylation solution: 0.1 M triethanolamine (Sigma, 90279), 0.5% [v:v] acetic anhydride (Sigma, catalogue no. 320102) and 0.4% [v:v] HCl in 1× PBS. Dehydration with an ethanol gradient (30, 50, 70, 80, 90, 100, 100%) followed.

Finally, SlowFade antifade Mountant (Invitrogen, catalogue no. S36967) was applied to the tissue, and the cover slip where the tissue sections were mounted was covered with another cover slip. A slide box was used to store the cover slips with root tissue, tightly sealed with parafilm and shipped with dry ice to Resolve Biosciences for messenger RNA (mRNA) detection and imaging, with Molecular Cartography technique.

In brief, preserved mRNA molecules were hybridized with specifically designed probes based on sequence complementarity. Each probe contained a long tail with many binding sites for various fluorescent dyes. These long tails facilitated several rounds of imaging of the same probe with different fluorescent colours, generating a unique barcode for each individual gene.

The probe–mRNA complexes were sequentially coloured, imaged and decoloured for several imaging rounds. Fluorescent signal images captured on the root tissue sections were processed to identify individual mRNA molecules. Detected mRNAs corresponding to the same gene were assigned a unified identity and false-coloured for clear visualization and presentation.

The raw data for the spatial transcriptomic data included in Figs. 1–3, Extended Data Figs. 2 and 5 can be found in Supplementary Data 4 (gel), Supplementary Data 6 (non-compacted soils) and Supplementary Data 8 (compacted soils).

## Spatial transcriptomic data analysis

The Resolve Biosciences dataset comprises both stained root images and transcript detection profiles. Staining images using Calcofluor white to visualize cell boundaries were processed using the ImageJ app provided by Resolve Biosciences. The Molecular Cartography plugin facilitated the visualization of mRNA detection. Transcript information was stored in a .txt file, which could be loaded using the Molecular Cartography plugin. Specific genes with mRNA detection were selected, and each assigned unique colours and dot diameters. The resulting mRNA detection images were saved as screenshots. Subsequently, image brightness and contrast were adjusted using the auto-setting in ImageJ, for presentation.

It is notable that the detected mRNA levels in roots grown in compacted soils were considerably lower compared to those grown in gel and non-compacted soil conditions. We suspect this may be due to reduced fixation efficiency. Roots grown in compacted soils undergo radial expansion, enhanced barrier formation and increased mucilage secretion (data not shown), all of which probably hinder formaldehyde penetration into inner cell layers. As a result, mRNA preservation efficiency is diminished, particularly for markers in the stele tissue. Despite these challenges, we successfully identified roughly 20 robust cell type-specific markers under compacted soil conditions, as detailed in Supplementary Data 8.

## Bulk RNA-seq of Xkitaake roots

Root tips (roughly 1 cm) from Xkitaake rice varieties were harvested from gel, non-compacted and compacted soil conditions, and flash-frozen in liquid nitrogen. For RNA isolation, root tips were ground to a fine powder in liquid nitrogen using a mortar and pestle, followed by the addition of 1 ml of RLT buffer to the powdered tissue. RNA was then isolated and purified using the RNeasy Mini Kit (Qiagen) according to the manufacturer's protocol. Raw reads were processed by removing adapter sequences and filtering out low-quality nucleotides (base quality lower than 5). HISAT2 was used to align reads to the *Oryza sativa* (Japonica) genome, and gene-expression levels were quantified using the fragments per kilobase of transcript sequence per million mapped reads) method. Differential gene expression (log$_2$ fold change greater than or equal to 1.0) was analysed through read count normalization, model-dependent $P$ value estimation ($P_{adj} \leq 0.05$), and false discovery rate (FDR) adjustment.

## Cell area quantification

A 4% agarose gel was prepared and poured into a square petri dish, allowing it to cool for 2 min. Rice roots were then embedded in the gel for 45 min. Subsequently, the agarose block containing the root tips was radially sectioned with a razor. Transverse sections, each with a thickness of 500 μm at 0.7 cm from the root tips, were transferred to slides. Calcofluor white staining, at a concentration of 10 mg ml$^{-1}$, was applied to the transverse sections on slides for 1 min. After aspirating the staining solution, a drop of sterilized water was added on top of the sections. The root transverse sections were imaged using Zeiss 880 Confocal microscopy (excitation wavelength 405 nm, emission wavelength 410–585 nm). For data collection with the confocal microscopy, we used Zen 2009 v.6.0.0.303.

The acquired confocal images in CZI format were converted to TIF format and opened with MorphoGraphX[53]. The images underwent the following processing steps: (1) Gaussian blur with $x$-sigma, $y$-sigma and $z$-sigma set to 1 μm. (2) Edge detect with a threshold of 100, multiplier of 2, adapt factor of 0.3 and fill value of 30,000. (3) Fill holes with the $x$ and $y$ radii both set to 10, threshold of 10,000, depth of 0 and fill value of 30,000. (4) Marching cube surface with a cube size of 5 μm and a threshold of 2,000. (5) Subdivide meshes and smooth meshes until the final vertices number was close to 700,000. (6) Project signal with minimum distance of 18, maximum distance of 22, minimum signal of 0 and maximum signal of 60,000.

The resulting mesh files, representing the sample structure, were then manually segmented to identify individual cells. The mesh number in segmented cells facilitated the final quantification of cell areas.

## Lignin and suberin imaging

Rice roots (WT (cv Nipponbare), *mhz5*, *aba1* and *aba2*) grown for 3 days under ± compaction conditions were gently removed from the 3D-printed soil columns, cleaned using deionized water and a

thin brush, and embedded in 4% melted agarose. The agarose blocks containing the roots were then positioned in a vibratome (Leica), cut into 100-µm thick primary root cross-sections (1–1.5 cm or 2–2.5 cm behind the root tip), and stored in 20% ethanol. For lignin staining, the cross-sections were incubated for 10 min in a 0.2% solution of basic fuchsin dissolved in ClearSee[54] and mixed 1:1 with aqueous calcofluor white to stain cell walls[55]. The stained cross-sections were quickly rinsed with ClearSee and then washed for 1.5 h in fresh ClearSee, replacing the solution halfway through.

For suberin staining, the primary root cross-sections were stained for 10 min in a 0.01% fluorol yellow solution dissolved in pure ethanol, prepared from a 1% fluorol yellow solution dissolved in dimethylsulfoxide. The stained cross-sections were rinsed once with deionized water and incubated for 10 min in aqueous calcofluor white. Finally, the cross-sections were washed 2–3 times in 50% ethanol for 20 min.

For confocal imaging, primary root tip cross-sections stained for lignin or suberin were mounted in a drop of ClearSee or 50% glycerol, respectively, and positioned on a Leica SP5 inverted confocal microscope. The excitation (Ex) and emission (Em) settings used were as follows: basic fuchsin, 561 nm (Ex), 600–650 nm (Em); calcofluor white, 405 nm (Ex), 425–475 nm (Em); fluorol yellow, 488 nm (Ex) and 520–550 nm (Em). For both basic fuchsin or fluorol yellow with calcofluor white, a sequential scanning was configured with the corresponding settings mentioned above.

## Lignin analysis from compacted root tips

The dried root tips (WT (cultivar Nipponbare) and *mhz5* mutant root tips) were grinded into a fine powder using a microcentrifuge tube with two metal beads (3 mm) for 1 min 30 s at 20 Hz, and then solvent extracted with sequential extractions of water (1 ml, 30 min, 98 °C), ethanol (1 ml, 30 min, 76 °C), chloroform (1 ml, 30 min, 59 °C) and acetone (1 ml, 30 min, 54 °C). The extract-free samples were dried under vacuum (overnight, 50 °C) and considered as cell wall residue.

Acetyl bromide lignin was determined as previously described[56] with modifications. In brief, 1–2 mg of cell wall residue was incubated in 200 µl of acetyl bromide solution (25% acetyl bromide in glacial acetic acid) in a 2-ml Eppendorf for 3 h at 50 °C. After cooling the samples on ice, 360 µl of 2 M NaOH, 65 µl of 0.5 M hydroxylamine hydrochloride and 375 ml of glacial acetic acid were added. After centrifuging for 5 min at 14,000*g*, 50 µl of supernatant and 150 µl of acetic acid were added to wells of a 96-well ultraviolet transparent plate (Thermo Scientific). The absorption was measured at 280 nm with a microtitre plate reader (Microplate-reader SpectraMax 250, Sopachem), SoftMax Pro v.5 was used for collecting data and applying the extinction coefficient for grasses 17.75 g l$^{-1}$ cm$^{-1}$. Two technical replicates of each biological replicate were analysed.

## Radial water loss assay

Rice seedlings (either WT or *mhz5*), grown for 3 days under ±compaction, were gently removed from the three-dimensional printed soil columns. They were then delicately brushed with deionized water to remove soil particles, and the diameter of each seminal root was measured. The primary root of each seedling (4–6 seedlings were used for each replicate) was cut into a 3-cm segment, including the root tip. After gently blotting with paper towels, each segment was positioned inside a five-digit balance closed chamber (Automatic balance, Mettler Toledo) over a thin nylon mesh. The cut ends of the segments were sealed using vacuum grease (Dow Corning) before placing them in the balance.

After 1 min of equalization inside the chamber, the fresh weight was recorded and subsequently, the weight was recorded every 30 s for up to 25–30 min. A constant relative humidity was maintained by adding bags with silica gel, which maintained the relative humidity inside the chambers at 30–35%. The silica gel was replaced after every three replicates. The temperature and relative humidity were monitored using a digital logger. Following the measurements, the root segments were wrapped and preweighed in aluminium foil and placed inside a 65 °C oven for 48 h to obtain the dry mass. The dry mass was subtracted from the initial fresh mass to obtain the total water content of each replicate. Water loss at every time point was recorded to plot the cumulative water loss (percentage of total water content). The length and diameter of the roots were used to calculate the total lateral surface, and the water loss at each time point was divided by this value to obtain the radial water loss rates (µmol m$^{-2}$ s$^{-1}$).

## Cell wall mechanical imaging in compacted soils (phonon imaging)

Phonon microscopy is an optical elastography technique that uses the phenomenon of Brillouin scattering to probe mechanical information in biological specimens with subcellular resolution. Phonon microscopy photoacoustically stimulates GHz frequency coherent acoustic phonons that, as they propagate through the specimen, periodically modulate the local refractive index that induces resonant optical scattering of a probe laser[57]. Through conservation of energy, the Brillouin scattered probe photons are frequency shifted by the phonon frequency (the so-called Brillouin frequency shift) and this can be detected either using a high-resolution spectrometer as with Brillouin microscopy[58], or interferometrically in the time domain[59].

Phonon microscopy is capable of measuring a specimen's mechanical properties through the relationship between the measured Brillouin frequency shift ($f_B$) and the sound velocity ($v$):

$$f_B = \frac{2nv}{\lambda_{probe}}$$

for normal optical incidence where $n$ is the refractive index and $\lambda_{probe}$ is the optical probing wavelength. Provided $n$ is known a priori, a measurement of the Brillouin frequency shift infers a measurement of the local sound velocity, which is determined by the elasticity of the specimen in the form of the longitudinal elastic modulus ($M = \rho v^2$).

An absolute measurement of $M$ requires knowledge about the mass density; however, refractive index and mass density of plant cells have been shown to vary substantially less than inter-specimen and inter-environmental variation in elasticity[60]. In this work, we use the relative difference in Brillouin frequency shift ($\Delta f_B$) between the cell wall and the water:ethanol filled cytoplasm as a proxy for the relative difference in cell wall elasticity in compacted and non-compacted conditions. It is worth noting that the longitudinal modulus should not be directly compared with the Young's modulus, as the two describe elasticity at very different time and frequency scales (for example, Hz to kHz deformations compared with GHz); however, it has been shown that there is an empirical relationship between the two quantities[61].

## Sample preparation and signal processing for phonon microscopy

The harvested and cleaned root tips (1.5 cm) were embedded in 4% molten agarose within a three-dimensionally printed root tip cassette. Agarose blocks containing the root tips were sectioned transversely into 50-µm slices. These root cross-sections were fixed in 20% ethanol for phonon imaging experiments. A cross-section was laid flat onto a photoacoustic transducer (200-nm thick partially transparent metal:dielectric cavity on a 170-µm sapphire cover slip), covered in roughly 50–100 µl of water:ethanol medium and then topped with a glass cover slip. Residual medium was wicked away and the cover slip sandwich was sealed shut using varnish.

Once placed into the phonon microscope, a region of interest was selected (for example, the endodermis) and a 2D raster scan was performed. A phonon time-of-flight signal was detected at each spatial pixel position, and the relative Brillouin frequency shift ($\Delta f_B$) and the acoustic attenuation ($\alpha_B$) were measured for each pixel using a fast Fourier transform and wavelet transform, respectively (Extended Data Fig. 7i,j and k,l, respectively). The spatial resolution of the technique

will be determined by the optical diffraction limit (a function of optical wavelength and numerical aperture), and in this case was roughly 300 nm. This is greater than the expected thickness of the cell wall, and so the technique is probing the average elasticity of the sample volume weighted by the optical intensity distribution.

To isolate the endodermal region of interest, the Brillouin and attenuation maps were manually segmented based on positioning, morphology and size. From these segmented datasets, $\Delta f_B$ versus $\alpha_B$ cluster maps were generated and then segmented using a two-component Gaussian mixture model. This grouped the data into two clusters that were labelled 'background' and 'cell wall'. Intervals of roughly 70% confidence were determined within these clusters and mean $\Delta f_B$ and $\alpha_B$ values were calculated. The distributions identified through the two-component Gaussian mixture model are in good agreement with the spatial positions of the cell walls and cytoplasm regions.

Using the above methodology, we report in Extended Data Fig. 7n that the relative Brillouin frequency shifts in compacted endodermal cell walls are statistically significantly greater than the equivalent cell walls grown in non-compacted conditions ($P < 0.0001$). Furthermore, the measurements extracted from the cytoplasm regions can be used as a control, and a Yuen's $t$-test indicates that the two groups are not statistically significantly different ($P > 0.05$). These data indicate that the compacted cell walls have greater elasticity than those grown in non-compacted soils.

## Reporting summary

Further information on research design is available in the Nature Portfolio Reporting Summary linked to this article.

## Data availability

All information supporting the conclusions are provided with the paper. scRNA-seq data for Xkitaake and Kitaake roots grown under gel and soil conditions is available at National Center for Biotechnology Information (NCBI) BioProject PRJNA1055099 (GSE251706). scRNA-seq from ref. 8 (PMID 33824350) is available at NCBI BioProject PRJNA706435 and PRJNA706099. Bulk RNA-seq data for developmental-stage annotation is available at NCBI BioProject PRJNA1082669 (GSE260671). Bulk RNA-seq data for protoplasting-induced genes is available at NCBI BioProject PRJNA1194134 (GSE283509). Bulk RNA-seq data for Xkitaake roots grown under compacted and non-compacted soil conditions are available at NCBI BioProject PRJNA1193632 (GSE283428). Raw data for spatial transcriptomics (Molecular Cartography) is provided in Supplementary Data 4 (gel), Supplementary Data 6 (non-compacted soils) and Supplementary Data 8 (compacted soils). Source data are provided with this paper. Gene accession number information is available in Supplementary Table 14. Supplementary tables are provided with this paper. Supplementary Data 1–10 are available on the Nature Figshare platform at https://doi.org/10.6084/m9.figshare.25146260. The processed scRNA-seq for gel-grown rice roots is now publicly accessible through a user-friendly platform hosted on Shiny (https://rice-singlecell.shinyapps.io/orvex_app/).

## Code availability

We adapted codes published in ref. 9 (https://doi.org/10.1016/j.xpro.2022.101729), ref. 45 (https://doi.org/10.1016/j.cell.2019.05.031), ref. 48 (https://doi.org/10.1038/s41467-020-19894-4) and ref. 51 (https://doi.org/10.12688/f1000research.24956.2) for our scRNA-seq analysis. The adapted codes for analysing the scRNA-seq data are available at GitHub at https://github.com/zhumy09/scRNA-seq-for-rice.

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

**Acknowledgements** We thank S. Brady, S.-Y. He, A. Roeder, K. Birnbaum, R. Shahan and J. Scharwies for critical reading of the manuscript. We thank S. Bhattacharya and J. Kläver for their valuable support on our spatial transcriptomic experiments with Resolve Biosciences. We thank Xiaoying Zhu for drawing and editing the anatomical diagrams. We thank B. Cole for sharing the codes to generate 3D scRNA-seq UMAP video. We thank A. Franzluebbers for sharing the natural soils collected in the North Carolina region. We thank J. Zhang (CAS, China) and G. Huang (SJTU, China) for sharing the *mhz5*, *aba1*, *aba2* and *proCSLD1-VENUS-N7* seeds. We thank the staff of the Duke Phytotron for plant care, and the Duke Center for Genomic and Computational Biology for Illumina sequencing services. This study was supported by the Howard Hughes Medical Institute; National Institutes of Health grant no. MIRA 1R35GM131725 (P.N.B.); NSF grant no. NSF PHY-1915445 (P.N.B.), grant no. USDA-NIFA 2021-67034-35139 (I.W.T). Biotechnology and Biological Sciences Research Council (BBSRC) Discovery Fellowship grant no. BB/V00557X/1, Royal Society Research grant no. RGS\R1\231374 and UKRI Frontiers Research (ERC StG, EP/Y036697/1) (B.K.P.); BBSRC grant nos. BB/W008874/1 and BB/S020551/1 (M.J.B.); ERC SYNERGY (grant no. 101118769—HYDROSENSING) and BB/V003534/1 (sLOLA) (M.J.B.) and EMBO-ALTF (grant no. 619-2022) (L.L.P.O.). D.M.O is indebted to the Research Foundation Flanders (FWO; grant no. 1246123N) for a postdoctoral fellowship. P.M. acknowledges BBSRC Discovery Fellowship (grant no. BB/Z514482/1) and Horizon Europe ERC Starting grant (no. 101161820-WATER-BLIND). S.L.C. acknowledges funding from the Royal Academy of Engineering Research Fellowships scheme (grant no. RF-2324-23-223) and the Nottingham Research Fellowship scheme. F.P.-C. acknowledges funding from the Royal academy of Engineering (grant no. RF\201718\17144) and the Engineering and Physical Sciences Research Council (grant no. EP/W031876/1).

**Author contributions** M.Z., M.J.B., P.N.B. and B.K.P. conceptualized the study. M.Z., C.-W.H., L.L.P.O., I.W.T., T.M.N. and B.K.P were responsible for the methodology. M.Z. performed spatial transcriptomics. M.Z., B.K.P., T.M.N. and I.W.T. generated scRNA-seq data. L.V., P.M. and B.K.P. generated bulk RNA-seq data. M.Z., C.-W.H. and B.K.P. analysed single-cell transcriptomics and bulk RNA-seq data. L.L.P.O. performed lignin, suberin imaging and relative water loss experiments. S.L.C., F.P.-C. and L.L.P.O. performed phonon imaging. D.M.O. and W.B. performed lignin measurement. M.M. performed confocal images on rice root sections and

cell area quantification. M.Z., C.-W.H. and B.K.P. vizualized the study. M.J.B., P.N.B. and B.K.P. were responsible for funding acquisition. M.J.B., P.N.B. and B.K.P. administered the project. M.J.B., P.N.B. and B.K.P. supervised the study. M.Z., M.J.B. and B.K.P. wrote the original draft. M.Z., C.-W.H., A.S., W.B., S.L.C., M.J.B. and B.K.P. reviewed and edited the manuscript.

**Competing interests** P.N.B. was the cofounder and Chair of the Scientific Advisory Board of Hi Fidelity Genetics, Inc., a company that works on crop root growth. The other authors declare no competing interests.

**Additional information**
**Correspondence and requests for materials** should be addressed to Malcolm J. Bennett, Philip N. Benfey or Bipin K. Pandey.

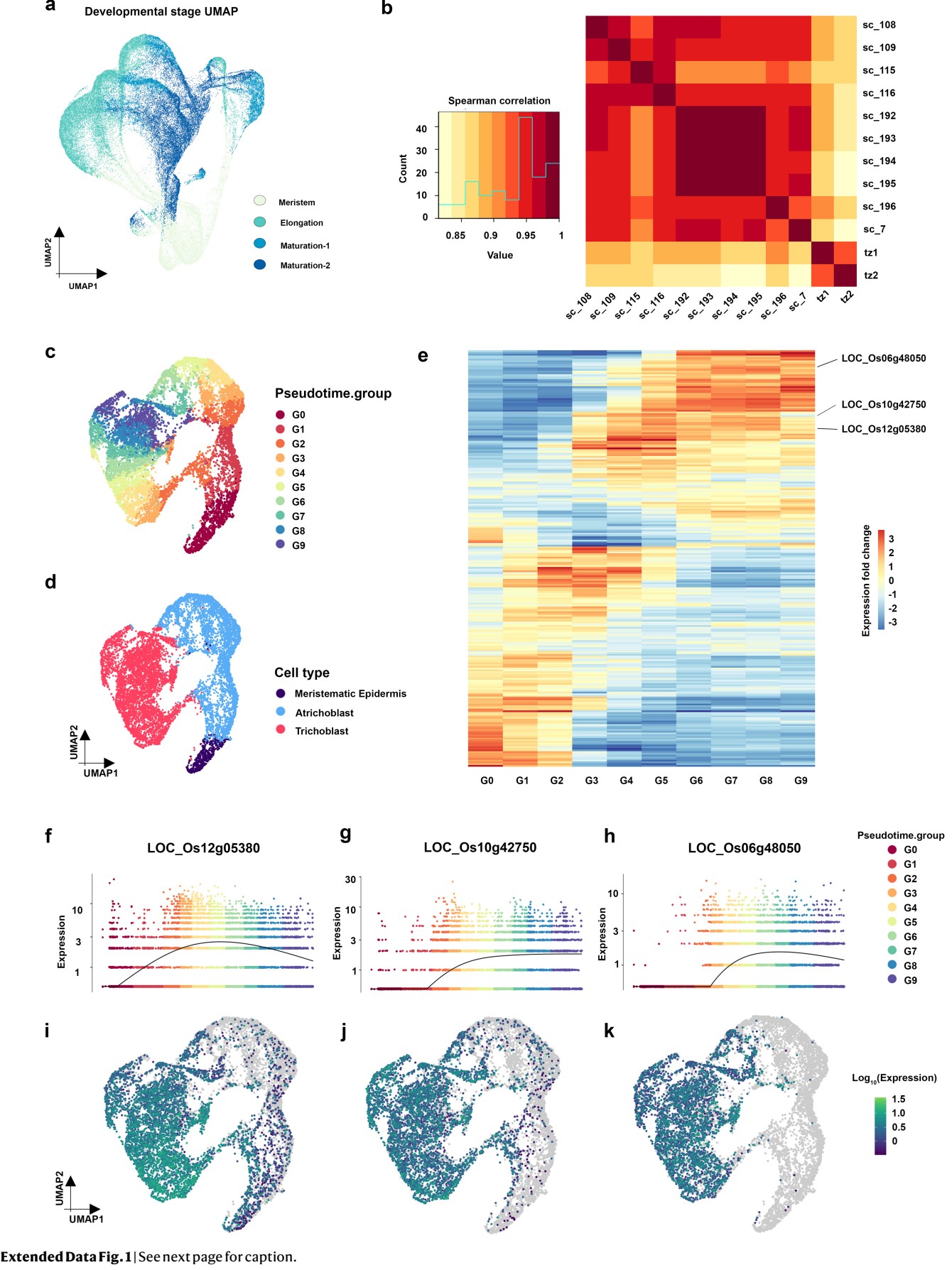

**Extended Data Fig. 1** | See next page for caption.

**Extended Data Fig. 1 | Differentiation trajectories of epidermal cells reveal expression pattern of root hair markers along root hair differentiation.** **a**, UMAP with annotations for rice root developmental stages. The cells labelled as "Maturation-1" and "Maturation-2" cannot be distinguished at this stage due to the current limitations in our knowledge. **b**, Correlation analysis between 12 scRNA-seq datasets. The dataset, tz1 and tz2 are from Zhang et al. Nature Communication 2021. The datasets, starting with "sc_" represent single-cell datasets from the current study. The relatively low correlation observed between "sc_" and "tz" samples could be attributable to differences in cultivars and growth conditions. **c**, UMAP of epidermal cell populations. Colors indicate groups of equally sized bins based on inferred pseudotime from R-Monocle3 pseudotime analysis. **d**, UMAP of epidermal cell types of rice primary root. **e**, Heatmap showing gene expression pattern during differentiation of rice root epidermal cells. Three genes with different expression enrichment timing are highlighted. **f-h**, Expression curve of selected trichoblast markers along the pseudotime trajectory. **i-k**, UMAP showing the expression pattern of three selected epidermal cell specific genes.

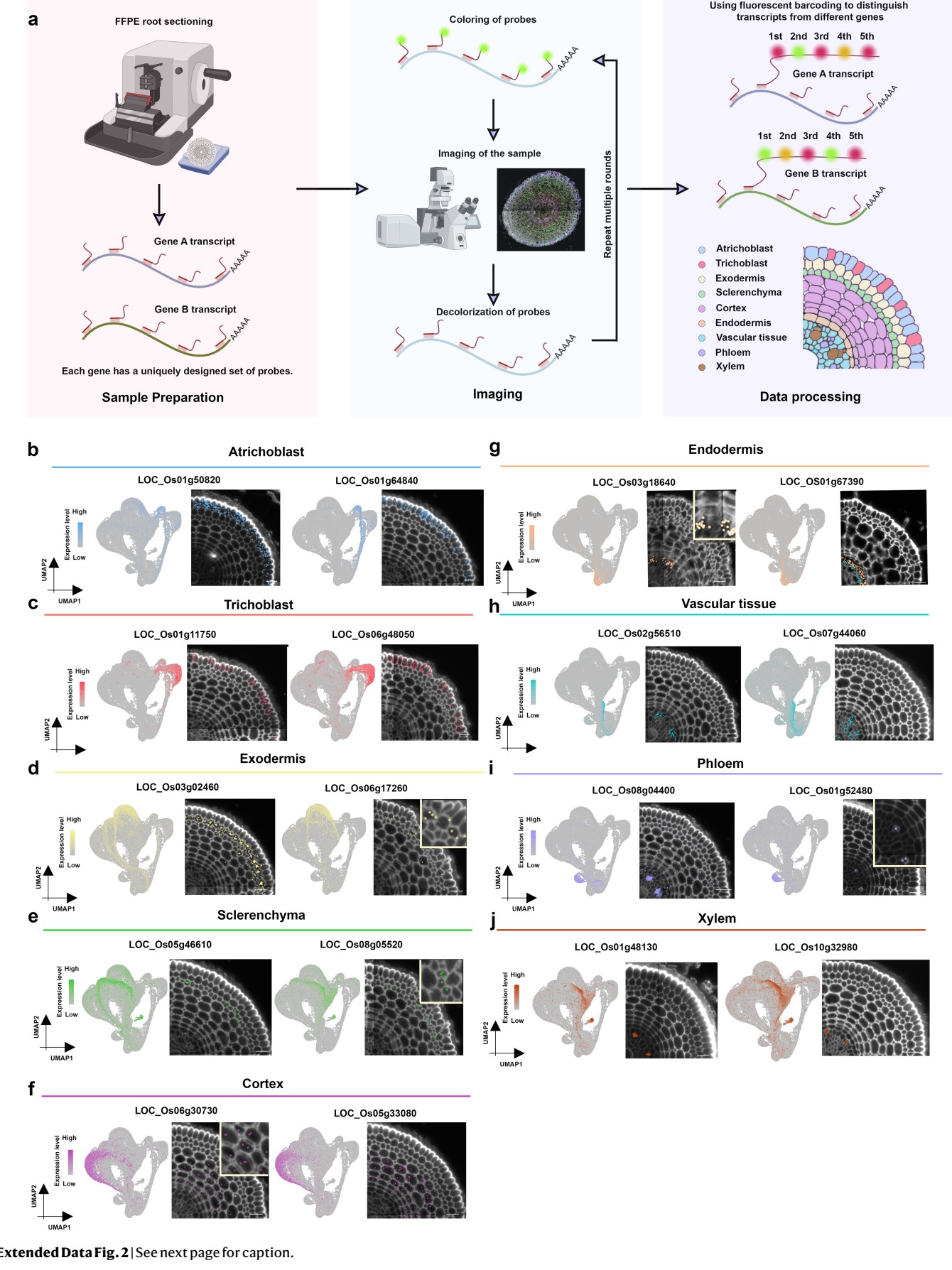

**Extended Data Fig. 2** | See next page for caption.

**Extended Data Fig. 2 | Cell type marker expressions are conserved in both scRNA-seq and spatial transcriptomics data. a**, Schematics illustrating the experimental procedures of spatial transcriptomics. Rice roots were fixed with formaldehyde and sectioned to a thickness of 10 μm. Preserved mRNA molecules were hybridized with specifically designed probes based on sequence complementarity. Each probe contained a long tail with multiple binding sites for various fluorescent dyes. These long tails facilitated multiple rounds of imaging of the same probe with different fluorescent colors, generating a unique barcode for each individual gene. The probe-mRNA complexes were sequentially colored, imaged, and de-colored for multiple imaging rounds. Fluorescent signal images captured on the root tissue sections were processed to identify individual mRNA molecules. Detected mRNAs corresponding to the same gene were assigned a unified identity and false-colored for clear visualization and presentation. **b-j**, Spatial expression pattern of identified cell type specific markers in both scRNA-seq and spatial transcriptomics data. The root transverse section anatomy illustration is displayed in the bottom right corner of panel **a**. The insets provide a magnified view (2X) of the target region to enhance visualization of the detected mRNA signals. For the images representing the expression of endodermis marker

*POEI32*, LOC_Os01g67390, arrows indicate the dislodgement of the endodermal layer. Magenta signal for vascular tissue marker expression is also shown to better indicate where the endodermis is. See also Supplementary Data. 3 and 4 for more gene expression data. $n$ = 9 biological replicates for gel-grown root transverse section spatial transcriptomic data. Scale bars: 25 μm. Marker annotations: **Atrichoblast**: LOC_Os01g50820, *OsNRT2.3*; LOC_Os01g64840, *NEP1_NEPGR Aspartic proteinase nepenthesin-1*; **Trichoblast**: LOC_Os01g11750, *OsGELP9*; LOC_Os06g48050, *Expressed protein*; **Exodermis**: LOC_Os03g02460, *Short-chain dehydrogenase TIC 32*; LOC_Os06g17260, *OsUGT*; **Sclerenchyma**: LOC_Os05g46610, *OsRLM1*; LOC_Os08g05520, *OsMYB103*; **Cortex**: LOC_Os06g30730, *OsABCG14*; LOC_Os05g33080, *Probable serine/threonine-protein kinase PBL7*; **Endodermis**: LOC_Os03g18640, *OsLAC12*; LOC_OS01g67390, *OsCOG2*; **Vascular tissue**: LOC_Os02g56510, *OsPHO1.2* LOC_Os07g44060, H*aloacid dehalogenase-like hydrolase family protein*; **Phloem**: LOC_Os08g04400, *Pentatricopeptide repeat-containing protein*; LOC_Os01g52480, *Senescence/dehydration-associated protein*; **Xylem**: LOC_Os01g48130, *OsSND2*; LOC_Os10g32980, *OsCesA7*. The full marker gene and their annotation list can be found in Supplementary Table 3. The draft of **a** was created using BioRender (https://biorender.com) and further edited with Photoshop.

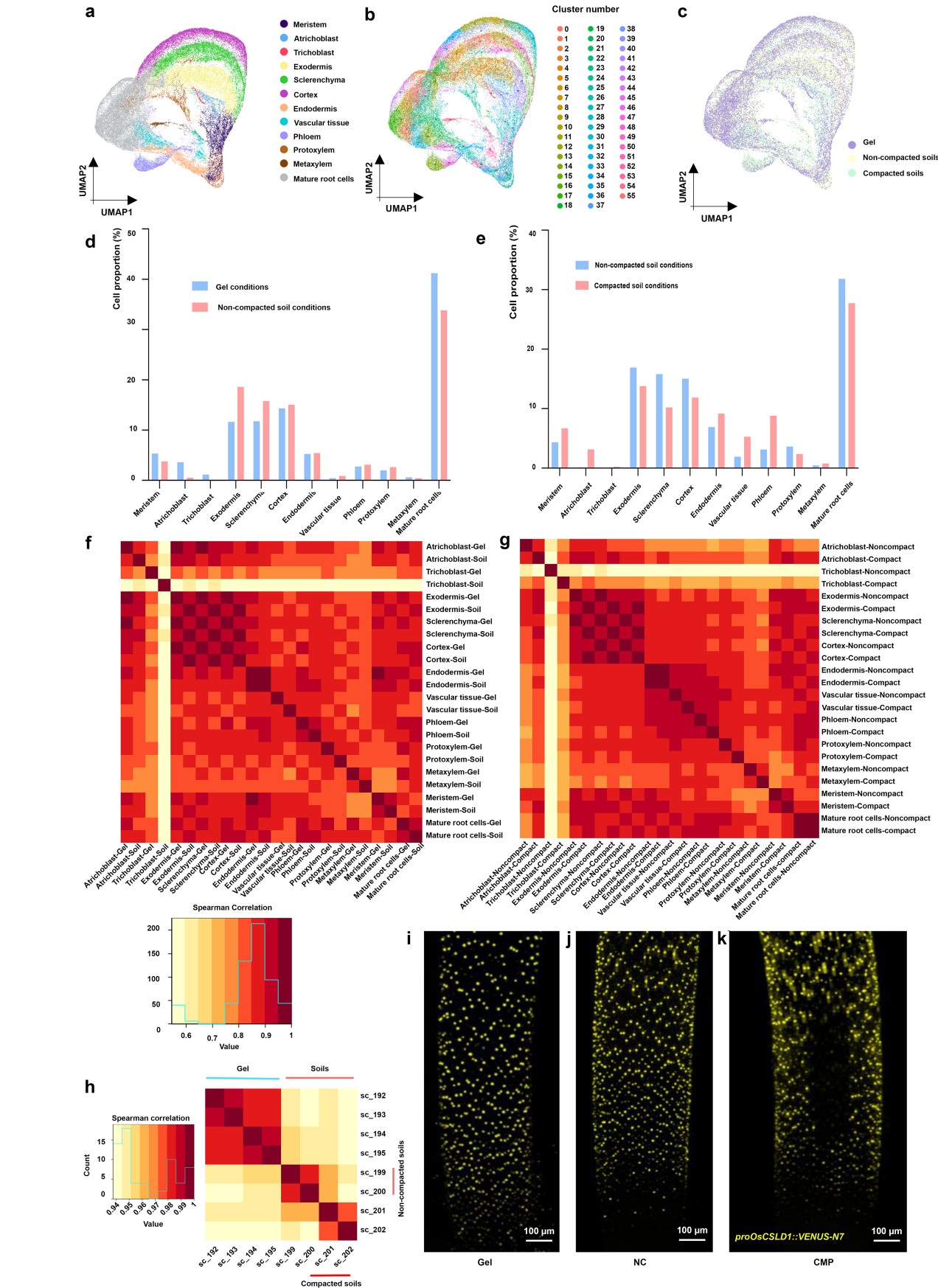

**Extended Data Fig. 3** | See next page for caption.

**Extended Data Fig. 3 | The single-cell gene expression profiles of soil grown roots are highly correlated with those of the gel-grown roots across almost all cell clusters. a**, UMAP visualization of cell distribution in the integrative scRNA-seq object, which includes gel-based, non-compacted-soil-based and compacted-soil-based scRNA-seq data. Major cell type cluster annotation is based on the expression of cell type marker genes. For cell clusters in the maturation stage, there was no clear enrichment of any cell type-specific markers. This lack of distinction may be attributed to the convergent nature of mature root cells, a phenomenon also observed in our gel-based scRNA-seq atlas (Fig. 1b). At this stage, due to the absence of markers for mature root cells, we provisionally annotated the large group of cells as "mature root cells". **b**, UMAP visualization of 55 cell clusters in the integrative scRNA-seq object, which includes gel-based, non-compacted-soil-based and compacted-soil-based scRNA-seq data. The z-scores (expression enrichment score) of major cell type markers were calculated for each cluster. We used the marker expression patterns (Supplementary Data 4, 6, 8) and the z-score maximum (Supplementary Table 4) to assign each cluster to different cell types. It is noteworthy that the number of captured epidermal cells (Atrichoblast and Trichoblast, cluster 39 and cluster 55) was significantly low under non-compacted soil conditions. To rule out the possibility that we accidentally filtered out epidermal cells as low-quality cells during scRNA-seq data processing with COPILOT, we examined the low-quality cell data. However, we did not observe any evident cell type enrichment in the low-quality cells, suggesting that epidermal cells were not erroneously filtered out as low-quality cells (Supplementary Data 9). **c**, UMAP visualization of cell distribution in the integrative scRNA-seq object. The high overlap level among almost all the cells indicates the similarity of scRNA-seq data originated from different growth conditions. **d**, Cell proportion of 10 major cell types and 2 developmental stages in both gel conditions and non-compacted soil conditions. Despite gentle cleaning of soil particles from root tips, a significant number of epidermal cells were likely removed, potentially altering the proportions of trichoblast and atrichoblast cells under different growth conditions. Growth condition itself does not change the trichoblast cell proportion dramatically. Details can be checked in Extended Data Fig. 4i-k. There is a notable increase of exodermis and sclerenchyma cell number in non-compacted-soils samples compared to that in gel conditions. **e**, Cell proportion of 10 major cell types and 2 developmental stages in both non-compacted soil and compacted soil conditions. The limited number of trichoblast cells detected under soil condition could be due to the cleaning of soil particles from root tips. **f**, Correlation analysis among the transcriptomic profiles of cells from 10 major cell types and 2 developmental stages in both gel conditions and non-compacted soil conditions. Low correlation was detected for the trichoblast cells, possibly due to the limited number of annotated root hair cells. **g**, Correlation analysis among the transcriptomic profiles of cells from 10 major cell types and 2 developmental stages (meristem and matured root cells) in both non-compacted soil conditions and compacted soil conditions. Low correlation was detected for the trichoblast cells, possibly due to the limited number of annotated root hair cells. **h**, Correlation analysis included 8 scRNA-seq datasets. The datasets sc_192 to sc_195 are gel-based scRNA-seq samples. The datasets sc_199 and sc_200 are for non-compacted soil samples while sc_201 and sc_202 are for compacted-soil samples. Although the correlation between gel-based and soil-based samples is high, they can still be distinguished from each other based on their differential expression pattern. **i-k**, Representative images (maximum projection) of *pOsCSLD1::VENUS-N7* expressing rice primary roots in gel, non-compacted (NC, 1.2 g/cm$^3$) and compacted (CMP, 1.6 g/cm$^3$) soil conditions. 3 days old rice roots were harvested from gel, and ± compacted soils. Soil grown samples were cleaned and fixed in 4% PFA (washed 5 times in PFA) and cleared for one day in ClearSee. Cleared root tips were imaged under SP8 confocal microscope. *n* = 3 biological replicates (roots), all showing similar trends. Scale bars represent 100 μm.

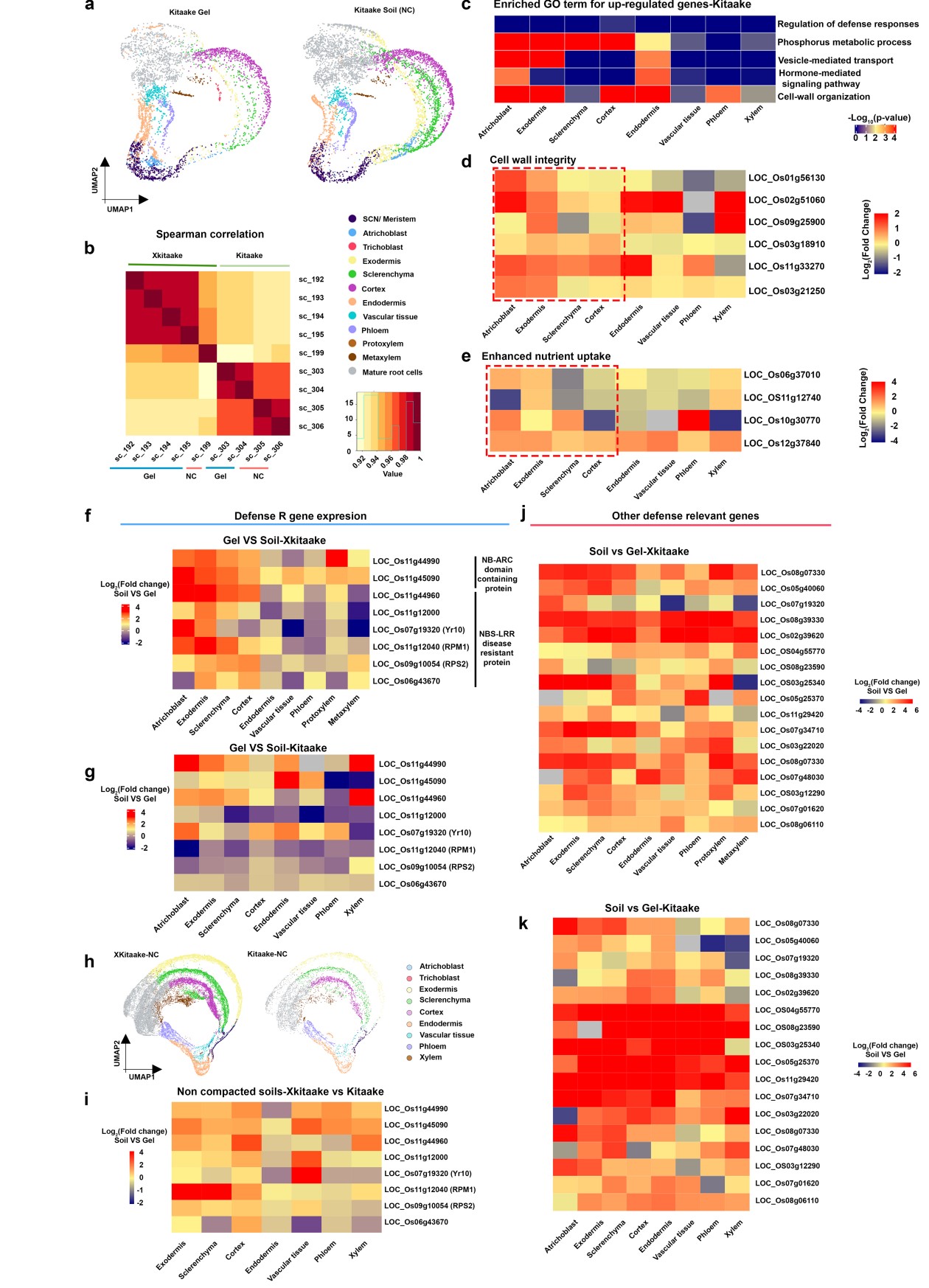

**Extended Data Fig. 4** | See next page for caption.

**Extended Data Fig. 4 | The *XA21* transgene in the Xkitaake background does not alter overall gene expression patterns when root growth conditions change from gel to soil conditions. a**, UMAP projection of scRNA-seq from roots grown in gel, and roots grown in non-compacted (NC) soils of a non-transgenic, Kitaake genotype. Colors indicate cell type annotation. **b**, Correlation analysis between 8 scRNA-seq datasets; gel-based scRNA-seq Xkitaake (sc_192 to sc_195), non-compacted-soil-based Xkitaake (sc_199), gel-based scRNA-seq Kitaake (sc_303 and sc_304) and non-compacted-soil-based Kitaake (sc_305 and sc_306) samples. High correlation values (>0.94) between Xkitaake and Kitaake scRNA-seq profiles, support their overall gene expression similarities. **c**, The GO term of "Phosphorus metabolic process", "Vesicle-mediated transport", "Hormone signalling pathway", and "Cell wall organization" are still the top enriched GO term for the up-regulated genes in Kitaake in contrasting growth conditions, sterilized gel vs natural soils. The absence of enrichment for the GO term "Defense response" in Kitaake suggests that the *XA21* transgene enhances defense responses in Xkitaake under changing growth conditions. The similarity in enriched GO terms for upregulated genes at outer root cells (highlighted with the red box) when the growth condition was changed from homogeneous gel to heterogeneous soils suggests that enhanced nutrient uptake and strengthened cell wall integrity in outer cell layers are common strategies for roots to cope with soil stresses. The one-tailed hypergeometric test with g:Profiler2 g:SCS (Set Counts and Sizes) algorithm for multiple comparison correction was used for the p-value calculation. **d,e**, Heatmap shows enhanced expression of genes involved in cell wall integrity and nutrient uptake in soil conditions (compared to gel) in Kitaake genotype. The similar induction of genes related to nutrient uptake and cell wall integrity in outer root cells (highlighted with the red box) suggests that Xkitaake and Kitaake respond similarly to the growth condition changes. This further validates that the major trends identified through scRNA-seq analysis on Xkitaake are independent of the *XA21* transgene. Grey boxes mean that the gene was not detected during the comparative analysis. Annotation for the included genes: **Cell wall integrity:** LOC_Os01g56130, *Xyloglucan glycosyltransferase 1*; LOC_Os02g51060, *Glucomannan 4-beta-mannosyltransferase 6*; LOC_Os09g25900, *Xyloglucan glycosyltransferase 2*; LOC_Os03g18910, *COBRA-like protein 7*; LOC_Os11g33270, *Xyloglucan endotransglucosylase*; LOC_Os03g21250, *Galacturonosyl transferase7*. **Nutrient uptake:** LOC_Os06g37010, *Zinc transporter 10*; LOC_OS11g12740, *NRT1*; LOC_Os10g30770, *Inorganic phosphate transporter*; LOC_Os12g37840, *Boron transporter 1*. **f**, Heatmap showing the induced expression of R (resistance) genes predominantly in outer cells in Xkitaake genotype in soil growth conditions compared to gel growth conditions. The R genes were induced when growth conditions shift to natural soils in Xkitaake, particularly in the outer cell layers, indicating the significant role of outer cell layers in the root's adaptation to soil environments. **g**, Heatmap showing the induced expression of R (resistance) genes in Kitaake genotype in soil growth conditions compared to gel growth conditions. The R genes were also induced when grown in natural soils in Kitaake, although the outer cell layer enrichment is not detected in Kitaake background. **h**, UMAP projection of scRNA-seq from Xkitaake and Kitaake roots grown in non-compacted soils. Colors indicate cell type annotation. **i**, Heatmap showing expression pattern of R genes in Xkitaake and Kitaake genotypes grown in soil conditions. The R genes show higher expression in Xkitaake roots grown under soil conditions compared to Kitaake roots grown under the same conditions. This suggests that the induction of R genes in soil conditions (compared to gel) can be further enhanced by the *XA21* transgene. However, *XA21* is not essential for this induction, as it is also observed in the Kitaake background. **j**, Heatmap showing the induced expression of the other defense response related genes in Xkitaake in soil growth conditions compared to gel growth conditions. Other defense response related genes show increased expression when compared in gel vs soil conditions in Xkitaake background. However, these genes do not exhibit a stronger induction pattern specifically in the outer cell layers. Grey boxes mean that the gene was not detected during the comparative analysis. **k**, Heatmap showing expression pattern of other defense response-related genes in Kitaake genotype in gel versus soil conditions. This analysis suggests that even in the absence of *XA21*, defense-related genes show increased expression in soil conditions compared to gel conditions. Grey boxes mean that the gene was not detected during the comparative analysis. Annotation for the included defense genes: **R gene family**: LOC_Os11g44990, *OsMGI*; LOC_Os11g45090, *OsPB3*; LOC_Os11g44960, *Yr2*; LOC_Os11g12000, *OsLRR*; LOC_Os07g19320, *Yr10*; LOC_Os11g12040, *RPM1*; LOC_Os09g10054, *RPS2*; LOC_Os06g43670, *Putative disease resistance protein RGA1*. Other defense relevant genes: LOC_Os08g07330, *Disease resistance protein RGA5*; LOC_Os05g40060, *OsWRKY48*; LOC_Os07g19320, *Disease resistance protein RGA5*; LOC_Os08g39330, *skin secretory protein xP2 precursor*; LOC_Os02g39620, *ATOZI1*; LOC_OS04g55770, *MYB/SANT-like DNA-binding domain protein*; LOC_OS08g23590, *Ankyrin repeats*; LOC_OS03g25340, *OsPRX46*; LOC_Os05g25370, *OsRLCK183*; LOC_Os11g29420, *OsLTPd12*; LOC_Os07g34710, *OsPRX104*; LOC_Os03g22020, *OsPRX40*; LOC_Os08g07330, *Disease resistance protein RGA5*; LOC_Os07g48030, *OsPOXgX9*; LOC_Os03g12290, *OsGLN1;2*; LOC_Os07g01620, *OsDIR14*; LOC_Os08g06110, *OsLHY*.

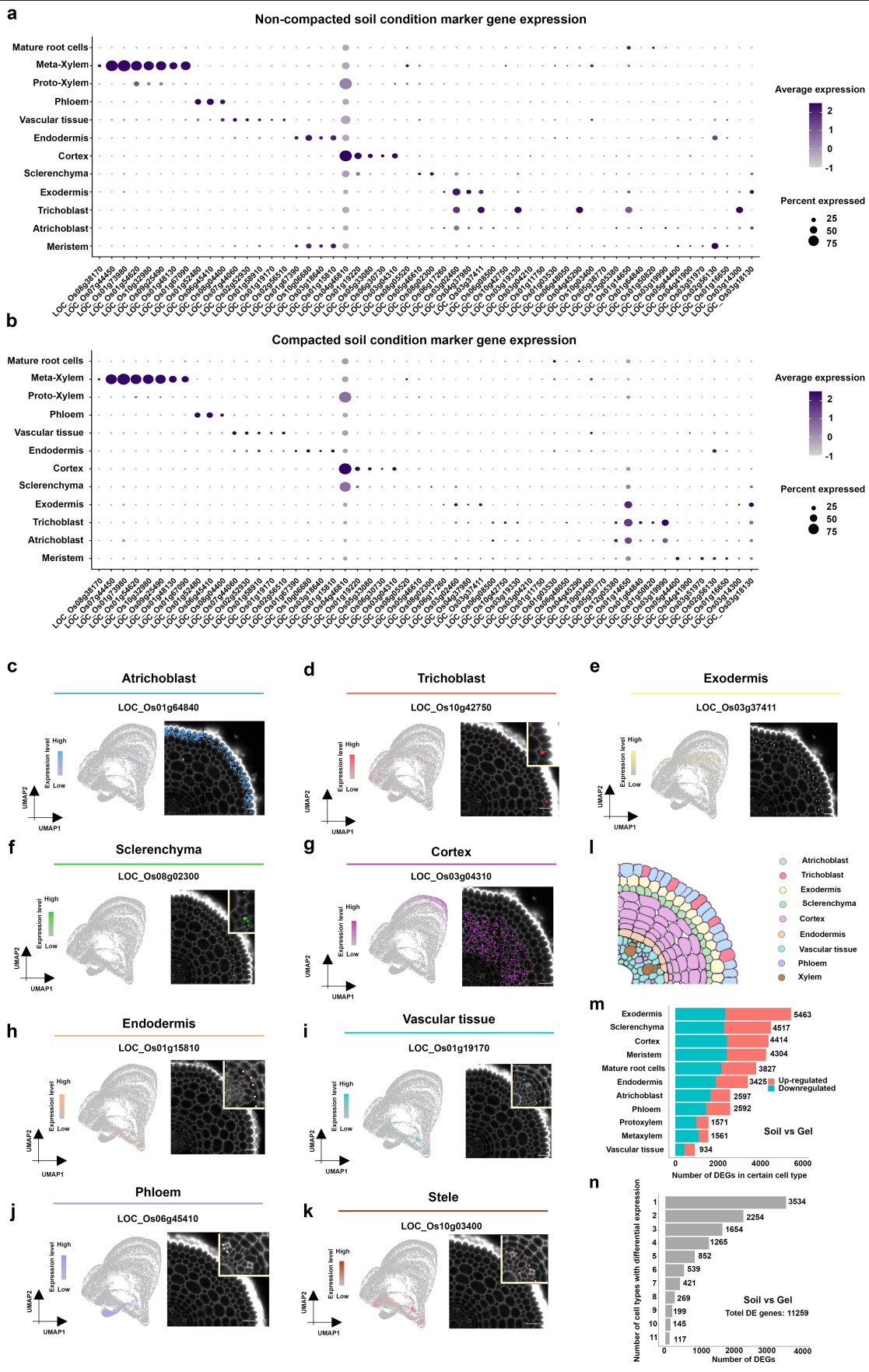

**Extended Data Fig. 5** | See next page for caption.

**Extended Data Fig. 5 | Marker gene expressions are used to annotate cell types for soil-based scRNA-seq samples. a**, Cell type expression for the identified marker genes in non-compacted-soil samples. Dot size represents the percentage of cells in which each gene is expressed (% expressed). Dot colors indicate the average scaled expression of each gene in each cell type group with darker colors indicating higher expression levels. **b**, Cell type expression for the identified marker genes in compacted-soils samples. Dot size represents the percentage of cells in which each gene is expressed (% expressed). Dot colors indicate the average scaled expression of each gene in each cell type group with darker colors indicating higher expression levels. **c-k**, Expression of identified cell type markers in both scRNA-seq and spatial data under compacted soil conditions. The color scale for each scRNA-seq feature-plot represents normalized, corrected UMI counts for the indicated gene. Spatial data of major cell type markers is visualized in rice root transverse sections. Each dot denotes a detected mRNA molecule, with different colors denoting different cell types. The insets provide a magnified view of the target region to enhance visualization of the detected mRNA signals. *n* = 4 biological replicates for compacted-soil-grown root transverse section spatial transcriptomic data. Scale bars: 40 μm. **l**, The root transverse section anatomy illustration. Marker annotations: **Atrichoblast**: LOC_Os01g64840, *NEP1_NEPGR Aspartic proteinase nepenthesin-1*; **Trichoblast**: LOC_Os10g42750, *OsCSLD1*; **Exodermis**: LOC_Os03g37411, *OsMATE12*; **Sclerenchyma**: LOC_Os08g02300, *OsSWN2*; **Cortex**: LOC_Os03g04310, *OsRAI1*; **Endodermis**: LOC_Os01g15810, *OsPRX5*; **Vascular tissue**: LOC_Os01g19170, *OsPGL13*; **Phloem**: LOC_Os06g45410, *MYB family transcription factor*; **Stele**: LOC_Os10g03400, *OsSNDP1*. The full marker gene and their annotation list can be found in Supplementary Table 3. **m**, The total number of differentially expressed genes (DEGs) for 9 major cell types and 2 developmental stages (meristem and mature root cells). Exodermis, as one of the outer cell layers, could be the most affected cell type with the growth condition change, as it has the most DEGs. **n**, The number of cell types in which one specific gene exhibits differential expression between gel-based and soil-based scRNA-seq data. Most differentially expressed genes (DEGs) are detected in only one or two major cell types.

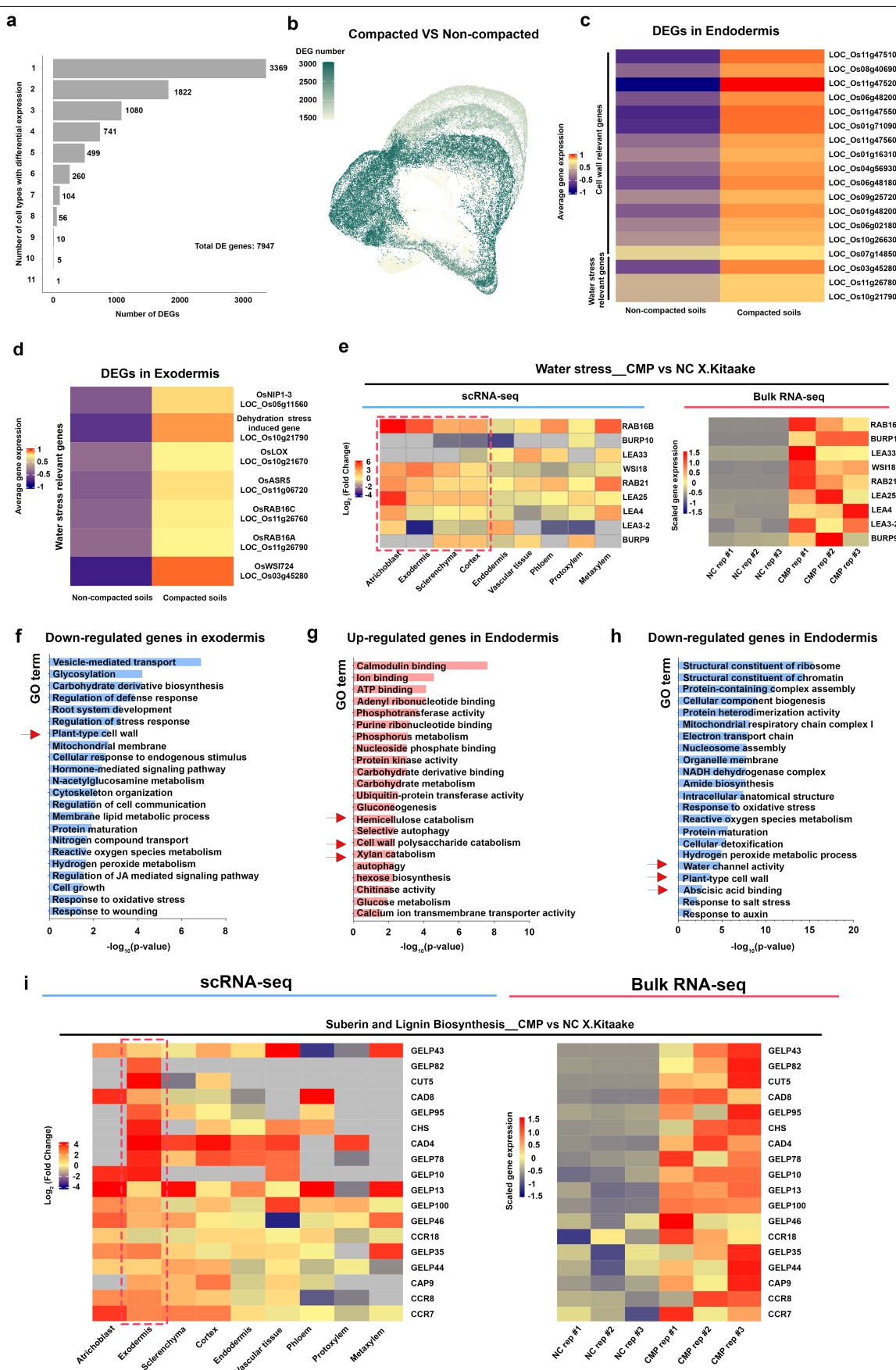

**Extended Data Fig. 6** | See next page for caption.

**Extended Data Fig. 6 | Comparative scRNA-seq for soil conditions identifies the soil compaction induced cell wall component metabolism change in exodermis and endodermis. a**, Most of the DEGs for the comparative analysis of non-compacted soils-based and compacted-soils-based scRNA-seq data are detected in only one or two major cell types. **b**, UMAP visualization of DEG number. Exodermis has the most DEGs. **c**, Gene expression heatmap for the up-regulated DEGs relevant to cell wall component metabolism and water stress response in endodermis. Color bars indicate the scaled expression level in the endodermis. **d**, Gene expression heatmap for the up-regulated DEGs relevant to water stress response in exodermis. Color bars indicate the scaled expression level in the exodermis. LOC_Os05g11560, *OsNIP1-3*; LOC_Os10g21790, *Dehydration stress induced gene*; LOC_Os10g21670, *OsLOX*; LOC_Os11g06720, *OsASR5*; LOC_Os11g26760, *OsRAB16C*, LOC_Os11g26790, *OsRAB16A*; LOC_Os03g45280, *OsWSI724*. The complete list of gene ID and annotations are included in Supplementary Table 14. **e**, Left panel: Heatmap showing differential expression ($log_2$ fold change) of water stress responsive genes in compacted soil conditions compared to non-compacted soils in Xkitaake as revealed by scRNA-seq analysis. scRNA-seq showed increased expression patterns for genes relevant to response to water stress, with stronger induction at outer cell layers (highlighted by the red box), suggesting the enhanced water stress response at outer cell layer under soil compaction. Right panel: Heatmap showing scaled expression of water stress responsive genes in non-compacted and compacted soil conditions in Xkitaake, as revealed by bulk RNA-seq. Bulk RNA seq further supported the upregulation of genes relevant to response to water stress. Bulk RNA-seq analysis was carried out using three independent biological replicates for non-compacted (NC rep #1-3) and compacted (CMP rep #1-3) soil conditions. Grey boxes mean that the gene was not detected during the comparative analysis. **f**, GO terms for the down-regulated genes in exodermis under compacted soils as compared to non-compacted soils. **g**, GO terms for the up-regulated genes in endodermis under compacted soils as compared to non-compacted soils. **h**, GO terms for the down-regulated genes in endodermis under compacted soils as compared to non-compacted soils. Cell wall remodelling and water stress relevant GO terms are highlighted with red arrows in panels f-h. The one-tailed hypergeometric test with g:Profiler2 g:SCS (Set Counts and Sizes) algorithm for multiple comparison correction was used for the p-value calculation in f-h. **i**, Left panel: Heatmap showing differential expression ($log_2$ fold change) of suberin/lignin biosynthesis genes in compacted soil conditions compared to non-compacted soils in Xkitaake as revealed by scRNA-seq analysis. Enhanced expression of suberin and lignin biosynthesis genes in exodermis (highlighted by the red box), suggest higher suberin and lignin accumulation in exodermis under soil compaction. Right panel: Heatmap showing scaled expression of suberin/lignin biosynthesis genes in non-compacted and compacted soil conditions in Xkitaake, as revealed by bulk RNA-seq. Bulk RNA-seq analysis further confirmed the upregulation of suberin/lignin biosynthesis genes in compacted soil conditions.

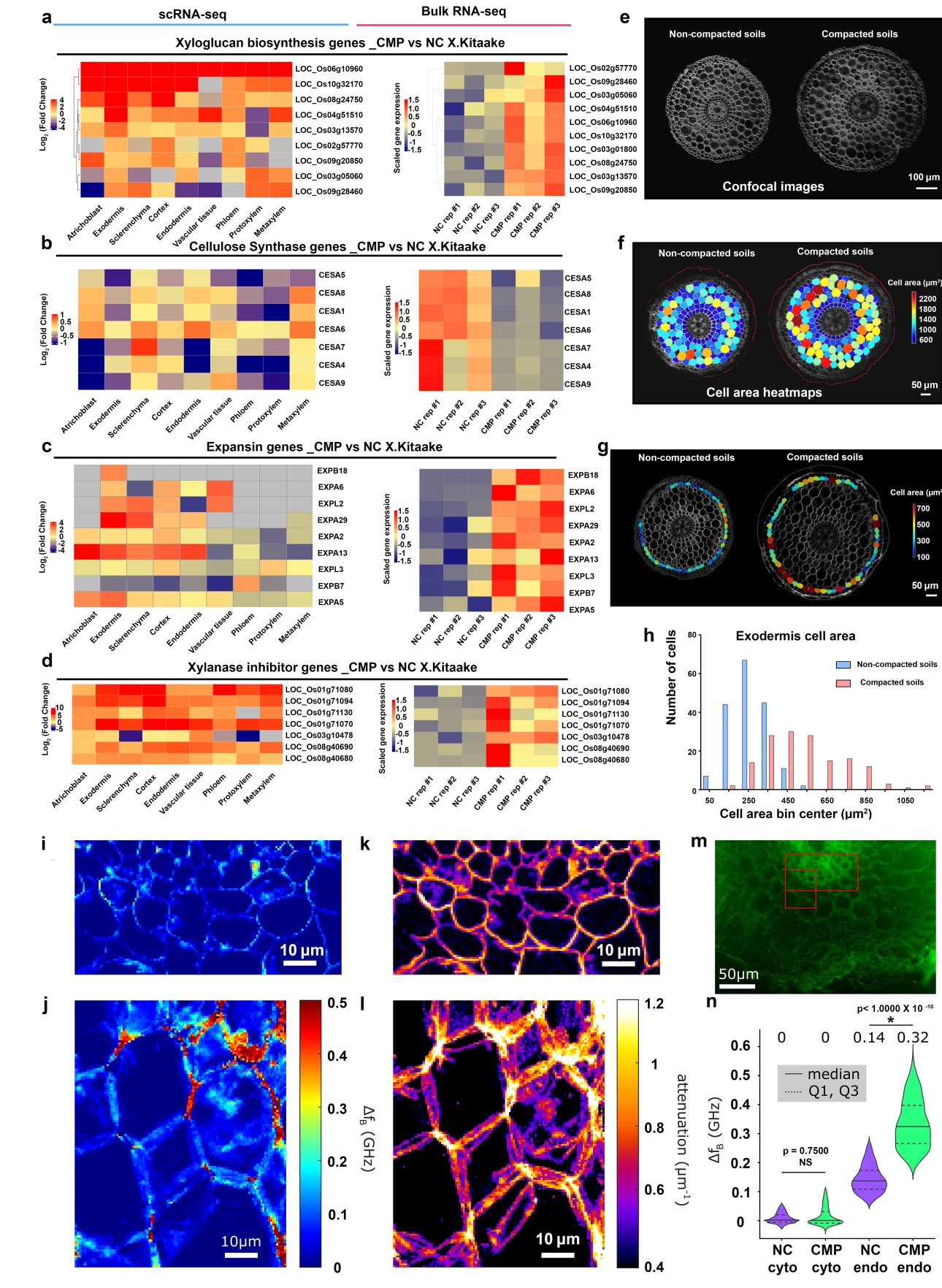

**Extended Data Fig. 7** | See next page for caption.

**Extended Data Fig. 7 | Comparative expression analysis of cell wall remodelling genes in compacted soil conditions with scRNA-seq and bulk RNA-seq approaches. a**, Left panel: Heatmap showing increased expression (log$_2$ fold change) of xyloglucan biosynthesis genes in compacted soil conditions compared to non-compacted soil conditions in Xkitaake as detected by scRNA-seq. Xyloglucan biosynthesis genes are broadly upregulated across major cell types, with a slightly higher induction observed in the exodermis. The stronger induction of xyloglucan biosynthesis genes in the exodermis aligns with the observed barrier reinforcement. NC: Non-compacted soils. CMP: Compacted soils. Right panel: Heatmap showing scaled expression of xyloglucan biosynthesis genes in non-compacted and compacted soil conditions in Xkitaake, as revealed by bulk RNA-seq. Bulk RNA-seq also reveals a general induction of xyloglucan biosynthesis genes in compacted soils. The upregulation of these genes suggests enhanced cell wall reinforcement in response to compacted soil conditions. **b**, Left panel: Heatmap showing differential expression (log$_2$ fold change) of cellulose synthase (*CESA*) genes in compacted soil conditions compared to non-compacted soil conditions in Xkitaake as detected by scRNA-seq. Notably, *CESA4*, *CESA7*, and *CESA8* exhibit increased gene expression in sclerenchyma, suggesting enhanced secondary cell wall formation in this tissue under soil compaction. Right panel: Heatmap showing scaled expression of *CESA* genes in non-compacted and compacted soil conditions in Xkitaake, as revealed by bulk RNA-seq. Interestingly, bulk RNA seq reveals a general down-regulation of *CESA* genes in compacted soils, although the decrease is subtle for most examined genes. The relatively stronger down-regulation of *CESA1*, *CESA5*, and *CESA6*, combined with the relatively weaker down-regulation of *CESA4*, *CESA7*, and *CESA8*, may suggest a transition toward secondary cell wall deposition. **c**, Left panel: Heatmap showing increased expression (log$_2$ fold change) of expansin genes in compacted soil conditions compared to non-compacted soil conditions in Xkitaake as detected by scRNA-seq. Increased expression of expansin genes, particularly in the exodermis and cortex cell layers, suggests the enhanced cell expansion at exodermis and cortex under soil compaction. Right panel: Heatmap showing scaled expression of expansin genes in non-compacted and compacted soil conditions in Xkitaake, as revealed by bulk RNA-seq. Bulk RNA-seq further supports the upregulation of expansin genes in compacted soil conditions. The up-regulation of expansin genes correlates with the radial expansion of rice roots in response to soil compaction. **d**, Left panel: Heatmap showing increased expression (log$_2$ fold change) of xylanase inhibitor genes in compacted soil conditions compared to non-compacted soil conditions in Xkitaake as detected by scRNA-seq. Xylanase inhibitor genes are broadly upregulated across major cell types. As xylanase inhibitor is tightly relevant to defense response, it further suggests that soil compaction could induce defense response in rice root. Right panel: Heatmap showing scaled expression of xylanase inhibitor encoding genes in non-compacted and compacted soil conditions in Xkitaake, as revealed by bulk RNA-seq. Bulk RNA-seq analysis further supports the upregulation of xylanase inhibitor genes in compacted soil conditions. Grey boxes mean that the gene was not detected during the comparative analysis. Annotation for the included genes: LOC-Os02g57770, *OsXTH22*; LOC-Os03g01800, *OsXTH19*; LOC-Os08g24750, *Xyloglucan fucosyltransferase8*; LOC-Os06g10960, *Xyloglucan fucosyltransferase2*; LOC-Os03g13570, *OsXTH28*; LOC-Os09g28460, *Xyloglucan fucosyltransferase7*; LOC-Os10g32170, *Xyloglucan galactosyltransferase KATAMARI1 homolog*; LOC-Os09g20850, *OsTBL41, Xyloglucan O-acetyltransferase 2*; LOC-Os03g05060, *Xyloglucan galactosyltransferase KATAMARI1 homolog*; LOC-Os04g51510, *OsXTH7*. LOC_Os01g71080, *xylanase inhibitor*; LOC_Os01g71094, *xylanase inhibitor*; *LOC_Os01g71130, xylanase inhibitor*; *LOC_Os01g71070, xylanase inhibitor*; LOC_Os03g10478, *endo-1,4-beta-xylanase 5-like*; LOC_Os08g40690, *xylanase inhibitor* LOC_Os08g40680, *xylanase inhibitor*. The complete list of gene ID and annotations are included in Supplementary Table 14. **e**, Confocal imaging of root transverse sections from non-compacted and compacted soil conditions. The cell boundary was visualized by the auto-fluorescence activated by a 405 nm wavelength laser. Scale bars: 100 μm. *n* = 3 biological replicates (roots), all showing similar trends. **f**, The heatmap of cortical cell areas under both non-compacted and compacted soil conditions. Red and blue colors indicate bigger and smaller cells, respectively. *n* = 3 biological replicates (roots), all showing similar trends. Scale bar: 50 μm. **g**, The quantification of exodermal cell areas in the root transverse sections. For the heatmap, red and blue indicate bigger and smaller cells, respectively. Segmented cells are outlined in cyan and superimposed on the meshed surface where the cell wall signals are projected (greyscale). *n* = 3 biological replicates (roots), all showing similar trends. Scale bar: 50 μm. **h**, The histogram showing the cell area distribution of exodermal cell under both non-compacted and compacted soil conditions. 3 biological replicates (roots) are included. **i,j**, Non-compacted and compacted (respectively) endodermal region maps of the Brillouin frequency shift (relative to the shift in the cytoplasm ΔfB = 0) demonstrating apparent greater cell-wall stiffness in the compacted case. The primary roots were harvested from compacted and non-compacted soils were radially sectioned and imaged using Brillouin microscopy. **k,l**, Similarly, maps of acoustic attenuation between the two cases demonstrate greater apparent longitudinal viscosity in the compacted soil conditions. Scale bars in i and k: 10 μm; Scale bar in j and l: 10 μm. **m**, Brightfield image of a rice root radial cross-section (red boxes are the relevant regions of interest in i-l). Scale bar: 50 μm. **n**, Violin plot showing the cell wall stiffness of rice primary root in compacted and non-compacted soil conditions. The width of each violin represents the kernel density estimation of the data distribution. The solid line within each violin denotes the median, while the dashed lines represent the first quartile (Q1) and third quartile (Q3). Q1 corresponds to the 25th percentile of the data, and Q3 corresponds to the 75th percentile. Yuen's t-tests (two-tailed) indicate that there is no statistically significant (NS, p value = 0.7500) relative shift in Brillouin frequency between the cytoplasm (control) regions in non-compacted (NC) and compacted (C) specimens. However, there is a clear frequency shift between endodermal cell-walls between the two cases (p value < $1.0000 \times 10^{-10}$; *: p < 0.05) indicating greater elasticity for compacted cell-walls. Four cross-sections for each case of compacted and non-compacted were imaged containing 35 and 44 cells respectively. For cell wall measurements, *n* = 1744 (non-compaction, endodermis), 1843 (compaction, endodermis), 11852 (non-compaction, cytoplasm), and 17592 (compaction, cytoplasm) units, were analyzed respectively.

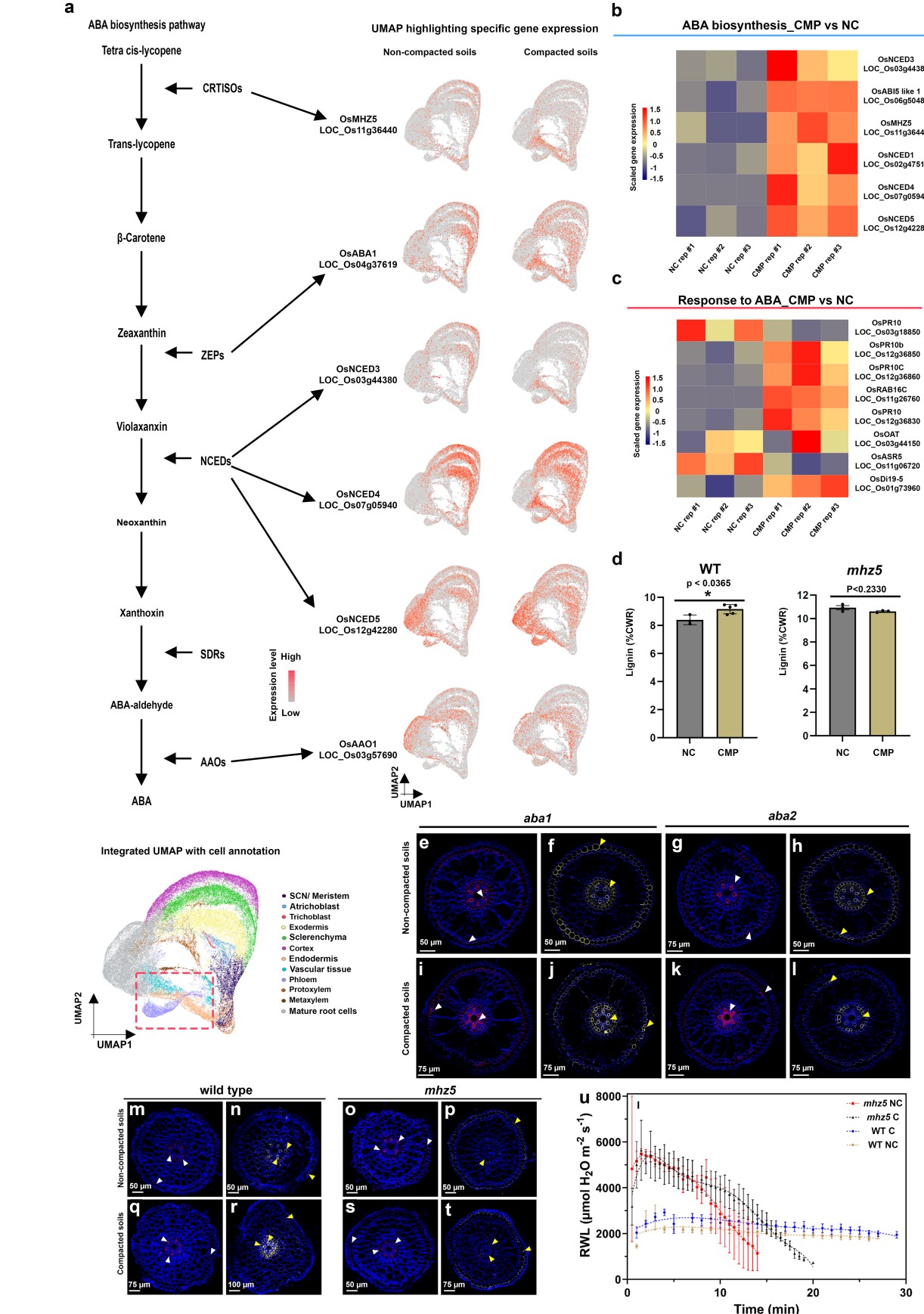

**Extended Data Fig. 8** | See next page for caption.

**Extended Data Fig. 8 | Key ABA biosynthesis genes are specifically induced in vascular cells. a**, Feature plots of key ABA pathway genes showing higher expression in phloem companion cells and pericycle cells in compacted soil conditions (right panel). The below in left panel displays a UMAP with cell type annotations for integrated scRNA-seq data, incorporating data from both non-compacted and compacted soil conditions. Cells representing phloem-derived vascular tissue are highlighted with a red rectangle. Left panel schematic shows the key step of ABA biosynthesis pathway and genes involved in these steps. **b**, Heatmap showing scaled expression of ABA biosynthesis genes in non-compacted and compacted soil conditions in Xkitaake, as revealed by bulk RNA-seq. The upregulation of genes involved in multiple ABA biosynthesis pathways suggests an increased ABA level in rice roots under soil compaction. NC: Non-compacted soils. CMP: Compacted soils. **c**, Heatmap showing scaled expression of ABA responsive genes in non-compacted and compacted soil conditions in Xkitaake, as revealed by bulk RNA-seq. Bulk RNA sequencing further supported the upregulation of most genes relevant to response to ABA, detected in scRNA-seq dataset. **d**, Lignin measurements from WT and *mhz5* root tips from non-compacted (NC, 1.2 BD) and compacted (CMP, 1.6 BD) soil conditions. 3 independent replicates for compacted soils ($n = 3$) and 5 independent replicates for non-compacted soils ($n = 5$) were used to measure the lignin amount in rice root tips grown in non-compacted and compacted soils. Each replicate contains 4 root tips for compacted soil and 6 root tips for non-compacted soil conditions to generate equal dry weight. The two-tailed *t*-tests were used to calculate the p-value. WT (p-value < 0.0365): *, significant difference with p-value < 0.05; mhz5 (p-value < 0.2330): no significant difference. **e-l**, ABA biosynthesis defects mitigate the accumulation of suberin and lignin at water barriers under soil compaction. Histochemical staining of two other ABA biosynthesis mutants, *aba1* and *aba2* rice mutant root cross sections grown in non-compacted or compacted soils (1.2 or 1.6 g/cm$^{-3}$ bulk density, panels e-h or i-l, respectively) for 3 days after germination. Lignin staining with Basic Fuchsin is shown with magenta color (panels e, g, i, k, white arrowheads) and suberin staining with Fluorol Yellow is shown as yellow (panels f, h, j, l, yellow arrowheads). The cross sections correspond to position -2 cm behind the root tip. The scale bar (50/75 μm) is indicated on each panel. Histochemical staining experiments were repeated 3 times with an n of 4 (compacted roots) 6 (noncompacted roots) each time. **m-t**, Soil compaction enhances suberin and lignin depositions closer to root tips. Histochemical staining of wildtype, or *mhz5* rice mutant root cross sections grown in non-compacted or compacted soils (1.2 or 1.6 g cm$^{-3}$ bulk density, panels m-p or q-t, respectively) for 3 days after germination. Lignin staining is shown with magenta colour (panels m, o, q, s, white arrowheads) and suberin as yellow (panels n, p, r, t, yellow arrowheads). The cross sections correspond to position -1 cm behind the root tip. The scale bar is indicated in each panel. Histochemical staining experiments were repeated 3 times with 4 (compacted) and 6 (noncompacted) roots each time. **u**, Radial water loss rates of WT of *mhz5* mutants from ± compactions of the same roots used for Fig. 4i and j. Data are mean ± *SD*. The models fitted are shown as a dashed line for both genotypes and growth conditions (4$^{th}$ order polynomial for WT and 6$^{th}$ order polynomial for *mhz5*). 4 (compaction) and 6 (non-compacted) root tips were used for each replicate and the experiment was repeated 3 times independently for both WT and *mhz5* ($n = 3$).

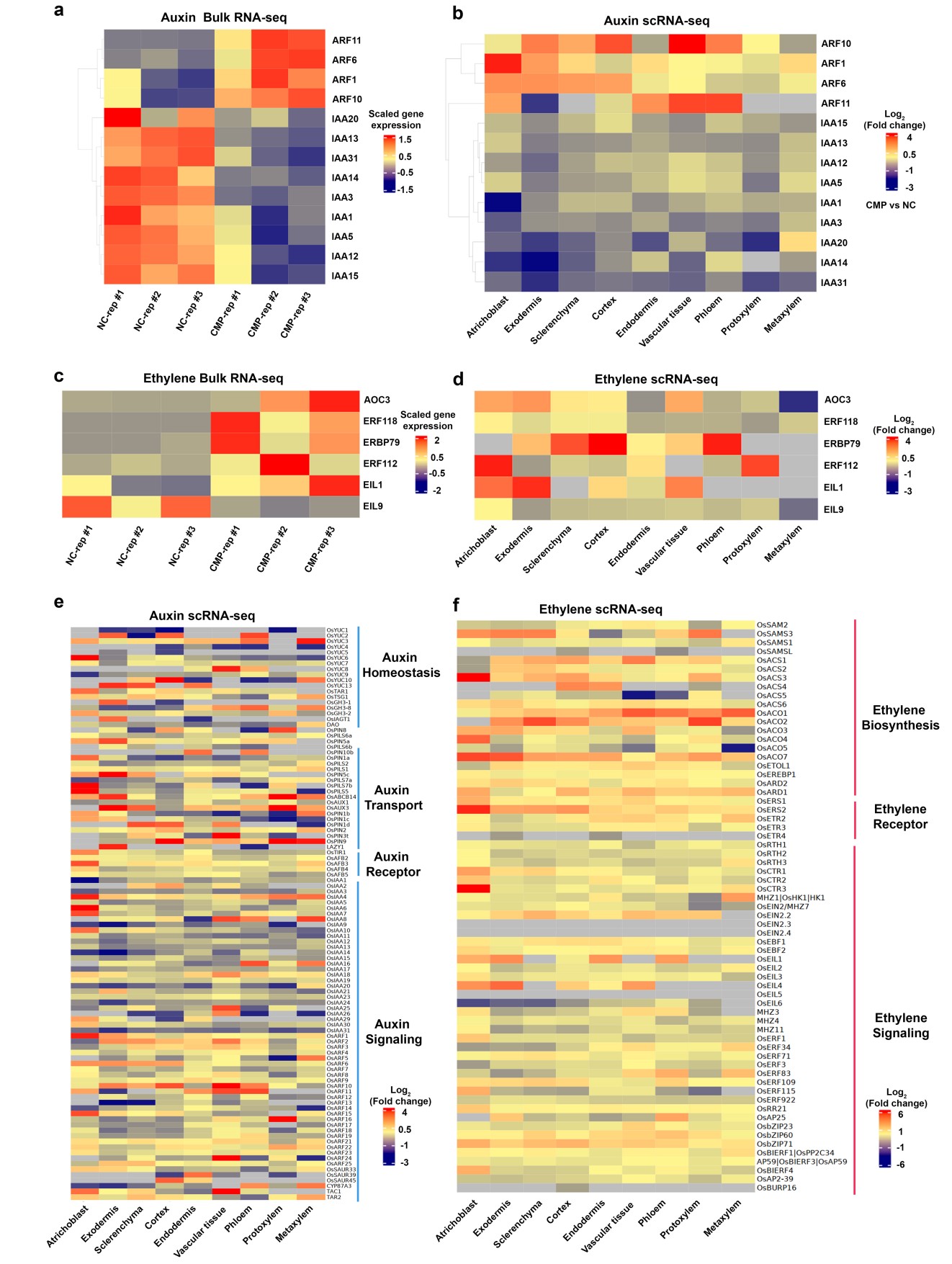

**Extended Data Fig. 9 |** See next page for caption.

**Extended Data Fig. 9 | Soil compaction induces auxin and ethylene signaling genes, but no cell type-specific induction patterns were detected. a**, Heatmap showing scaled expression of auxin signalling genes in non-compacted and compacted soil conditions in Xkitaake, as revealed by bulk RNA-seq. Bulk RNA-seq analysis revealed increased expression of genes encoding Auxin Response Factors (ARFs) and decreased expression of genes encoding auxin/indole-3-acetic acid (Aux/IAA) proteins, indicating enhanced auxin signalling in response to soil compaction. **b**, Heatmap showing differential expression (log$_2$ fold change) of auxin signalling genes in compacted soil conditions compared to non-compacted soils in Xkitaake as revealed by scRNA-seq analysis. scRNA-seq showed similar expression patterns for ARFs and Aux/IAAs, further supporting the activation of auxin signalling under soil compaction. **c**, Heatmap showing scaled expression of ethylene signalling genes in non-compacted and compacted soil conditions in Xkitaake, as revealed by bulk RNA-seq. Bulk RNA sequencing demonstrated increased expression of ethylene signalling components, suggesting enhanced ethylene responses under soil compaction. **d**, Heatmap showing differential expression (log$_2$ fold change) of ethylene signalling genes in compacted soil conditions compared to non-compacted soils in Xkitaake as revealed by scRNA-seq analysis.

scRNA-seq confirmed the upregulation of ethylene signalling components, further corroborating the activation of ethylene responses in response to soil compaction. **e**, Heatmap showing differential expression (log$_2$ fold change) of auxin pathway genes in compacted soil conditions compared to non-compacted soils in Xkitaake as revealed by scRNA-seq analysis. Cell type-specific expression patterns of genes involved in auxin homeostasis, transport, receptor activity, and downstream signalling were analyzed. No distinct cell type-specific patterns were observed. **f**, Heatmap showing differential expression (log$_2$ fold change) of ethylene pathway genes in compacted soil conditions compared to non-compacted soils in Xkitaake as revealed by scRNA-seq analysis. Cell type-specific expression patterns of genes involved in ethylene biosynthesis, perception, and downstream signalling were analyzed. While an overall increase in ethylene signalling-related gene expression was detected, no distinct cell type-specific patterns were observed. Bulk RNA-seq analysis was carried out using three independent biological replicates for non-compacted (NC-rep #1-3) and compacted (CMP-rep #1-3) soil conditions. Grey boxes mean that the gene was not detected during the comparative analysis. The complete list of gene ID and annotations are included in Supplementary Table 14.

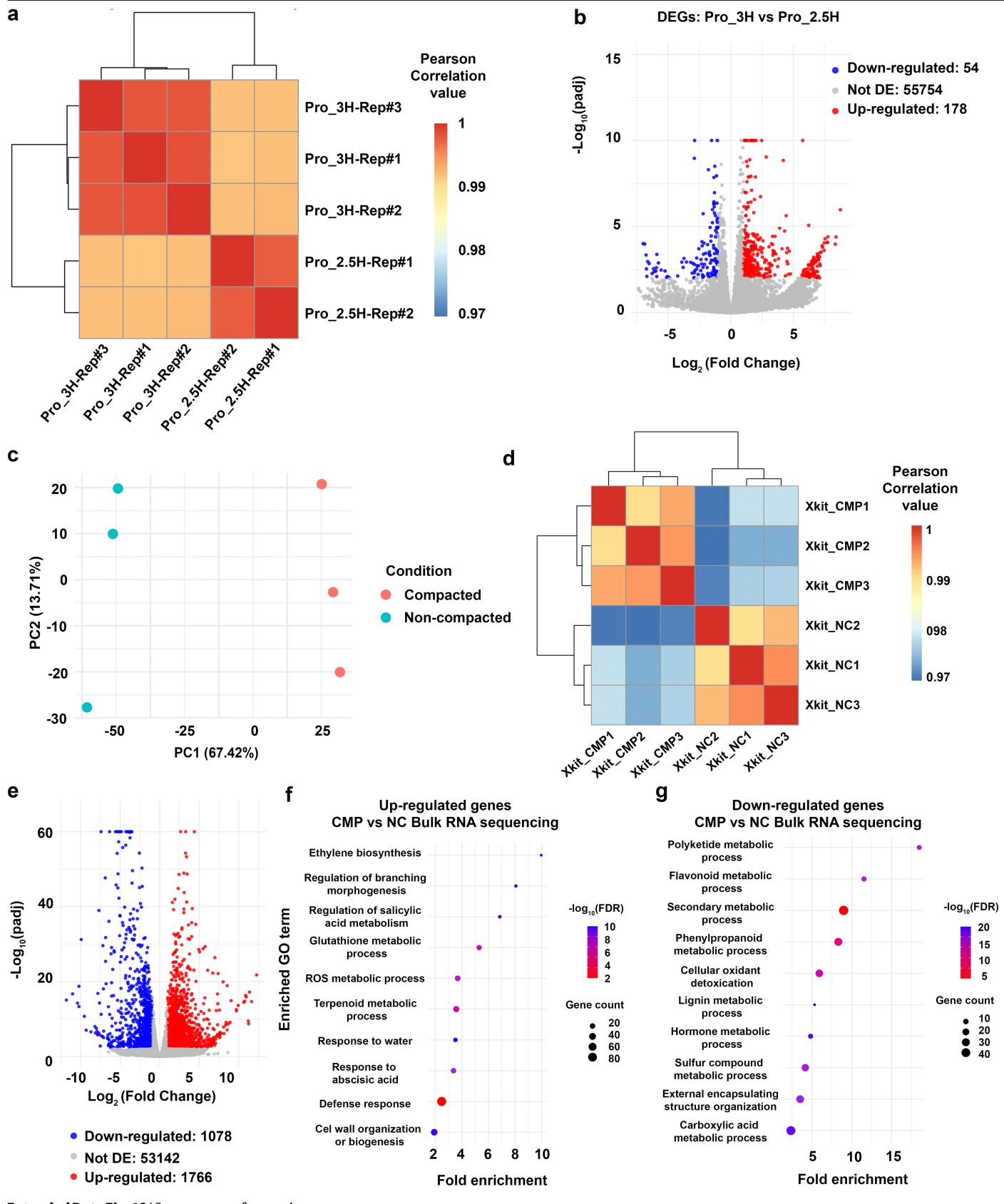

**Extended Data Fig. 10** | See next page for caption.

**Extended Data Fig. 10 | Bulk RNA sequencing validates the gene expression changes identified by single-cell RNA sequencing under compacted soil conditions. a**, The bulk RNA sequencing data for root tissues protoplasted for 2.5 h show a strong correlation with those for root tissues protoplasted for 3 h (Pearson correlation values > 0.99). **b**, Only a limited number of genes exhibit differential expression between root tissues protoplasted for 2.5 h and 3 h. To avoid introducing potential artifacts from protoplasting into our comparative analysis of scRNA-seq data from roots grown in gel and soil conditions, we excluded the 232 differentially expressed genes identified here from the scRNA-seq data analysis. The two-tailed Wald test with Benjamini–Hochberg FDR for multiple comparison correction was used for the p-value calculation. **c**, PCA plot showing clear separation of bulk RNA sequencing data for root samples grown under non-compacted soil and compacted soil conditions. **d**, Pearson correlation plot illustrating the distinct clustering of bulk RNA sequencing data for root samples grown under noncompacted (NC) and compacted (CMP) soil conditions. **e**, Volcano plot depicting the number of upregulated and downregulated genes under soil compaction as identified in the bulk RNA sequencing data. The two-tailed Wald test with Benjamini–Hochberg FDR for multiple comparison correction was used for the p-value calculation. **f**, Enriched GO terms for the upregulated genes in bulk RNA sequencing data in compacted soils compared to non-compacted soil conditions. Notably, both ABA and ethylene signalling pathways are induced. **g**, Enriched GO terms for the downregulated genes in bulk RNA sequencing data in compacted soils compared to non-compacted soil conditions. Notably, lignin metabolism is suppressed. The two-tailed Fisher's exact test with Benjamini–Hochberg FDR for multiple comparison correction was used for the p-value calculation of GO term analysis.

| | |
|---|---|

# Reporting Summary

## Statistics

For all statistical analyses, confirm that the following items are present in the figure legend, table legend, main text, or Methods section.

| n/a | Confirmed | |
|---|---|---|
| ☐ | ☒ | The exact sample size (*n*) for each experimental group/condition, given as a discrete number and unit of measurement |
| ☐ | ☒ | A statement on whether measurements were taken from distinct samples or whether the same sample was measured repeatedly |
| ☐ | ☒ | The statistical test(s) used AND whether they are one- or two-sided<br>*Only common tests should be described solely by name; describe more complex techniques in the Methods section.* |
| ☐ | ☒ | A description of all covariates tested |
| ☐ | ☒ | A description of any assumptions or corrections, such as tests of normality and adjustment for multiple comparisons |
| ☐ | ☒ | A full description of the statistical parameters including central tendency (e.g. means) or other basic estimates (e.g. regression coefficient) AND variation (e.g. standard deviation) or associated estimates of uncertainty (e.g. confidence intervals) |
| ☐ | ☒ | For null hypothesis testing, the test statistic (e.g. *F*, *t*, *r*) with confidence intervals, effect sizes, degrees of freedom and *P* value noted<br>*Give P values as exact values whenever suitable.* |
| ☒ | ☐ | For Bayesian analysis, information on the choice of priors and Markov chain Monte Carlo settings |
| ☒ | ☐ | For hierarchical and complex designs, identification of the appropriate level for tests and full reporting of outcomes |
| ☐ | ☒ | Estimates of effect sizes (e.g. Cohen's *d*, Pearson's *r*), indicating how they were calculated |

*Our web collection on statistics for biologists contains articles on many of the points above.*

## Software and code

Policy information about availability of computer code

| | |
|---|---|
| Data collection | For confocal microscopy data collection, we used Zen 2009 (version 6.0.0.303).For sequencing data collection, we mainly used the NovaSeq 6000, NovaSeq X and Nextseq 500 platforms.For spatial transcriptomics experiments, we used the Molecular Cartography platform developed by Resolve Biosciences. |
| Data analysis | For the analysis of bulk RNA-seq, we used STAR aligner_2.6.1b, UMI-Tools_1.1.2 and HTSeq-Count_2.0.0, and Deseq2_1.36.0.<br>For the analysis of scRNA-seq data, we used CellRanger mkfastq (v3.1.0, 10X Genomics), R version 4.2.0 (2022-04-22), R package COPILOT (PMID: 36181683) , together with major packages:<br>circlize (0.4.15), ComplexHeatmap (2.14.0), ggplot2 (3.4.2), ggrepel (0.9.3), cowplot (1.1.1), RColorBrewer (1.1-3), tidyr (1.3.0), dplyr (1.1.2), Seurat (3.1.5), scran (1.26.0), SingleCellExperiment (1.20.0), muscat (1.12.0), limma (3.54.0), gprofiler2 (0.2.2).<br><br>For the analysis of spatial transcriptomic data, we used ImageJ 1.52n.<br>For the analysis of confocal images, we used MorphoGraphX 2.0 r1-32-gf272c434.<br><br>Code availability:<br>We mainly adapted codes published in Hsu et al., 2022 (https://doi.org:10.1016/j.xpro.2022.101729), Stuart et al., 2019 (https://doi.org:10.1016/j.cell.2019.05.031), Crowell et al., 2020 (https://doi.org:10.1038/s41467-020-19894-4), and Kolberg et al., 2020 (https://doi.org:10.12688/f1000research.24956.2) for our scRNA-seq analysis. The adapted codes for analysing the scRNA-seq data are available at GitHub: https://github.com/zhumy09/scRNA-seq-for-rice |

For manuscripts utilizing custom algorithms or software that are central to the research but not yet described in published literature, software must be made available to editors and reviewers. We strongly encourage code deposition in a community repository (e.g. GitHub). See the Nature Portfolio guidelines for submitting code & software for further information.

# Data

Policy information about availability of data

All manuscripts must include a data availability statement. This statement should provide the following information, where applicable:

- Accession codes, unique identifiers, or web links for publicly available datasets
- A description of any restrictions on data availability
- For clinical datasets or third party data, please ensure that the statement adheres to our policy

All information supporting the conclusions are provided with the paper. scRNA-seq data for Xkitaake and Kitaake roots grown under gel and soil conditions is available at NCBI BioProject PRJNA640389 (GSE251706). scRNA-seq from Zhang et al., 2021 (PMID: 33824350) is available at NCBI BioProject PRJNA706435 and PRJNA706099.Bulk RNA-seq data for developmental stage annotation is available at NCBI BioProject PRJNA1082669 (GSE260671). Bulk RNA-seq data for protoplasting induced genes is available at NCBI BioProject PRJNA1194134 (GSE283509). Bulk RNA-seq data for Xkitaake roots grown under compacted and non-compacted soil conditions are available at NCBI BioProject PRJNA1193632 (GSE283428). Raw data for Spatial transcriptomics (Molecular Cartography) is provided in Supplementary data 4 (gel), 6 (non-compacted soils), 8 (compacted soils). Source Data for Main Figures and Extended Data Figures are provided in Supplementary Data 11 as separated excel files. Gene accession number information is available in Supplementary Table 14. Supplementary tables are provided with this manuscript. Supplementary data 1-11 are available on the Nature Figshare platform: https://doi.org/10.6084/m9.figshare.25146260. The processed scRNA-seq for gel-grown rice roots is now publicly accessible through a user-friendly platform hosted on Shiny: https://rice-singlecell.shinyapps.io/orvex_app/

# Research involving human participants, their data, or biological material

Policy information about studies with human participants or human data. See also policy information about sex, gender (identity/presentation), and sexual orientation and race, ethnicity and racism.

| | |
|---|---|
| Reporting on sex and gender | NA |
| Reporting on race, ethnicity, or other socially relevant groupings | NA |
| Population characteristics | NA |
| Recruitment | NA |
| Ethics oversight | NA |

Note that full information on the approval of the study protocol must also be provided in the manuscript.

# Field-specific reporting

Please select the one below that is the best fit for your research. If you are not sure, read the appropriate sections before making your selection.

☒ Life sciences ☐ Behavioural & social sciences ☐ Ecological, evolutionary & environmental sciences

For a reference copy of the document with all sections, see nature.com/documents/nr-reporting-summary-flat.pdf

# Life sciences study design

All studies must disclose on these points even when the disclosure is negative.

| | |
|---|---|
| Sample size | Sample sizes are based on the standard practice in plant biology research and the objectives of the experiments. For scRNA-seq, we included over 20,000 high-quality cells for each experimental conditions, which is a similar setup with recent scRNA-seq study in Arabidopsis roots (PMID: 36996230) and in rice roots (PMID: 33824350). Also this number of cells ensure a relatively unbiased capture of cells from different cell types. For spatial transcriptomic experiments, more than three rice root transverse sections with good mRNA detection were included for each experimental conditions. The reproducible patterns of gene detection support major experimental conclusions. For confocal imaging, over 3 root sections with reproducible cell phenotype or fluorescent dye staining patterns were included for each experimental conditions. |
| Data exclusions | For scRNA-seq, the low-quality cells which had low UMI counts or high protoplasting-inducible gene expression were excluded. For spatial transcriptomic experiment, the root sections with low mRNA detection were excluded. For confocal imaging, the root sections with obvious damages were excluded. |
| Replication | All results represented in the manuscript were replicated for at least three time. The number of cells and roots were specified in either the figure legends or the supplementary tables. |
| Randomization | The root sections used for scRNA-seq and spatial transcriptomics were harvested in at least three different experimental rounds. The root sections used for confocal imaging came from different roots of multiple experimental rounds. The culture conditions were kept consistent for different experimental set-up. All roots of a given genotype were taken from a single allele (Wild type or a single mutant). |

| Blinding | The cell annotation for scRNA-seq data was performed with unbiased processing codes. The procedures and relevant parameters of analyzing the spatial transcriptomic data and confocal images were kept consistent for different experimental conditions. |
|---|---|

# Reporting for specific materials, systems and methods

We require information from authors about some types of materials, experimental systems and methods used in many studies. Here, indicate whether each material, system or method listed is relevant to your study. If you are not sure if a list item applies to your research, read the appropriate section before selecting a response.

## Materials & experimental systems

| n/a | Involved in the study |
|---|---|
| ☒ | Antibodies |
| ☒ | Eukaryotic cell lines |
| ☒ | Palaeontology and archaeology |
| ☒ | Animals and other organisms |
| ☒ | Clinical data |
| ☒ | Dual use research of concern |
| ☐ | ☒ Plants |

## Methods

| n/a | Involved in the study |
|---|---|
| ☒ | ChIP-seq |
| ☒ | Flow cytometry |
| ☒ | MRI-based neuroimaging |

## Dual use research of concern

Policy information about dual use research of concern

### Hazards

Could the accidental, deliberate or reckless misuse of agents or technologies generated in the work, or the application of information presented in the manuscript, pose a threat to:

| No | Yes | |
|---|---|---|
| ☒ | ☐ | Public health |
| ☒ | ☐ | National security |
| ☒ | ☐ | Crops and/or livestock |
| ☒ | ☐ | Ecosystems |
| ☒ | ☐ | Any other significant area |

### Experiments of concern

Does the work involve any of these experiments of concern:

| No | Yes | |
|---|---|---|
| ☒ | ☐ | Demonstrate how to render a vaccine ineffective |
| ☒ | ☐ | Confer resistance to therapeutically useful antibiotics or antiviral agents |
| ☒ | ☐ | Enhance the virulence of a pathogen or render a nonpathogen virulent |
| ☒ | ☐ | Increase transmissibility of a pathogen |
| ☒ | ☐ | Alter the host range of a pathogen |
| ☒ | ☐ | Enable evasion of diagnostic/detection modalities |
| ☒ | ☐ | Enable the weaponization of a biological agent or toxin |
| ☒ | ☐ | Any other potentially harmful combination of experiments and agents |

# Plants

| | |
|---|---|
| Seed stocks | We used Rice transgenic line XKitaake, which was shared by Dr. Pamela C. Ronald (UC Davis) and mhz5, aba1, aba2 and proCSLD1-VENUS-N7 seeds, which were shared by Prof. Jinsong Zhang (CAS, China) and Dr. Guoqiang Huang (SJTU,China). |
| Novel plant genotypes | NA |
| Authentication | This transgenic lines has been published in PMID: 25841037 DOI: 10.1105/tpc.15.00080 (mhz5);PMID: 31775618 DOI: 10.1186/s12864-019-6262-4 (XKitaake); PMID: 35858424 DOI: 10.1073/pnas.2201072119 (aba1, aba2); PMID: 38653244 DOI: 10.1016/j.cub.2024.03.064 (proCSLD1-VENUS-N7) |

