## [Peer Review File · Nature]

Single-cell transcriptomics reveal how root tissues adapt to soil stress

Corresponding Author: Dr Bipin Pandey

Version 0:

Reviewer comments:

Referee #1

(Remarks to the Author)

In the manuscript entitled "Single-cell transcriptomics reveal how root tissues adapt to soil stress", Mingyuan Zhu and colleagues employ single-cell sequencing to explore the mechanisms that are associated with responses to compacted soil environments. Through this impressive effort, they find that genes involved in cell wall and barrier (suberin and lignin) remodeling, and in abscisic acid related processes are responsive to soil compaction in a cell-type specific manner. They cross-validate scRNA-seq clusters with spatial transcriptomics on root cross- and longitudinal sections. They furthermore experimentally show that in a mutant defective in ABA biosynthesis (mhz5) soil compaction dependent deposition of suberin and lignin is abolished. They then test the hypothesis that the lack of ABA dependent deposition of lignin and suberin leads to increased water loss and find support for this hypothesis as the mhz5 mutant does display significantly increased water loss when grown under compacted conditions and compared to wildtype.

This is beautiful and impressive work. It is written well and in a concise manner. It will be a highly useful resource to those who study root responses to soil compaction, and it constitutes exemplary work on how scRNA-seq can be utilized to illuminate important biological processes, and how it can be leveraged to derive comprehensive models and hypotheses. It reveals important (albeit mostly known) processes contributing to the response and presumably acclimation to soil compaction and hints towards a high relevance of ABA dependent barrier formation in responding to soil compaction. While I believe this is a significant study, there are several issues.

Major:

1. While this is a beautiful study, there are only limited novel insights on a mechanistic or conceptual level. The impressive amount of data and analyses highlight described relationship between responses to soil compaction and cell wall modification (e.g. see Schneider et al., 2021) and ABA (e.g. see Huang et al., 2022). Moreover, a model of the role of ABA and radial water loss was also published in Huang et al. There are also several studies that show ABA leads to increased lignin and suberin deposition.
2. The mhz5 mutant is used to test the hypothesis regarding ABA, lignin and suberin, as well as radial water loss. As ABA has several functions in the plant, it remains unclear whether a strong case can be made that the radial water loss due to the absence of lignin and suberin barriers is functionally connected to growth in compacted soils. This evidence might be strengthened if the authors refer to or provide the mhz5 root phenotype in soil in compacted and non-compacted conditions.
3. Validation of cluster assignment: Fig. S2/Spatial transcriptomics has multiple issues: The pericycle annotated genes don't look like being expressed in the pericycle as the pericycle is a ring inward from the endodermis. I am also not convinced about at the endodermis. Also, I don't see signal for one of the exodermis markers, one of the phloem markers, one of the sclerenchyma and one of the cortex layers. This should be carefully checked. Also, it would be helpful to provide in Fig. S2 a cartoon of the cell-types of the rice root to help readers.
4. Marker genes in Ext. Data Fig. 9: Much like for some of the images in Fig. S2, I couldn't see signals in several of the presented cross-sections. Overall, I don't think the conclusions based on the presented data in Ext. Data Fig. 9 are fully

justified.

Minor:

1. Fig. S2/Spatial transcriptomics: It is unclear whether signal in a subset of cells of the respective tissue is due to technical reasons or corresponds to something biological meaningful (cell sub-population).
2. As the staining-based evidence for suberin and lignin is presented as very important finding, it might be worthwhile to consider inclusion of a non-stain based method such as GCMS to quantitatively test changes of lignin and suberin levels.
3. "Gene Ontology (GO) analyses on these DEGs revealed the functional classes enriched in outer tissues of soil grown roots notably includes defence genes". What about the other categories?
4. It is unclear what L31 ff "The developmental stage annotation is unique in our scRNA-seq dataset, compared to the previously published ones" means, please elaborate.
5. Line 31 typo: "transcriptomic". Please correct.

Referee #2

(Remarks to the Author)

This MS reports gene expression profiles in different cell types of rice roots in response to compaction stress by using scRNA-seq and spatial transcriptomic approach. Some published scRNA-seq data from rice roots were also integrated. After extensive analyses, the authors found that the defense-related genes were enhanced in outer cell layers from soil-grown rice roots compared to the gel-grown roots. They further found that gene expression involved in the cell wall remodeling and barrier formation is activated after compaction stress, and ABA biosynthesis and response pathway may be activated for such a barrier formation and further prevention of water loss from rice roots. While this study provides some novel insights into the gene expression and pathways in response to soil and compaction stress at cell type levels in rice roots, the following points may be addressed further.

- 1, The authors integrated the previously published scRNA-seq datasets with their own datasets. While such integration is probable, the two sets of data are derived from two different cultivars, namely ZH11 and Kitaake, with different genetic background and different treatments. It should be mentioned that the authors used a Kitaake line transformed with the XA21 gene driven by an ubiquitin promoter. The XA21 is localized on plasma membrane and confers resistance to Xoo in rice including roots. Although the root development should be very similar anyway, the relevant gene expression patterns could be quite different between the transgenic line and a real non-transgenic control in response to protoplasting and/or environmental changes.
- 2, Protoplasting process may affect single cell response, which could be stronger than the actual root response to environments. The protoplasts seem to be 'naked' cells without their coat 'cell wall'. What significant responses may be activated during the protoplasting process compared to the normal cells in planta? These could be analyzed if data are available and/or discussed.
- 3, Fig. 1c-h: Patterns for the cell type markers in epidermis should be provided for comparison. Can the atrichoblast cells represent epidermal cells? Fig. 1j-p: The annotation of each gene should be provided for clarity at least in figure legends.
- 4, The rationale of the mRNA detection on root sections in Resolve Biosciences should be added in methods section in a bit more detail.
- 5, Fig 2a,b: It seems that the gel-grown roots have much more trichoblast cells and atrichoblast cells compared to the soil-grown roots. Does this mean that these cells were damaged during isolation of cells in soil-grown roots? These should be explained.
- 6, Fig. 2d, lower panel: It seems that the outer cell layers have more defense gene expression. However, considering that the authors used a Kitaake line transformed with the XA21 gene, and the XA21 gene is actually involved in defense/disease responses, it is quite possible that the observed defense gene expression is due to the XA21 gene functions but not related to the soil-grown condition, although the gel-grown roots were compared. To solve this issue, the authors may want to check other published scRNA-seq data from rice and/or Arabidopsis to see if the defense gene is related to soil growth condition. Also, the authors may check and compare the defense gene expression in roots of the XA21-transgenic line and a non-transgenic control under both gel-grown and soil condition to see if there is any difference in expression.
- 7, Page 5, lower part: The Xylanase inhibitors are actually defense players, which are plant cell wall proteins largely distributed in monocots that inhibit the hemicellulose degrading activity of microbial xylanases. These should be mentioned in the text. Since the compact soil promotes root expansion, cellulose synthase gene and/or xyloglucan biosynthesis genes should be activated. Are these included in the scRNA-seq data? If have, these should be added to Fig. 3.
- 8, Fig. 3a: The number of trichoblast cells in roots from compact soils appeared to be much less than the gel-grown roots. The authors may want to explain this. Is it a real situation or an artifact during treatments?

9, Fig. 3e,f: The GO term 'response to ABA' is enriched in exodermis cells. Some of these genes should be listed in Fig. 3f for comparison with those from the phloem-pericycle to see the difference. The overall expression of both the ABA biosynthesis, signaling and responsive genes should be evaluated in these cell types, and then we may know the direction of the communication. I would say that generally the outer cell layers should first sense the surrounding compact soil situations and then transmit the signal to the internal parts of roots. The conclusion in p. 6, the last two lines in the second paragraph, should be clarified.

10, The authors pay more attention to the water stress that roots may meet in compact soil. However, since both the non-compact soil and compact soil are water-saturated during rice root growth for only 2-3 days, water stress should not be a big problem for roots anyway. In contrast, I would suggest that the mechanical pressure/stress should be a strong factor for the DEGs observed in these cell types. The authors should also pay attention to this part.

11, The same team has published a work regarding to the role of ethylene signaling and compaction stress in rice roots (Science, 2021). The authors may want to analyze the relevant changes to see if ethylene has any correlation with the cell types in response to compaction stress.

12, Fig. 4k: Although it is possible that ABA plays a role in prevention of water loss in roots, it is also more likely that ABA plays an essential role in protection and support of root growth for mechanical stress in compact soil condition, especially with the water-saturated soil and rice seedlings grown for only 2-3 days, as described in the Methods section.

13, In the first paragraph of Discussion, the authors mentioned that the immune responses are strong in outer cell layers. I would suggest that more tests may be performed using rice roots to see if the difference is still present considering that the authors used the XA21-transgenic line and XA21 is a disease resistance gene.

14, Throughout the MS and figures, where the gene code is presented, a gene annotation is better added for clarity.

15, Ext. Data Fig. 11e-g: 'gens' should be 'genes'? Please check and correct.

16, Ext. Data Fig. 13: A typical map of different cell types in Ext. Data Fig. 6a may be included for easy comparison. It seems that some ABA biosynthesis genes in outer cell layers, cortex and mature roots regions are also enriched.

Referee #3

(Remarks to the Author)

This manuscript presents a transcriptional atlas of rice roots under different soil compaction environments (agar, low and high compaction in soil). The authors interrogate these atlases for cell type-specific transcriptional signals that may be responsible for how roots adapt to higher compaction environments, as well as compare sterile and non-sterile conditions. The authors validate their atlas using spatial transcriptomics, as well as explore the possibility of ABA's effect in mediating suberin and lignin formation in the root.

The atlas itself is of high quality and the authors do a good job using spatial transcriptomics in the root as a complementary assay. However, I have a few concerns regarding some of the novelty presented in the manuscript, as well as the biological validation of ABA's role in suberin deposition. My comments are as follows:

1. The authors use protoplasting to build their rice atlases. Protoplasting is unlikely to change the transcriptional signatures that define cell type identity. However I do worry that the stripping of the cell wall and isolating cells in liquid media for over 30 min will change transcriptional signals underlying environmental responses (such as soil compaction). Furthermore, it appears that protoplasting from gel grown roots was for 30 min, and soil was for 3 hr, which compromises the comparison between the two datasets. Similarly, if outer layer cells are protoplasted first, this may be the reason these cell types have more differentially expressed genes. Sequencing the nuclei of flash-frozen tissue will overcome these hurdles. Similarly, control experiments with flash-frozen bulk tissue could be done to indicate that the transcriptional signals the authors detect are bona fide.

2. The section "scRNA-seq reveals soil-grown roots primarily exhibited expression changes in outer tissues" compares agar-grown root transcriptomes to that of soil-grown. One of the major findings presented is that soil-grown root systems have a higher occurrence of defense gene expression, particularly in the outer roots. The authors correctly point out that roots grown in soil will be exposed to biotic signals, and thus will likely have these sorts of expression patterns compared to sterile environments. Indeed this result is not that surprising, and likely could have been discovered with a simple bulk-RNA-seq experiment. As a reviewer I wonder whether a stronger narrative may lie within the other ~11,000 DEGs across the cell types they detect.

3. The section "Compaction stress upregulates ABA and barrier formation genes in stele and exodermis" compares two different transcriptomes grown at different soil densities. I believe the analysis itself is solid, and commend the authors for verifying marker gene expression using spatial sequencing. I think it is also interesting that each cell type appears to have different amounts of differential expression. However I see three weaknesses with this section that I think the authors could address:

a. The first is that the number of DEGs presented in 3a does not appear to be normalized to the number of nuclei sequenced. It would be expected that the exodermis would have more DEGs than the pericycle, for example, because the exodermis has

greater statistical power in which to detect DEGs in the first place. Normalizing by cell number could help overcome this effect.

b. The over-reliance on GO terms and specific marker genes does a disservice to the authors. These findings are more confirmatory of what would be expected, rather than anything novel. Given there are ~7k DEGs among the cell types, the authors might present on how these patterns of differential expression vary between treatments as well as cell types.

c. I think the final claim of this section “Hence, our scRNA-seq dataset reveals how inner and outer root cell types communicate” is not supported by the data. It is an interesting hypothesis, but cannot be supported by the findings of this section alone.

4. The section “ABA promotes barrier formation to reduce water loss during soil compaction stress” presents a model of how ABA synthesized in phloem/vasculature tissue might end up coordinating the formation of lignin and suberin in the outer cell layers. I believe this is an intriguing hypothesis and the authors go some way to functionally validate it using the mh5 ABA biosynthesis mutant. However, I would recommend the authors perform additional functional tests to strengthen their findings. Specifically:

a. Mh5 is not only involved in ABA biosynthesis, but this mutant also displays reduced ethylene responses in roots. Given ethylene’s own role in responding to soil compaction, it might be worth disentangling these two hormones. This could be achieved by assessing other ABA biosynthesis mutants for changes in suberin/lignin deposition.

b. While there is an association with the ABA biosynthesis mutant and decreased lignin/suberin, this does not necessarily mean that ABA moves from inner to outer cell layers. The authors would need to provide additional evidence that such movement takes place.

Version 1:

Reviewer comments:

Referee #1

(Remarks to the Author)

The authors have addressed all issues that I raised. It is notable that in addition to their findings regarding compaction, they now also emphasize the unique cell-type specific responses between homogenous agar medium and soil medium, which are very interesting and remarkable. I commend the authors on their work. A few minor issues:

Line 7: “in outer root cell types with growth condition transition”; unclear what growth condition transition means.

Line 35 typo “soil conditons”

Referee #2

(Remarks to the Author)

The MS has been greatly improved with more experiments based on my previous comments. I have only one minor point. In Fig 4k, while the suberin and lignin accumulation provides water barriers for water loss prevention, they also support and protect root systems and plants under compaction stress. This should be briefly mentioned in figure legends and/or discussion part.

Referee #3

(Remarks to the Author)

I appreciate the efforts made and am satisfied with many of the author’s responses. In particular, the bulk-RNA-seq comparisons appear to agree well with what is discovered in the single-cell data. Please note that Extended Data Fig. 15 and Fig. 17 would benefit from greater attention to detail in their preparation. In particular, the figure legends provide an interpretation of the data, but not a description of the data itself - e.g., “Bulk RNA sequencing further supported the upregulation of biosynthesis genes”. Also, 15a second panel is missing x-axis labels. CMP and NC are not defined in Extended Data Fig. 17.

There also may be a miscommunication. My original comment was that the induction of biotic signals in soil environments compared to sterile environments was “not that surprising.” The authors’ reply indicating that I considered it “fascinating” is thus incorrect. To reiterate, in nature, plant roots never exist in sterile environments, and thus, pointing out that defense responses are upregulated in soil environments seems a little tenuous.

Furthermore, while considerable effort has been made (and is appreciated) to compare XKitaake vs. Kitaake single-cell transcriptomes, it instead leads to further confusion in what the reader is meant to glean from this section (Page 5, lines 26 to 44). The justification for using XKitaake over Kitaake in the first place does not appear to be mentioned in the text, and thus, why the comparison between the genotypes needs to be made is also unclear. If Kitaake is the appropriate line to be used then XKitaake should be removed from the analysis. Extended Data Fig. 10 seeks to demonstrate that the gene expression

differences between the two lines are minimal. While this may be true, Extended Data Fig. 10 D, E, F, G and H don't quite answer this question. Extended Data Fig 11's analysis of R gene expression is more helpful, however A and C are not normalized to the same colors, and thus comparing these two datasets is not visually possible. Until this is resolved, I still have major concerns about using this line.

Minor.

Finally, I could not locate where Figure 2g's hypothesis of "get nutrients in" vs. "keep stress out" is mentioned or explained in the main body of the text. It appears relevant to paragraphs on Page 6, lines 1 to 9. The purpose of this model figure is unclear. It's well established that nutrients flow into roots. It's also clear they need to withstand mechanical stress.

Version 2:

Reviewer comments:

Referee #3

(Remarks to the Author)

All of my comments have been adequately addressed.

We sincerely thank the editor and reviewers for taking the time to review our manuscript and for providing thoughtful and constructive comments. In response to these suggestions, we have made several significant improvements and additions to strengthen our revised manuscript. These new datasets include:

- (1) Single-cell RNA sequencing (scRNA-seq) of non-transgenic *Kitaake* roots grown either in gel or natural soil conditions. This revealed that, whilst the resistance transgene *XA21* (in the rice root material used in the original manuscript) did elevate defence response expression, *XA21* is not required for defence gene induction under soil conditions.
- (2) More comprehensive analysis of differentially expressed genes (DEGs) relevant to plant hormone signalling and cell wall remodeling in our scRNA-seq data to reveal cell type-specific transcriptomic response to soil compaction.
- (3) Bulk RNA sequencing (RNA-seq) for *X.Kitaake* roots grown in non-compacted soil (NC) and compacted soil (CMP) conditions. This new dataset supported our earlier findings obtained with scRNA-seq for induction of plant hormone and cell wall remodelling pathways in root tissues by soil compaction stress.

Also, Bulk RNA sequencing for root tissue protoplasted for 2.5 hours and 3 hours, validating that differences in protoplasting setups have minimal impact on gene expression in gel-grown and soil-grown roots.

- (4) Single nuclei sequencing of rice roots in gel, non-compacted soil and compacted soil conditions (Data included in this document).
- (5) Lignin quantification of roots grown \pm compacted soil conditions, independently confirmed enhanced accumulation of lignin after soil compaction stress.
- (6) Non-invasive Phonon imaging (Brillouin microscopy) on roots grown \pm compacted soil conditions, revealed stiffer cell walls in the root endodermal tissue after soil compaction stress.
- (7) Our scRNA-seq data is now publicly accessible through a user-friendly platform hosted on Shiny: https://rice-singlecell.shinyapps.io/orvex_app/

Please find below our point-to-point response to reviewers' comments.

Referees' comments:

Referee #1 (Remarks to the Author):

In the manuscript entitled “Single-cell transcriptomics reveal how root tissues adapt to soil stress”, Mingyuan Zhu and colleagues employ single-cell sequencing to explore the mechanisms that are associated with responses to compacted soil environments. Through this impressive effort, they find that genes involved in cell wall and barrier (suberin and lignin) remodeling, and in abscisic acid related processes are responsive to soil compaction in a cell-type specific manner. They cross-validate scRNA-seq clusters with spatial transcriptomics on root cross- and longitudinal sections. They furthermore experimentally show that in a mutant defective in ABA biosynthesis (mhz5) soil compaction dependent deposition of suberin and lignin is abolished. They then test the hypothesis that the lack of ABA dependent deposition of lignin and suberin leads to increased water loss and find support for this hypothesis as the mhz5 mutant does display significantly increased water loss when grown under compacted conditions and compared to wildtype.

This is beautiful and impressive work. It is written well and in a concise manner. It will be a highly useful resource to those who study root responses to soil compaction, and it constitutes exemplary work on how scRNA-seq can be utilized to illuminate important biological processes, and how it can be leveraged to derive comprehensive models and hypotheses. It reveals important (albeit mostly known) processes contributing to the response and presumably acclimation to soil compaction and hints towards a high relevance of ABA dependent barrier formation in responding to soil compaction.

We sincerely appreciate your thoughtful summary regarding our work.

Major:

1. While this is a beautiful study, there are only limited novel insights on a mechanistic or conceptual level. The impressive amount of data and analyses highlight the described relationship between responses to soil compaction and cell wall modification (e.g. see Schneider et al., 2021) and ABA (e.g. see Huang et al., 2022). Moreover, a model of the role of ABA and radial water loss was also published in Huang et al. There are also several studies that show ABA leads to increased lignin and suberin deposition.

Response:

While earlier studies have connected soil compaction with cell wall modification and ABA biosynthesis, these papers have exclusively focused at the root tissue to organ scales. Our manuscript provides novel cell and molecular scale insights and mechanisms by exploiting (for the very first time) single cell RNAseq and spatial transcriptomics in soil grown rice roots.

While Schneider et al., (PNAS, 118.6: e2012087118, 2021) reported lignin accumulation in the multiseriate cortical sclerenchyma (MCS) tissues and its association with improved maize root responses to hard soil, our new study reveals *cell type-specific* expression changes of cell wall remodelling, suberin and lignin biosynthesis genes (specifically in the exodermis and endodermis). We go further by demonstrating lignin and suberin deposition in the exodermis and endodermis is controlled by the abiotic stress signal ABA through (a) upregulation of this plant hormone's biosynthesis in inner root vascular tissues in response to compaction stress;

(b) where the lignified and suberised exodermis and endodermis act as root barriers for water loss. Additionally, we addressed the unique adaptation of rice, which lacks MCS, by highlighting enhanced suberin and lignin accumulation in the exodermis and endodermis to cope with soil compaction. These points represent important mechanistic insights that were previously not known or unproven and go beyond the current state of knowledge in the field.

Huang et al. (2022) examined the interaction of multiple hormones—ethylene, ABA, and auxin—in regulating root cortical expansion under soil compaction. Here, we demonstrate cell-group-specific ABA signaling patterns (**Fig. 3g-h, Extended Data Figure 18**), showing that ABA, rather than auxin or ethylene (**Extended Data Figure 23**), affect gene expression in a cell type specific manner in response to soil compaction. We have now strengthened the connection between soil compaction, ABA biosynthesis in vascular tissue, ABA signaling in outer cell layers, and subsequent suberin and lignin accumulation at exodermis and endodermis, which reduces water loss.

In summary, our scRNA-seq data identifies key molecular regulation in a cell type-specific manner, paving the way for future genetic engineering without sacrificing overall root growth.

2. The *mhz5* mutant is used to test the hypothesis regarding ABA, lignin and suberin, as well as radial water loss. As ABA has several functions in the plant, it remains unclear whether a strong case can be made that the radial water loss due to the absence of lignin and suberin barriers is functionally connected to growth in compacted soils. This evidence might be strengthened if the authors refer to or provide the *mhz5* root phenotype in soil in compacted and non-compacted conditions.

Response:

Thank you for raising this important point, which is missing in our previous version of manuscript. In our previously published paper (Huang et al., 2022, PNAS, <https://doi.org:10.1073/pnas.2201072119>), we have shown that *mhz5*, as well as other ABA biosynthesis mutants, *aba1* and *aba2*, all have relatively longer roots than WT in compacted soils (As shown in the figure below).

[FIGURE REDACTED]

Figure S1, [TEXT REDACTED]

To independently validate our original observations using *mhz5* to reveal how ABA regulated barrier formation in response to soil compaction, we have now included equivalent lignin and suberin imaging data for two additional ABA biosynthesis mutants *abal* and *aba2* grown \pm compacted soil stress (**Extended Data Figure 21**). These results confirm that perturbing ABA biosynthesis in 3 independent rice mutant loci all result in disruption of lignin and suberin barrier formation under compacted soil conditions.

Regarding the role of radial water loss in regulating root growth under compacted soils, we hypothesize that increased water loss triggers enhanced root elongation (i.e. to increase root surface area) which could provide an explanation for the longer root phenotype of the three ABA mutant lines described above.

3. Validation of cluster assignment: Fig. S2/Spatial transcriptomics has multiple issues: The pericycle annotated genes don't look like being expressed in the pericycle as the pericycle is a ring inward from the endodermis. I am also not convinced about at the endodermis. Also, I don't see signal for one of the exodermis markers, one of the phloem markers, one of the sclerenchyma and one of the cortex layers. This should be carefully checked. Also, it would be helpful to provide in Fig. S2 a cartoon of the cell-types of the rice root to help readers.

Response:

Thank you for bringing this to our attention. We acknowledge that several "Pericycle" markers are not expressed in the pericycle tissue. Additionally, our current cell type annotation does not encompass all the cell types within the vascular tissue. To address this, we have refined our cell type annotation by defining the non-conducting stele tissue—including the Pericycle, Procambium, and ground tissue elements within the stele—as "Vascular tissue" for subsequent annotations. This updated definition has been included in the Figure 1 legend.

Also, as suggested, we have now included zoomed-in images of the spatial transcriptomics data to clearly highlight the cell-specific expression of the marker genes. Also, as suggested, we have now included zoomed-in images of the spatial transcriptomics data to clearly highlight the cell-specific expression of the marker genes. These updates can be found in the revised figures (**Extended Data Figure 2,5 and 12**).

For marker genes with relatively low expression, we have provided further magnified views as insets to ensure clarity. Additionally, we really appreciate your suggestion of adding a schematic illustration of the rice root cell types to enhance reader's understanding and broaden the accessibility of the content. This illustration has been incorporated into the bottom-right corner of panel a, to improve the clarity of our data presentation.

4. Marker genes in Ext. Data Fig. 9: Much like for some of the images in Fig. S2, I couldn't see signals in several of the presented cross-sections. Overall, I don't think the conclusions based on the presented data in Ext. Data Fig. 9 are fully justified.

Response:

The lower mRNA detection in compaction soil grown root radial sections appears to be due to reduced fixation efficiency. Roots grown in compacted soils undergo radial expansion, enhanced barrier formation, and increased mucilage secretion, all of which likely hinders formaldehyde penetration into inner root cell layers. As a result, mRNA preservation efficiency is reduced, particularly for markers in the root stele tissues. Despite these challenges, we

successfully identified ~ 20 robust cell-type specific markers under compacted soil conditions, as detailed in **Supplementary Table 3** and **Supplementary Data 8**.

As suggested, we have included zoomed-in images of the spatial transcriptomics data to clearly highlight the cell-specific expression of the marker genes. For marker genes with relatively low expression levels, we have added further magnified views as insets to enhance clarity. These updates are reflected in the revised figures (**Extended Data Figure 2, 5 and 12**). By providing zoomed-in views of the spatial transcriptomic data for rice roots grown in compacted soils, we aim to provide clearer evidence to support the cell-type-specific expression of these markers, which is the key conclusion of our spatial transcriptomic analysis.

Minor:

1. Fig. S2/Spatial transcriptomics: It is unclear whether signal in a subset of cells of the respective tissue is due to technical reasons or corresponds to something biological meaningful (cell sub-population).

Response:

We observed that not all root cells or mRNAs may be well-preserved during sample preparation (as noted above), necessitating the use of multiple tissue sections to investigate gene expression both qualitatively and quantitatively. The detection of signals in a subset of cells within specific tissues is often influenced by the combination of relatively low marker gene expression and the preservation quality of cells during preparation. We have already added this consideration to the Methods section.

Despite these technical issues, we strongly agree with you that in many cases the signal in a subset of root cells has biological meaning. A good example of this is **Extended Data Fig. 3 k-m**. The root hair markers showed a developmental stage relevant expression order, indicating their involvement in the different stage of root hair formation. Another good example is the cortex markers. There is low expression of *RAII* in the outmost cell layer of cortex, indicating that there is potential subgroup within Cortex cells. We believe that the key to be confident with the potential biological meaning is the reproducibility of the data. We have added these discussions into our discussion section. Additionally, we plan to include quantitative analyses of mRNA levels across cells in future studies to provide a more robust interpretation of gene expression patterns.

2. As the staining-based evidence for suberin and lignin is presented as very important finding, it might be worthwhile to consider inclusion of a non-stain based method such as GCMS to quantitatively test changes of lignin and suberin levels.

Response:

As suggested, we now provide non-stain-based data for lignin quantification from WT and *mhz5* roots from compacted and noncompacted soil conditions (**Extended Data Figure 20**) a well-established Genesys 10 S UV-Vis (Thermo Scientific) system.

3. "Gene Ontology (GO) analyses on these DEGs revealed the functional classes enriched in outer tissues of soil grown roots notably includes defence genes". What about the other categories?

Response:

We first explored other defence-related genes beyond the R gene family and found enhanced expression of these genes as well. In addition to examining defence-related genes, we also analysed genes associated with nutrient uptake, prompted by the nutrient stress indicated by the enriched GO terms "Phosphorus metabolic process" and "Organonitrogen metabolic process." Furthermore, we investigated genes related to the "Cell wall organization" GO term and observed a trend of cell wall strengthening as the roots transitioned from gel conditions to natural soil conditions.

Integrating these findings, we propose a working model for how rice roots adapt to natural but heterogeneous soil environments. This strategy involves efficient nutrient uptake to support proper development while reinforcing defences to mitigate external stresses (Please check the revised **Figure 2f and Extended Data Figures 10g, h**).

4. It is unclear what L31 ff "The developmental stage annotation is unique in our scRNA-seq dataset, compared to the previously published ones" means, please elaborate.

Response:

In previously published scRNA-seq datasets, developmental stages are often inferred using pseudotime analysis. Pseudotime analysis is a computational approach that infers the trajectory of cellular differentiation or other dynamic processes based on gene expression profiles.

However, pseudotime analysis does not represent actual time; rather, it provides a relative ordering of cells based on gene expression similarities. This approach is heavily influenced by the selection of the starting point (root), which can impact the interpretation of results.

In our manuscript, we incorporated bulk RNA sequencing data from rice root tissues harvested at distinct developmental stages. We use genes enriched at specific developmental stages to annotate the developmental stages of our scRNA-seq dataset. This method aligns more directly with experimental developmental stage information.

As suggested by you, we have elaborated on this point in the main text for clarity (**Page 3, Line 38 – 44**).

5. Line 31 typo: "transcriptomic". Please correct.

Response:

Thank you for catching this typo. We have now corrected it.

Thank you once again for your constructive comments regarding the conceptual novelty, spatial transcriptomic data presentation, and the relationship between water loss prevention and root growth. We have revised our manuscript, accordingly, placing greater emphasis on the detected cell group-specific gene expression changes to enhance clarity and highlight the key findings.

Referee #2 (Remarks to the Author):

This MS reports gene expression profiles in different cell types of rice roots in response to compaction stress by using scRNA-seq and spatial transcriptomic approach. Some published scRNA-seq data from rice roots were also integrated. After extensive analyses, the authors found that the defense-related genes were enhanced in outer cell layers from soil-grown rice roots compared to the gel-grown roots. They further found that gene expression involved in the cell wall remodeling and barrier formation is activated after compaction stress, and ABA biosynthesis and response pathway may be activated for such a barrier formation and further prevention of water loss from rice roots. While this study provides some novel insights into the gene expression and pathways in response to soil and compaction stress at cell type levels in rice roots, the following points may be addressed further.

We greatly appreciate your positive comments highlighting the important findings of our manuscript.

1, The authors integrated the previously published scRNA-seq datasets with their own datasets. While such integration is probable, the two sets of data are derived from two different cultivars, namely ZH11 and Kitaake, with different genetic background and different treatments. It should be mentioned that the authors used a Kitaake line transformed with the XA21 gene driven by an ubiquitin promoter. The XA21 is localized on plasma membrane and confers resistance to Xoo in rice including roots. Although the root development should be very similar anyway, the relevant gene expression patterns could be quite different between the transgenic line and a real non-transgenic control in response to protoplasting and/or environmental changes.

Response:

Thank you for pointing out the effects of *XA21* may have had in our previous analysis. To address this issue, we have conducted single cell transcriptomic analysis of roots from non-transgenic *Kitaake* (without *XA21*) in gel vs soil conditions. When we checked for specific defence response genes, we were able to detect induced expression in *Kitaake* with the gel vs soil growth conditions. However, the outer cell layers vs inner cell layer pattern is not significant.

We also compared the single cell transcriptomic of *X. Kitaake* genotype with *Kitaake* under the non-compacted soil conditions. We also observed a strong enhancement of defence responsive genes when comparing the *X. Kitaake* and *Kitaake* single-cell transcriptome (**Extended Data Figure 11**). It indicates that *XA21* indeed enhances defence response genes.

To further check whether *XA21* is required for the defence response triggered by the gel to soil growth conditions, we also conducted bulk RNA sequencing for both *X. Kitaake* and *Kitaake* Gel vs Soil. Hence, even without the *XA21* transgene, the defence response is induced when growing in soil conditions.

We have incorporated these points into the main text: **Page 5, Line 27 - 45**.

2, Protoplasting process may affect single cell response, which could be stronger than the actual root response to environments. The protoplasts seems to be 'naked' cells without their coat 'cell wall'. What significant responses may be activated during the protoplasting process compared to the normal cells in planta? These could be analyzed if data are available and/or discussed.

Response:

We agree that protoplasting can impact gene expression. To minimize its confounding effects and ensure the robustness of the conclusions drawn from our single-cell RNA sequencing (scRNA-seq) analysis, we have excluded Protoplasting-induced genes.

As a standard step in scRNA-seq data processing, we identified protoplasting-induced genes using bulk RNA-seq (**Supplementary Table 2**). These genes were excluded from our analysis. Specifically, we filtered out protoplasting-induced genes during the integration of scRNA-seq data, ensuring they were not included in the differential expression (DE) analysis. To highlight the occurrence of this filtering process, we have revised the main text (**Page 3, Line 30-34**) and updated the Methods section (**Page 15, Line 10-15**).

To address your question directly, we have included GO term analysis data here. However, since protoplasting-induced genes were filtered out during the integration of scRNA-seq data, they should not impact the subsequent DE analysis or the major biological conclusions of our manuscript.

Figure. S2, Enriched GO terms for the protoplasting induced genes.

“Regulation of response to osmotic stress”, “Response to wounding” and “regulation of salicylic acid biosynthesis” are among the top enriched GO terms.

3, Fig. 1c-h: Patterns for the cell type markers in epidermis should be provided for comparison. Can the atrichoblast cells represent epidermal cells? Fig. 1j-p: The annotation of each gene should be provided for clarity at least in figure legends.

Response:

We acknowledge that using atrichoblast markers alone representing epidermal cells is not ideal. To address this, we have updated the FeaturePlot to include the expression of both atrichoblast and trichoblast markers (please see the revised **Figure 1c**). Additionally, we have replaced the spatial transcriptomic data of the "epidermal" marker with data showing both atrichoblast and

trichoblast markers (please see the revised **Figure 1k**). In the revised manuscript, the distinction between root hair and non-root hair cells in the epidermis should now be clearer.

We also agree that including gene annotations is essential for clarity. We have added gene annotations in the figure legends for all relevant figures.

4, The rationale of the mRNA detection on root sections in Resolve Biosciences should be added in methods section in a bit more detail.

Response:

As suggested, we have added an illustrative schematic in the **Extended Data Figure 2a**. It should better explain the experimental procedures of our spatial transcriptomic analysis.

We also added the details of the mRNA detection in both figure legend of **Extended Data Figure 2a**, as well as in the method part in **Page 18, Line 15-22**.

5, Fig 2a,b: It seems that the gel-grown roots have much more trichoblast cells and atrichoblast cells compared to the soil-grown roots. Does this mean that these cells were damaged during isolation of cells in soil-grown roots? These should be explained.

Response:

Although great care was taken while washing the soil attached to root tips, this is likely to lead to losses of trichoblast (root hair) cells. To test this, we conducted root trichoblast cell specific reporter image analysis in compacted and noncompacted soil conditions. We did not see differences in the number of cells expressing the *proCSLD1-VENUS-N7* reporter (**Extended Data Figure 8**). However, this potential artifact introduced by our experimental procedure has now been highlighted in the method section of the manuscript (**Page 14, Line 19-23**), as well as the figure legend for **Extended Data Figure 7**.

6, Fig. 2d, lower panel: It seems that the outer cell layers have more defence gene expression. However, considering that the authors used a Kitaake line transformed with the XA21 gene, and the XA21 gene is actually involved in defense/disease responses, it is quite possible that the observed defense gene expression is due to the XA21 gene functions but not related to the soil-grown condition, although the gel-grown roots were compared. To solve this issue, the authors may want to check other published scRNA-seq data from rice and/or Arabidopsis to see if the defense gene is related to soil growth condition. Also, the authors may check and compare the defense gene expression in roots of the XA21-transgenic line and a non-transgenic control under both gel-grown and soil condition to see if there is any difference in expression.

Response:

We again appreciate your concern regarding interference of *XA21* gene promoting defence related genes.

To address this point, we compared and single cell transcriptome analysis of non-transgenic (*Kitaake* background) roots in gel vs soil conditions. We see again the enrichment of defence related genes in non-transgenic genotype as well (**Extended Data Figure 11**). We also compared the single-cell transcriptomes of the *X. Kitaake* genotype with *Kitaake* under non-compacted soil conditions and observed a significant enhancement of defence response genes.

Our current conclusion is that while *XA21* is not essential for the enhanced defence response observed during the transition from sterilized gel to natural soil conditions, it does further amplify the defence response triggered by these growth condition differences.

7, Page 5, lower part: The Xylanase inhibitors are actually defense players, which are plant cell wall proteins largely distributed in monocots that inhibit the hemicellulose degrading activity of microbial xylanases. These should be mentioned in the text. Since the compact soil promotes root expansion, cellulose synthase gene and/or xyloglucan biosynthesis genes should be activated. Are these included in the scRNA-seq data? If have, these should be added to Fig. 3.

Response:

Thank you for highlighting the relevance of xylanase inhibitors to plant defence. We have added the following to the main text (**Page 6, Line 42-45**): "Xylanase inhibitors are important defence components, primarily found in the cell walls of monocots, where they inhibit the hemicellulose-degrading activity of microbial xylanases."

As suggested, we also examined the expression of cellulose synthase (CESA) genes and xyloglucan biosynthesis genes in our scRNA-seq data (**Extended Data Figure 15a, b**). Our analysis confirms the enhanced expression of both groups of genes. Interestingly, while CESAs were slightly induced in sclerenchyma and xylem cells, xyloglucan biosynthesis genes exhibited strong but less cell type-specific patterns. This suggests that different aspects of cell wall remodeling processes may involve distinct cell-type-specific regulatory mechanisms. We have incorporated these points into the main text: **Page 9, Line 44-45 and Page 10, Line 1-4**.

8, Fig. 3a: The number of trichoblast cells in roots from compact soils appeared to be much less than the gel-grown roots. The authors may want to explain this. Is it a real situation or an artifact during treatments?

Response:

The way how we handle the roots grown in soils for scRNA-seq is likely to lead to losses of trichoblast (root hair) cells. We did not see differences in the number of cells expressing the *proCSLD1-VENUS-N7* reporter between roots grown in gel and compacted soil conditions.

Our conclusion is that decreased trichoblast cells under compacted soil conditions compared to the gel-grown cells is an artefact caused during the protoplast isolation process.

We have incorporated this point into the main text **Page 4, Line 41-45** "*To confirm the trichoblast cell number under gel versus soil conditions, we imaged the expression of a root hair cell-specific marker line (proCSLD1::VENUS-N7), which showed highly similar expression patterns in both conditions (Extended Fig. 8).*" as well as the figure legend for **Extended Data Figure 13**.

9, Fig. 3e,f: The GO term 'response to ABA' is enriched in exodermis cells. Some of these genes should be listed in Fig. 3f for comparison with those from the phloem-pericycle to see the difference. The overall expression of both the ABA biosynthesis, signaling and responsive genes should be evaluated in these cell types, and then we may know the direction of the communication. I would say that generally the outer cell layers should first sense the surrounding compact soil situations and then transmit the signal to the internal parts of roots. The conclusion in p. 6, the last two lines in the second paragraph, should be clarified.

Response:

Thank you for your thoughtful suggestions on emphasizing the coordinated contributions of both external (exodermis) and internal (phloem-pericycle) cell groups to root adaptation under soil compaction.

As suggested, we have included heatmaps for both ABA biosynthesis and ABA response genes in the **Fig. 3g-h, Extended Data Figures 14c-d and 17a-b**. These heatmaps clearly demonstrate that ABA biosynthesis predominantly occurs in the inner cell layers, while ABA responses are primarily activated in the outer cell layers. This strongly supports a working model in which ABA is synthesized within the stele and moves radially with water flux, triggering responses in the outer cell layers. These responses lead to the accumulation of suberin and lignin in the two water barriers, reinforcing them and effectively preventing radial water loss.

Our scRNA-seq analysis, which simultaneously examines the transcriptomes of individual cells across all major cell types, provides strong evidence for cell type-specific responses and coordination among different cell groups in the root's adaptation to soil compaction. The working model, along with the concept you proposed regarding the outer cell layers sensing the external environment (Water stress-related genes also exhibit significantly stronger expression induction in the outer cell layers, as shown in **Extended Data Figure 17d** while we did not detect clear patterns for mechano-sensing genes, data not included), has been incorporated into the main text (**Page 8, Line 6-13**).

We have also revised the last sentences of the original p.6 to “Is the ABA-mediated radial water loss prevention functionally connected to root growth in compacted soils? Our previous study has revealed that *mhz5*, as well as other ABA biosynthesis mutants, *aba1* and *aba2*, all have relatively longer roots than WT in compacted soils¹⁸, we hypothesize that increased water loss triggers enhanced root elongation. This may be a direct consequence of impaired cortical radial expansion and potentially reflects a root strategy to rapidly explore water resources.” for better clarification.

10, The authors pay more attention to the water stress that roots may meet in compact soil. However, since both the non-compact soil and compact soil are water-saturated during rice root growth for only 2-3 days, water stress should not be a big problem for roots anyway. In contrast, I would suggest that the mechanical pressure/stress should be a strong factor for the DEGs observed in these cell types. The authors should also pay attention to this part.

Response:

We apologize for the incorrect statement in the methods section of our original manuscript. The soil (both compacted and non-compacted) used in our experiments were first saturated with water, then excess water was drained through gravitational pull. Therefore, the experiments were conducted under near-field capacity conditions, rather than fully saturated conditions. We have revised the methods section accordingly to clarify this important point.

Both our bulk RNA sequencing and scRNA-seq results revealed water stress as a significant GO factor under compacted soil conditions (**Extended Data Figure 24f**). Based on these findings, we are confident that water stress responses accurately represent plant status in compacted soil conditions.

In addition to the prevention of water loss, the induction of barrier formation may also provide enhanced mechanical stability to the root tips. To test this, we performed Brillouin imaging which revealed higher cell wall stiffness in the endodermis layer under compacted soil conditions (**Extended Data Figure 22**). Further, in our sister manuscript, our AFM and mathematical modelling also confirms the stiffness of endodermal cell layer in compacted soil conditions (Zhang et al., co-submitted).

11, The same team has published a work regarding to the role of ethylene signaling and compaction stress in rice roots (Science, 2021). The authors may want to analyze the relevant changes to see if ethylene has any correlation with the cell types in response to compaction stress.

Response:

Many thanks for raising this point. As suggested, we have included the comparative heatmap of ethylene biosynthesis and signalling related genes in compacted vs non-compacted soil conditions (**Extended Data Figure 23**) which revealed induction of multiple ethylene related signalling components (e.g. ERFs and EILs). Their induction was independently confirmed by our bulk RNA sequencing data comparing roots grown in non-compacted and compacted soil conditions. However, our scRNA-seq datasets did not reveal any cell type specific expression induction pattern for these ethylene signalling genes.

Please note: We also examined the expression of auxin signalling genes (**Extended Data Figures 23 a,b and e**). The trend is similar to ethylene: multiple response genes show enhanced expression during soil compaction, but no cell type-specific induction is observed.

Taken together, these findings suggest that ABA drives cell type-specific gene expression changes (e.g. barrier formation) in response to soil compaction, whereas auxin or ethylene responses control root growth inhibitory effects in multiple tissues,

12, Fig. 4k: Although it is possible that ABA plays a role in prevention of water loss in roots, it is also more likely that ABA plays an essential role in protection and support of root growth for mechanical stress in compact soil condition, especially with the water-saturated soil and rice seedlings grown for only 2-3 days, as described in the Methods section.

Response:

We apologize again for the mistaken statement in the methods section – please see response to point 10 above. Our results reveal soil compaction imposes both water and mechanical stress on root tissues. We have highlighted water stress, as it is directly linked to the reinforcement of barriers in the exodermis and endodermis, which we independently observed through transcript profiling, suberin and lignin specific dyes, direct measurement of lignin and radial water loss measurements in the ABA *mhz5* mutant.

13, In the first paragraph of Discussion, the authors mentioned that the immune responses are strong in outer cell layers. I would suggest that more tests may be performed using rice roots to see if the difference is still present considering that the authors used the XA21-transgenic line and XA21 is a disease resistance gene.

Response:

XA21 appears to enhance expression of defence-related genes in outer cell layers (see response to point 1, reviewer 2 above). Nevertheless, *XA21* is not required for elevated defence gene expression in soil versus gel conditions.

Specifically, we have compared single-cell transcriptome data of non-transgenic (*Kitaake*) roots grown in Gel vs. Soil conditions and observed enrichment of defence-related genes in the non-transgenic genotype as well (**Extended Data Figure 11**). Additionally, comparing the single-cell transcriptomes of *X. Kitaake* and *Kitaake* under non-compacted soil conditions revealed a significant enhancement of defence response genes (**Extended Data Figure 11**).

14, Throughout the MS and figures, where the gene code is presented, a gene annotation is better added for clarity.

Response:

We also agree that including gene annotations is essential for clarity. To address this, we have added gene annotations in both the main text and figure legends. For most genes, we provide either their assigned names or functional annotations for clarity. For genes without assigned names, we have included their putative functional annotations. We have also improved our gene ID/annotation summary table: **Supplementary Table 14**.

15, Ext. Data Fig. 11e-g: 'gens' should be 'genes'? Please check and correct.

Response:

We have now corrected this typo error.

16, Ext. Data Fig. 13: A typical map of different cell types in Ext. Data Fig. 6a may be included for easy comparison. It seems that some ABA biosynthesis genes in outer cell layers, cortex and mature roots regions are also enriched.

Response:

As suggested, we have included a rice root scRNA-seq UMAP with cell type annotation (the original Ext. Data Fig. 6a) in the Ext. Data Fig. 13 (now the new **Extended Data Figure 18**).

We agree that selected ABA biosynthesis genes are not exclusively expressed in the stele tissue. This could be due to their inherently broad expression patterns as well as experimental noise in the scRNA-seq data. Nevertheless, the induced expression of key ABA biosynthesis genes later in the pathway within stele tissues remains consistent. To clarify this, we have highlighted these stele tissues in the UMAPs presented in the revised Extended Data Fig. 13a (now updated as **Extended Data Figure 18**).

Thank you once again for your constructive comments regarding the effects of the transgene *XA21*, the mechanical stress induced by soil compaction, and the changes in epidermal cell proportions with varying growth conditions. In response, we have revised our manuscript by incorporating the *Kitaake* scRNA-seq analyses, placing greater emphasis on mechanical stress and cell wall remodeling genes, and examining the fluorescent reporter for root hair cells. These revisions have further refined the major conclusions of our study and strengthened our confidence in them.

Referee #3 (Remarks to the Author):

This manuscript presents a transcriptional atlas of rice roots under different soil compaction environments (agar, low and high compaction in soil). The authors interrogate these atlases for cell type-specific transcriptional signals that may be responsible for how roots adapt to higher compaction environments, as well as compare sterile and non-sterile conditions. The authors validate their atlas using spatial transcriptomics, as well as explore the possibility of ABA's effect in mediating suberin and lignin formation in the root.

The atlas itself is of high quality and the authors do a good job using spatial transcriptomics in the root as a complementary assay.

We greatly appreciate your positive comments about our single-cell RNA sequencing (scRNA-seq) and spatial transcriptomic analysis.

However, I have a few concerns regarding some of the novelty presented in the manuscript, as well as the biological validation of ABA's role in suberin deposition. My comments are as follows:

1. The authors use protoplasting to build their rice atlases. Protoplasting is unlikely to change the transcriptional signatures that define cell type identity. However I do worry that the stripping of the cell wall and isolating cells in liquid media for over 30 min will change transcriptional signals underlying environmental responses (such as soil compaction). Furthermore, it appears that protoplasting from gel grown roots was for 30 min, and soil was for 3 hr, which compromises the comparison between the two datasets. Similarly, if outer layer cells are protoplasted first, this may be the reason these cell types have more differentially expressed genes. Sequencing the nuclei of flash-frozen tissue will overcome these hurdles. Similarly, control experiments with flash-frozen bulk tissue could be done to indicate that the transcriptional signals the authors detect are bona fide.

Response:

We agree that protoplasting can impact expression but only of a small subset of genes. To minimize the confounding effects of protoplasting and ensure the robustness of the conclusions drawn from our scRNA-seq analysis, we have implemented the following:

(1) As a standard step in scRNA-seq data processing, we identified protoplasting-induced genes using bulk RNA-seq (**Supplementary Table 2**). These genes were excluded from our analysis. Specifically, we filtered out protoplasting-induced genes during the integration of scRNA-seq data, ensuring they were not considered in our differential expression (DE) analysis.

As you pointed out, the protoplasting conditions for gel-based and soil-based scRNA-seq were slightly different, with a half-hour time difference in the digestion treatment. To address this, we compared the bulk RNA transcriptomes of rice root tips treated under the two conditions: a 2.5-hour digestion for gel and a 3-hour digestion for soil-grown roots. This comparison revealed a high correlation between the two conditions (**Extended Data Figure 24 a-b**) and only a limited number of differentially expressed genes (**Supplementary Table 2**).

To further strengthen the comparison between gel- and soil-based scRNA-seq data, we excluded these differentially expressed genes from our DE analysis. This additional filtering step ensures that our comparisons are not biased by differences in protoplasting conditions.

(2) We validated our single cell transcriptomic results comparing the bulk RNA sequencing roots grown under non-compacted (NC) and compacted (CMP) soil conditions. Our bulk RNA sequencing mirrors the gene expression pattern observed in single cell transcriptomic data, showing induction of cell wall remodelling, ABA signalling, barrier formation related genes.

(3) To further validate the root responses observed in scRNA-seq of sterilized gel vs natural soil environments, we also conducted single-nucleus RNA sequencing (snRNA-seq) on flash-frozen root tissues. This analysis successfully identified major cell types and developmental stages, as shown in **Fig. S3 a-b** in our responses. However, the quality of the snRNA-seq data was noticeably lower than that of the scRNA-seq data, as indicated by reduced UMI counts **Fig. S3 c-d** and fewer detected genes.

The lower number of detected genes reduced our confidence in the DE and GO term enrichment analyses. Our DE analysis did not distinguish between snRNA-seq data for NC and CMP conditions (**Fig. S3 e**). We also only identified only a small number of upregulated genes when comparing snRNA-seq data from roots grown in gel and NC conditions. Moreover, we observed only slight enrichment of GO terms similar to those in the scRNA-seq data when comparing gel and NC conditions.

While our snRNA-seq data provide partial support for the observations and conclusions drawn from the scRNA-seq data, we have chosen not to include it in this manuscript due to its suboptimal quality at this stage.

In summary, we have filtered out the protoplasting induced genes in our data analysis processing. We have also conducted bulk RNA sequencing on flash-frozen *X. Kitaake* root tissues under non-compacted soil and compacted soil conditions. The results (DE genes) closely align with our single-cell transcriptomic findings, confirming that the detected transcriptional signals are bona fide.

Figure S3. Single-nucleus RNA sequencing (snRNA-seq) reveals major cell types and developmental stages in rice roots under varied growth conditions.

a, UMAP visualization of snRNA-seq data from rice primary roots grown in gel, non-compacted soils (NC), and compacted soils (CMP). Colors indicate cell type annotations.

b, UMAP displaying developmental stage annotations for rice primary roots in gel, NC, and CMP conditions based on snRNA-seq data.

c, UMI counts in snRNA-seq and single-cell RNA sequencing (scRNA-seq) data from rice primary roots grown in gel conditions. Yellow lines highlight the high UMI counts observed in snRNA-seq data, facilitating direct comparison between snRNA-seq and scRNA-seq results.

d, UMI counts in snRNA-seq and scRNA-seq data from rice primary roots grown in soil (NC and CMP combined). Yellow lines indicate the high UMI counts in snRNA-seq data, facilitating direct comparison between snRNA-seq and scRNA-seq results.

e, Differentially Expressed Genes (DEGs) between NC and CMP conditions across 7 annotated major cell types in rice primary roots. Cell types without detected DEGs are not shown. Numbers next to each bar represent the total DEGs in each cell type, with low DEG counts suggesting limited transcriptional differences between NC and CMP samples in these cell types.

f, Differentially Expressed Genes (DEGs) between NC soil and gel conditions for 9 major cell types and 2 developmental stage groups in rice primary roots. Numbers next to each bar indicate the total DEGs in each category.

g, “Cell wall organization” and “Hormone-mediated pathway” are the top enriched Gene Ontology (GO) terms for DEGs between NC and gel conditions. Additionally, slight enrichment was observed for “Vesicle-mediated transport,” “Regulation of defense responses,” and “Phosphorus metabolic process”—the top three GO terms for upregulated genes when transitioning from sterilized gel to natural soil environments.

2. The section “scRNA-seq reveals soil-grown roots primarily exhibited expression changes in outer tissues” compares agar-grown root transcriptomes to that of soil-grown. One of the major findings presented is that soil-grown root systems have a higher occurrence of defense gene expression, particularly in the outer roots. The authors correctly point out that roots grown in soil will be exposed to biotic signals, and thus will likely have these sorts of expression patterns compared to sterile environments. Indeed this result is not that surprising, and likely could have been discovered with a simple bulk-RNA-seq experiment. As a reviewer I wonder whether a stronger narrative may lie within the other ~11,000 DEGs across the cell types they detect.

Response:

We agree with you that the enhancement of defence responses in roots in natural soil versus gel conditions is fascinating. Please note that the enhanced expression of defence genes in outer root cell layers appears to be due to the presence of the *XA21* transgene in the original rice line we performed scRNA-seq (see our response to point 1 by reviewer 2). Nevertheless, *XA21* is not required for elevated defence gene expression in soil versus gel conditions.

As suggested, we further explored the differentially expressed genes (DEGs) across cell types when comparing gel and soil conditions. We focused on genes related to nutrient stress and uptake, as well as those involved in cell wall strengthening. Notably, we observed again an enrichment of induced expression in the outer cell layers for these genes. These results underscore the critical role of the outer cell layers in enabling plant roots to sense and respond to the heterogeneous external soil environment, including water, nutrient, and biotic factors.

We have updated Figure 2 to reflect these findings and revised our key conclusion for the gel vs. soil comparison, emphasizing the role of the outer cell layers as mechanisms to “take nutrients in” and “keep stress out.”

3. The section “Compaction stress upregulates ABA and barrier formation genes in stele and exodermis” compares two different transcriptomes grown at different soil densities. I believe the analysis itself is solid, and commend the authors for verifying marker gene expression using spatial sequencing. I think it is also interesting that each cell type appears to have different amounts of differential expression. However I see three weaknesses with this section that I think the authors could address:

a. The first is that the number of DEGs presented in 3a does not appear to be normalized to the number of nuclei sequenced. It would be expected that the exodermis would have more DEGs than the pericycle, for example, because the exodermis has greater statistical power in which to detect DEGs in the first place. Normalizing by cell number could help overcome this effect.

Response:

For the DE analysis, we used a published R package *muscat* to do the pseudobulk. Based on the *muscat* paper (<https://doi.org/10.1038/s41467-020-19894-4>), pseudobulk differential expression (DE) analysis inherently considers the number of cells in each subpopulation. When creating pseudobulk data, counts from individual cells are aggregated at the sample level for each subpopulation. This aggregation reflects the total number of cells in the subpopulation because it sums the expression counts across all cells in each subpopulation per sample.

As a result, methods like *limma-voom* (the gene expression comparison tool) that operate on these pseudobulk counts take into account the total library size, which is influenced by the number of cells. This means that the variability in cell numbers between subpopulations is already factored into the differential testing process, reducing the need for further normalization based on cell numbers.

Therefore, although having more cells does improve the power to detect DEGs in *muscat*, the *muscat* package's pseudobulk approach together with *limma*, already adjusts for cell number differences across samples to some extent during the aggregation process.

b. The over-reliance on GO terms and specific marker genes does a disservice to the authors. These findings are more confirmatory of what would be expected, rather than anything novel. Given there are ~7k DEGs among the cell types, the authors might present on how these patterns of differential expression vary between treatments as well as cell types.

Response:

We agree that placing too much emphasis on GO terms does not fully leverage the comprehensive insights provided by single-cell transcriptome analysis. To enhance the depth of our analysis and the novelty of our manuscript, we performed more detailed investigations into gene expression associated with various biological processes. Beyond simple up- or down-regulation trends, our scRNA-seq data reveal distinct cell-type-specific expression changes.

We analysed genes involved in water stress sensing in response to soil compaction (**Extended Data Figure 14c-d and Extended Figure 17d**). These exhibit significantly stronger expression induction in the outer cell layers compared to the inner cell layers, indicating that outer cell layers are more involved in the water stress sensing.

We also analysed genes relevant to cell wall organization in response to soil compaction, including: Suberin and lignin biosynthesis genes (**Extended Data Figure 17c**), which are strongly induced in the exodermis; Expansin genes (**Extended Data Figure 15c**), which are induced primarily in the exodermis and cortex; Cellulose synthase genes (**Extended Data Figure 15b**), which are slightly induced in sclerenchyma and xylem. All these expression induction patterns align well with observed root phenotypes under compacted soil conditions, implying that different cell groups employ different cell wall remodelling strategy to handle soil compaction stress.

Additionally, we performed a thorough examination of ABA, auxin, and ethylene-related genes in our revised manuscript (**Extended Data Figures 17 and 23**). The distinct expression patterns indicate that ABA, rather than auxin or ethylene, plays a key role in driving cell-type-specific gene expression in response to soil compaction.

In summary, our scRNA-seq data provide critical insights into the molecular regulation of root responses at the cell-type-specific level, pinpoint specific genes within larger gene families that are directly implicated in the root's response to compacted soil conditions and paving the way for future genetic engineering approaches to optimize root adaptation without compromising overall root growth. We have incorporated these key points into our main text.

c. I think the final claim of this section “Hence, our scRNA-seq dataset reveals how inner and outer root cell types communicate” is not supported by the data. It is an interesting hypothesis, but cannot be supported by the findings of this section alone.

Response:

We agree that the evidence supporting communication between inner and outer root cell types was insufficient in the previous version of the manuscript.

To address this, we have included heatmaps for both ABA biosynthesis and ABA response genes in **Figure 3g-h**. These heatmaps clearly show that ABA biosynthesis predominantly occurs in the inner cell layers, while ABA responses are primarily activated in the outer cell layers. This strongly supports a working model in which ABA is synthesized in the stele and moves radially with water flux, triggering responses in the outer cell layers. These responses result in the accumulation of suberin and lignin in the two water barriers, reinforcing them and effectively preventing radial water loss.

We have also revised this claim to “Hence, our scRNA-seq dataset demonstrates how compaction stress drives coordinated, cell-specific responses to stress signals, such as ABA, progressing from the inner to the outer root cell layers.” (**Page 7, Line 23-25**).

4. The section “ABA promotes barrier formation to reduce water loss during soil compaction stress” presents a model of how ABA synthesized in phloem/vasculature tissue might end up coordinating the formation of lignin and suberin in the outer cell layers. I believe this is an intriguing hypothesis and the authors go some way to functionally validate it using the mhz5 ABA biosynthesis mutant. However, I would recommend the authors perform additional functional tests to strengthen their findings. Specifically:

a. *Mhz5* is not only involved in ABA biosynthesis, but this mutant also displays reduced ethylene responses in roots. Given ethylene's own role in responding to soil compaction, it might be worth disentangling these two hormones. This could be achieved by assessing other ABA biosynthesis mutants for changes in suberin/lignin deposition.

Response:

Thank you for raising the important point that both ethylene and ABA are involved in the root response to soil compaction. As suggested, we provide lignin and suberin imaging of ABA biosynthetic mutants *aba1* and *aba2* in compacted soil conditions (**Extended Data Figure 21**) which shows that *mhz5*, *aba1* and *aba2* mutants all have attenuated barrier formation in compacted soil conditions.

b. While there is an association with the ABA biosynthesis mutant and decreased lignin/suberin, this does not necessarily mean that ABA moves from inner to outer cell layers. The authors would need to provide additional evidence that such movement takes place.

Response:

The scRNA-seq dataset for roots grown in compacted soil reveals genes in the final steps of ABA biosynthesis are upregulated in vascular cells (**Fig. 3f-g**) and ABA responsive genes are induced in exodermis cells (**Fig. 3h**), consistent with this signal moving to outer cell files to trigger barrier formation (as proposed in **Fig 4k**). As ABA biosynthesis machinery prominently expressed in vascular cells, it requires to move either symplastically or through apoplastic pathway to reach to the outer cell files. We hypothesise that based on our recent finding (Mehra et al., 2022, Science), where we have shown that ABA does moves from inner cell files to outer cell files. Unfortunately, such ABA biosensor (ABACUS) is currently not available in rice which will take considerable time to actually prove the inner to outer movement of ABA.

Thank you once again for your constructive comments regarding the effects of protoplasting, the focus on differentially expressed genes (DEGs) rather than GO terms, and the evidence supporting ABA movement across cell types. In response, we have revised our manuscript by incorporating bulk RNA-seq data, emphasizing specific genes rather than broad biological processes, and emphasizing more on the cell type-specific ABA response (updated Figure 3). These revisions have significantly enhanced the clarity of our findings.

Dear Editor,

In response to your invitation, we are re-submitting our revised manuscript entitled, "Single-cell transcriptomics reveal how root tissues adapt to soil stress" to *Nature*.

We have made the following revisions to figures and text requested by reviewers to further strengthen our manuscript:

- (1) We refined Figure 2 and Extended Data Figures 10, 11, and 15.
- (2) We edited the figure legends of Extended Data Figures 10,11, 15, 17 and 23.
- (3) We incorporated the rationale for using Xkitaake in our single-cell study in the main text and discussed the role of mechanical support in mitigating soil compaction stress in the discussion section.

Please also find below our point-to-point response to each reviewers' comments.

Referees' comments:

Referee #1 (Remarks to the Author):

The authors have addressed all issues that I raised. It is notable that in addition to their findings regarding compaction, they now also emphasize the unique cell-type specific responses between homogenous agar medium und soil medium, which are very interesting and remarkable. I commend the authors on their work.

Your thoughtful insights have been invaluable in strengthening our manuscript throughout the review process. Thank you!

A few minor issues:

Line 7: “in outer root cell types with growth condition transition”; unclear what growth condition transition means.

Response:

Thank you for highlighting this statement in the Abstract. We have revised the text to now read “the data revealed major expression changes in outer root cell types when comparing the single-cell transcriptomes of rice roots grown in gel versus soil conditions”. (Page 2, Line 7)

Line 35 typo “soil condtions”

Response:

Now corrected. (Page 6, Line 37)

Referee #2 (Remarks to the Author):

The MS has been greatly improved with more experiments based on my previous comments.

Thank you for your invaluable input during the review process, which has significantly improved our manuscript!

I have only one minor point. In Fig 4k, while the suberin and lignin accumulation provides water barriers for water loss prevention, they also support and protect root systems and plants under compaction stress. This should be briefly mentioned in figure legends and/or discussion part.

Response:

We agree that the induction of barrier formation also provide enhanced mechanical stability to the root tips. We have incorporated this point into the Figure 4k legend (Page 35, Lines 22–25) as follows:

“The enhanced accumulation of suberin and lignin in these water-impermeable barriers not only provides structural support and protection for root systems but also mitigates radial water loss in rice roots under soil compaction stress.”

We also expanded on this point in the Discussion section (Page 9, Lines 30–33) as follows:

“The accumulation of lignin and suberin in the exodermis and endodermis may also serve to enhance the mechanical stability of root tips. Indeed, our phonon imaging (Brillouin microscopy) provides direct evidence that rice roots reinforce cell wall rigidity at these barriers to support and protect root systems and plants under compaction stress.”

Referee #3 (Remarks to the Author):

I appreciate the efforts made and am satisfied with many of the author's responses. In particular, the bulk-RNA-seq comparisons appear to agree well with what is discovered in the single-cell data.

Thank you again for your insightful comments and constructive suggestions during the first round of review, which have been instrumental in strengthening our manuscript.

Please note that Extended Data Fig. 15 and Fig. 17 would benefit from greater attention to detail in their preparation. In particular, the figure legends provide an interpretation of the data, but not a description of the data itself - e.g., "Bulk RNA sequencing further supported the upregulation of biosynthesis genes".

Response:

In response to your helpful suggestion, we have now added a more detailed interpretation of the data in the figure legends for Extended Data Fig. 15 and Fig. 17. Specifically, we have now described the data, highlighted key pattern details, and discussed the potential biological significance of the observed gene expression changes. Please refer to Page 62-63 and Page 67 for more details.

Also, 15a second panel is missing x-axis labels. CMP and NC are not defined in Extended Data Fig. 17.

Response:

We have now added the X-axis label in Extended Data Figure 15a (Page 61) and clarified the definitions of CMP (Compacted soils) and NC (Non-compacted soils) in the Extended Data Fig. 17 figure legend (Page 67, Line 7).

There also may be a miscommunication. My original comment was that the induction of biotic signals in soil environments compared to sterile environments was "not that surprising." The authors' reply indicating that I considered it "fascinating" is thus incorrect. To reiterate, in nature, plant roots never exist in sterile environments, and thus, pointing out that defense responses are upregulated in soil environments seems a little tenuous.

Response:

We completely agree with the reviewer's point that the induction of biotic signals in soil environments, compared to sterile conditions, is expected given that plant roots naturally interact with microbes. In light of this, we have revised the discussion, i.e. removing "Strikingly," at Page 9, Line 2 to ensure our interpretation is appropriately framed.

Furthermore, while considerable effort has been made (and is appreciated) to compare XKitaake vs. Kitaake single-cell transcriptomes, it instead leads to further confusion in what the reader is meant to glean from this section (Page 5, lines 26 to 44). The justification for using XKitaake over Kitaake in the first place does not appear to be mentioned in the text, and thus, why the comparison between the genotypes needs to be made is also unclear. If Kitaake is the appropriate line to be used then XKitaake should be removed from the analysis.

Response:

We agree that the justification for use of the Xkitaake (and Kitaake) lines needs to be made in the main text, leading us to redraft this section of the paper as follows.

“In our study, we used the model rice variety Xkitaake to establish our single-cell RNA sequence resource given the wealth of functional resources available in this background including mutant collections¹⁵ and exploited in our recent study¹. However, Xkitaake is a transgenic line containing the *XA21* gene which encodes a plasma membrane-localized protein that confers resistance to *Xanthomonas oryzae* pv. *oryzae* (Xoo) in rice. To assess the potential influence of *XA21* on rice root gene expression, we conducted scRNA-seq on Kitaake genotype under both gel and soil conditions (Extended Data Fig. 10a-e, Cell type annotation revealed similar enrichment of DEGs in outer root cells as observed in Xkitaake (Extended Data Fig. 10a-c, Supplementary Table 8). We further validated the enriched GO terms through comparative scRNA-seq analysis of the Kitaake genotype. The GO term enrichment patterns and associated gene expression changes (Supplementary Table 9), related to nutrient homeostasis, cell wall integrity, hormone-mediated signalling, and vesicle-mediated transport were consistent between Xkitaake and Kitaake in the scRNA-seq analysis (Extended Data Fig. 10f-h).” (Page 5, Line 27-39).

Extended Data Fig. 10 seeks to demonstrate that the gene expression differences between the two lines are minimal. While this may be true, Extended Data Fig. 10 D, E, F, G and H don't quite answer this question. Extended Data Fig 11's analysis of R gene expression is more helpful, however A and C are not normalized to the same colors, and thus comparing these two datasets is not visually possible. Until this is resolved, I still have major concerns about using this line.

Response:

To address this important point, we have refined Extended Figure 10 and its legend (Please check Page 50–51 for details).

Through comparative single-cell analysis of Xkitaake roots grown in gel and soil conditions, we reached a key conclusion: nutrient uptake, cell wall strengthening (Fig. 2e, f), and defense responses (Extended Data Fig. 11a, b) are the major biological processes induced by heterogeneous soil conditions. These inductions primarily occur in outer cell layers.

Since Xkitaake carries the *XA21* transgene, which could influence gene expression under stress conditions; we aimed to further confirm whether the above conclusion remains valid in the Kitaake background.

Extended Data Fig. 10c shows a high correlation (>0.94 for the comparisons under the same growth conditions) between Xkitaake and Kitaake scRNA-seq profiles, supporting their overall similarity.

Extended Data Fig. 10d, e, and f further confirm the enhancement of nutrient uptake (d, e) and cell wall strengthening (d, f) in the Kitaake background, consistent with our findings in Xkitaake.

Additional analysis of defense-related gene expression in Kitaake (Extended Data Fig. 11 c and f) supports that *XA21* is not required for the elevated defense gene expression observed in roots grown in soil versus gel conditions.

As suggested, we have normalized the expression data and used a consistent heatmap scale to compare expression profiles in Extended Fig. 11a and c. They now look similar to each other. Please refer to Page 52 for further details.

Minor.

Finally, I could not locate where Figure 2g's hypothesis of "get nutrients in" vs. "keep stress out" is mentioned or explained in the main body of the text. It appears relevant to paragraphs on Page 6, lines 1 to 9. The purpose of this model figure is unclear. It's well established that nutrients flow into roots. It's also clear they need to withstand mechanical stress.

Response:

We have revised the manuscript to better clarify the purpose of the model figure, specifically refining Figure 2g and citing it when discussing how roots employ cell-group-specific strategies to adapt to heterogeneous soils (Page 6, lines 5-11).

"The outer cell layers are more responsive compared to the inner cell layers, reinforcing nutrient uptake (i.e. 'get nutrients in') and cell wall integrity, to facilitate root exploration for heterogeneous resources in soil (Fig. 2g). This cell layer-specific response also helps protect developing roots from abiotic and biotic signals (i.e. 'keep stress out') that are unevenly distributed in natural soils (Fig. 2g). These important insights highlight the benefit of applying single-cell profiling approaches on samples grown in a natural soil environment."

Thank you once again for your very constructive comments on the figure interpretation and the rationale for using Xkitaake in our single-cell experiments!